# SKRIPS v1.0: A regional coupled ocean–atmosphere modeling framework (MITgcm–WRF) using ESMF/NUOPC, description and preliminary results for the Red Sea

Rui Sun[1], Aneesh C. Subramanian[1], Arthur J. Miller[1], Matthew R. Mazloff[1], Ibrahim Hoteit[2], and Bruce D. Cornuelle[1]

[1]Scripps Institution of Oceanography, La Jolla, California, USA
[2]Physical Sciences and Engineering Division, King Abdullah University of Science and Technology (KAUST), Thuwal, Saudi Arabia

**Correspondence:** Rui Sun (rus043@ucsd.edu); Aneesh Subramanian (acsubram@ucsd.edu)

**Abstract.** A new regional coupled ocean–atmosphere model is developed and its implementation is presented in this paper. The coupled model is based on two open-source community model components: (1) MITgcm ocean model; (2) Weather Research and Forecasting (WRF) atmosphere model. The coupling between these components is performed using ESMF (Earth System Modeling Framework) and implemented according to National United Operational Prediction Capability (NUOPC) protocols. The coupled model is named the Scripps–KAUST Regional Integrated Prediction System (SKRIPS). The SKRIPS is validated with a real-world example by simulating a 30-day period including a series of extreme heat events occurring on the eastern shore of the Red Sea region in June 2012. The results obtained by using the coupled model, along with those in forced stand-alone oceanic or atmospheric simulations, are compared with observational data and reanalysis products. We demonstrate that the coupled model is capable of performing coupled ocean–atmosphere simulation, although all configurations of coupled and uncoupled models have good skill in modeling the heat events. In addition, a scalability test is performed to investigate the parallelization of the coupled model. The results indicate that the coupled model code scales well and the ESMF/NUOPC coupler accounts for less than 5% of the total computational resources in the Red Sea test case. The coupled model, documentation, and tutorial cases used in this work are available at https://library.ucsd.edu/dc/collection/bb1847661c, and the source code is maintained at https://github.com/iurnus/scripps_kaust_model.

## 1 Introduction

Accurate and efficient forecasting of oceanic and atmospheric circulation is essential for a wide variety of high-impact societal needs, including extreme weather and climate events (Kharin and Zwiers, 2000; Chen et al., 2007), environmental protection and coastal management (Warner et al., 2010), management of fisheries (Roessig et al., 2004), marine conservation (Harley et al., 2006), water resources (Fowler and Ekström, 2009), and renewable energy (Barbariol et al., 2013). Effective forecasting relies on high model fidelity and accurate initialization of the models with the observed state of the coupled ocean–atmosphere system. Although global coupled models are now being implemented with increased resolution, higher-resolution regional

coupled models, if properly driven by the boundary conditions, can provide an affordable way to study air–sea feedback for frontal-scale processes.

A number of regional coupled ocean–atmosphere models have been developed for various goals in the past decades. An early example of building a regional coupled model for realistic simulations focused on accurate weather forecasting in the Baltic Sea (Gustafsson et al., 1998; Hagedorn et al., 2000; Doscher et al., 2002), and showed that the coupled model improved the SST (Sea Surface Temperature) and atmospheric circulation forecast. Enhanced numerical stability in the coupled simulation was also observed. These early attempts were followed by other practitioners in ocean-basin-scale climate simulations (e.g. Huang et al., 2004; Aldrian et al., 2005; Xie et al., 2007; Seo et al., 2007; Somot et al., 2008; Fang et al., 2010; Boé et al., 2011; Zou and Zhou, 2012; Gualdi et al., 2013; Van Pham et al., 2014; Chen and Curcic, 2016; Seo, 2017). For example, Huang et al. (2004) implemented a regional coupled model to study three major important patterns contributing to the variability and predictability of the Atlantic climate. The study suggested that these patterns originate from air–sea coupling within the Atlantic Ocean or by the oceanic responses to atmospheric internal forcing. Seo et al. (2007) studied the nature of ocean–atmosphere feedbacks in the presence of oceanic mesoscale eddy fields in the eastern Pacific Ocean sector. The evolving SST fronts were shown to drive an unambiguous response of the atmospheric boundary layer in the coupled model, and lead to model anomalies of wind stress curl, wind stress divergence, surface heat flux, and precipitation that resemble observations. This study helped substantiate the importance of ocean–atmosphere feedbacks involving oceanic mesoscale variability features.

In addition to basin-scale climate simulations, regional coupled models are also used to study weather extremes. For example, the COAMPS (Coupled Ocean/Atmosphere Mesoscale Prediction System) was applied to investigate idealized tropical cyclone events (Hodur, 1997). This work was then followed by other realistic extreme weather studies. Another example is the investigation of extreme bora wind events in the Adriatic Sea using different regional coupled models (Loglisci et al., 2004; Pullen et al., 2006; Ricchi et al., 2016). The coupled simulation results demonstrated improvements in describing the air–sea interaction processes by taking into account ocean surface heat fluxes and wind-driven ocean surface wave effects (Loglisci et al., 2004; Ricchi et al., 2016). It was also found in model simulations that SST after bora wind events had a stabilizing effect on the atmosphere, reducing the atmospheric boundary layer mixing and yielding stronger near-surface wind (Pullen et al., 2006). Regional coupled models were also used for studying the forecasts of hurricanes, including hurricane path, hurricane intensity, SST variation, and wind speed (Bender and Ginis, 2000; Chen et al., 2007; Warner et al., 2010).

Regional coupled modeling systems also play important roles in studying the effect of surface variables (e.g., surface evaporation, precipitation, surface roughness) in the coupling processes of oceans or lakes. One example is the study conducted by Powers and Stoelinga (2000), who developed a coupled model and investigated the atmospheric frontal passages over the Lake Erie region. Sensitivity analysis was performed to demonstrate that taking into account lake surface roughness parameterization in the atmosphere model can improve the calculation of wind stress and heat flux. Another example is the investigation by Turuncoglu et al. (2013), who compared a regional coupled model with uncoupled models and demonstrated the improvement of the coupled model in capturing the response of Caspian Sea levels to climate variability.

In the past ten years, many regional coupled models have been developed using modern model toolkits (Zou and Zhou, 2012; Turuncoglu et al., 2013; Turuncoglu, 2019) and include waves (Warner et al., 2010; Chen and Curcic, 2016), sediment

transport (Warner et al., 2010), sea ice (Van Pham et al., 2014), and chemistry packages (He et al., 2015). However, it is still desirable and useful to develop a new coupled regional ocean–atmosphere model implemented using an efficient coupling framework and with state estimation capabilities. The goal of this work is to (1) introduce the design of a newly developed regional coupled ocean–atmosphere modeling system, (2) describe the implementation of the modern coupling framework,

(3) validate the coupled model using a real-world example, and (4) demonstrate and discuss the parallelization of the coupled model. In the coupled system, the oceanic model component is the MIT general circulation model (MITgcm) (Marshall et al., 1997) and the atmospheric model component is the Weather Research and Forecasting (WRF) model (Skamarock et al., 2005). To couple the model components in the present work, the Earth System Modeling Framework (ESMF) (Hill et al., 2004) is used because of its advantages in conservative re-gridding capability, calendar management, logging and error handling, and

parallel communications. The National United Operational Prediction Capability (NUOPC) layer in ESMF (Sitz et al., 2017) is also used between model components and ESMF. Using the NUOPC layer can simplify the implementation of component synchronization, execution, and other common tasks in the coupling. The innovations in our work are: (1) we use ESMF/NUOPC, which is a community supported computationally efficient coupling software for earth system models, and (2) we use MITgcm together with WRF. The resulting coupled model is being developed as a coupled forecasting tool for coupled data assimilation

and subseasonal to seasonal (S2S) forecasting. By coupling WRF and MITgcm for the first time with ESMF, we can provide an alternative regional coupled model resource to a wider community of users. These atmospheric and oceanic model components have an active and well-supported user-base.

After implementing the new coupled model, we demonstrate it on a series of heat events that occurred on the eastern shore of the Red Sea region in June 2012. The simulated surface variables of the Red Sea (e.g., sea surface temperature, 2-m

temperature, and surface heat fluxes) are examined and validated against available observational data and reanalysis products. To demonstrate the coupled model can perform coupled ocean–atmosphere simulations, the results are compared with those obtained using stand-alone oceanic or atmospheric models. This paper focuses on the technical aspects of the SKRIPS, and is not a full investigation of the importance of coupling for these extreme events. In addition, a scalability test of the coupled model is performed to investigate its parallel capability.

The rest of this paper is organized as follows. The description of the individual modeling components and the design of the coupled modeling system are detailed in Section 2. Section 3 introduces the design of validation experiment and the validation data. Section 4 discusses the preliminary results in the validation test. Section 5 details the parallelization test of the coupled model. The last section concludes the paper and presents an outlook for future work.

## 2   Model Description

The newly developed regional coupled modeling system is introduced in this section. The general design of the coupled model, descriptions of individual components, and ESMF/NUOPC coupling framework are presented below.

## 2.1 General design

The schematic description of the coupled model is shown in Fig. 1. The coupled model is comprised of five components: oceanic component MITgcm, atmospheric component WRF, MITgcm–ESMF interface, WRF–ESMF interface, and ESMF coupler. They are to be detailed in the following sections.

5     The coupler component runs in both directions: (1) from WRF to MITgcm, and (2) from MITgcm to WRF. From WRF to MITgcm, the coupler collects the surface atmospheric variables (i.e., radiative flux, turbulent heat flux, wind velocity, precipitation, evaporation) from WRF and updates the surface forcing (i.e., net surface heat flux, wind stress, freshwater flux) to drive MITgcm. From MITgcm to WRF, the coupler collects ocean surface variables (i.e., SST and ocean surface velocity) from MITgcm and updates them in WRF as the bottom boundary condition. Re-gridding the data from either model component

10   is performed by the coupler, in which various coupling intervals and schemes can be specified by ESMF (Hill et al., 2004).

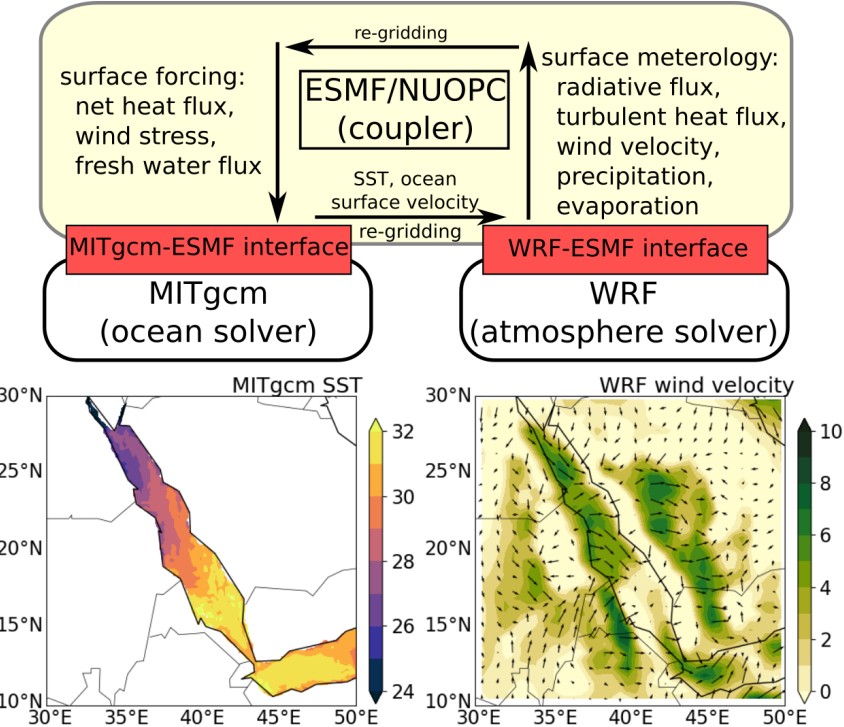

**Figure 1.** The schematic description of the coupled ocean–atmosphere model. The white blocks are the oceanic and atmospheric components; the red blocks are the implemented MITgcm–ESMF and WRF–ESMF interfaces; the yellow block is the ESMF/NUOPC coupler. From WRF to MITgcm, the coupler collects the surface atmospheric variables (i.e., radiative flux, turbulent heat flux, wind velocity, precipitation, evaporation) and updates the surface forcing (i.e., net surface heat flux, wind stress, freshwater flux) to drive MITgcm. From MITgcm to WRF, the coupler collects ocean surface variables (i.e., SST and ocean surface velocity) and updates them in WRF as the bottom boundary condition.

## 2.2 MITgcm Ocean Model

The MITgcm (Marshall et al., 1997) is a 3-D, finite-volume, general circulation model used by a broad community of researchers for a wide range of applications at various spatial and temporal scales. The model code and documentation, which are under continuous development, are available on the MITgcm webpage http://mitgcm.org/. The 'Checkpoint 66h' (June 2017) version of MITgcm is used in the present work.

The MITgcm is designed to run on high-performance computing (HPC) platforms and can run in non-hydrostatic and hydrostatic modes. It integrates the primitive (Navier-Stokes) equations, under the Boussinesq approximation, using finite volume method on a staggered 'Arakawa C-grid'. The MITgcm uses modern physical parameterization schemes for subgrid-scale horizontal and vertical mixing and tracer properties. The code configuration includes build-time C pre-processor (CPP) options and run-time switches, which allow for great computational modularity in MITgcm to study a variety of oceanic phenomena (Evangelinos and Hill, 2007).

To implement the MITgcm–ESMF interface, we separate the MITgcm main program into three subroutines that handle initialization, running, and finalization, shown in Fig. 2(a). These subroutines are used by the ESMF/NUOPC coupler that controls the oceanic component in the coupled run. The surface boundary fields on the ocean surface are exchanged online[1] via the MITgcm–ESMF interface during the simulation. The MITgcm ocean surface variables are the export boundary fields, and the atmospheric surface variables are the import boundary fields (see Fig. 2(b)). These boundary fields are registered in the coupler following NUOPC protocols and timestamps[2] are added to them for the coupling. In addition, MITgcm grid information is also provided to the coupler in the initialization subroutine for online re-gridding of the exchanged boundary fields. To carry out the coupled simulation on HPC clusters, the MITgcm–ESMF interface runs in parallel via MPI communications. The implementation of the present MITgcm–ESMF interface is based on the baseline MITgcm–ESMF interface (Hill, 2005), but updated for compatibility with the modern version of ESMF/NUOPC. We also modify the baseline interface to receive atmosphere surface fluxes and send ocean surface variables.

## 2.3 WRF Atmospheric Model

The Weather Research and Forecasting (WRF) Model (Skamarock et al., 2005) is developed by NCAR/MMM (Mesoscale and Microscale Meteorology Division). It is a 3-D, finite-difference atmospheric model with a variety of physical parameterizations of sub-grid scale processes for predicting a broad spectrum of applications. WRF is used extensively for operational forecasts (http://www.wrf-model.org/plots/wrfrealtime.php) as well as realistic and idealized dynamical studies. The WRF code and documentation are under continuous development on Github (https://github.com/wrf-model/WRF).

In the present work, the Advanced Research WRF dynamic version (WRF-ARW, version 3.9.1.1, https://github.com/NCAR/WRFV3/releases/tag/V3.9.1.1) is used. It solves the compressible Euler non-hydrostatic equations, and also includes a run-time

---

[1]In this manuscript, 'online' means the manipulations are performed via subroutine calls during the execution of the simulations; 'offline' means the manipulations are performed when the simulations are not executing.

[2]In ESMF, 'timestamp' is a sequence of numbers, usually based on the time, to identify ESMF fields. Only the ESMF fields having the correct timestamp will be transferred in the coupling.

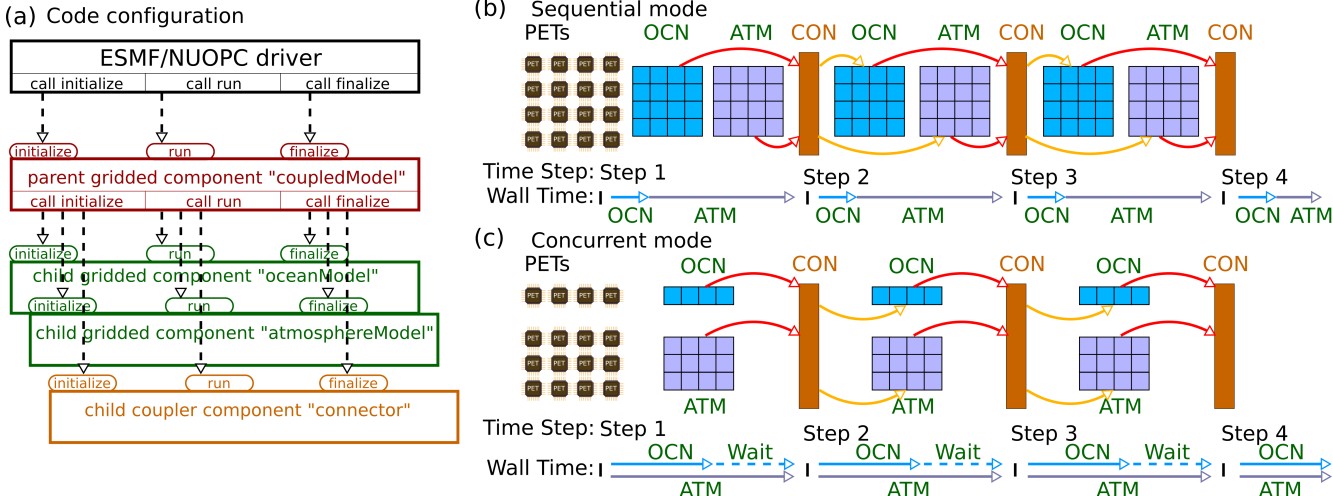

**Figure 2.** The general code structure and run sequence of the coupled ocean–atmosphere model. In panel (a), the black block is the *application driver*; the red block is the *parent gridded component* called by the *application driver*; the green/brown blocks are the *child gridded/coupler components* called by the *parent gridded component*. Panel (b) and (c) shows the sequential and concurrent mode implemented in SKRIPS, respectively. PETs (Persistent Execution Threads) are single processing units (e.g., CPU or GPU cores) defined by ESMF. OCN, ATM, and CON denote oceanic component, atmospheric component and connector component, respectively. The blocks under PETs are the CPU cores in the simulation; the small blocks under OCN or ATM are the small sub-domains in each core; the block under CON is the coupler. The red arrows indicate the model components are sending data to the connector and the yellow arrows indicate the model components are reading data from the connector. The horizontal arrows indicate the time axis of each component and the ticks on the time axis indicate the coupling time step.

hydrostatic option. The WRF-ARW uses a terrain-following hydrostatic pressure coordinate system in the vertical direction and utilizes the 'Arakawa C-grid'. WRF incorporates various physical processes including microphysics, cumulus parameterization, planetary boundary layer, surface layer, land surface, and longwave/shortwave radiations, with several options available for each process.

5    Similar to the implementation in MITgcm, WRF is also separated into initialization, run, and finalization subroutines to enable the WRF–ESMF interface to control the atmosphere model during the coupled simulation, shown in Fig. 2(a). The implementation of the present WRF–ESMF interface is based on the prototype interface (Henderson and Michalakes, 2005). In the present work, the prototype WRF–ESMF interface is updated to modern versions of WRF-ARW and ESMF, based on the NUOPC layer. This prototype interface is also expanded to interact with the ESMF/NUOPC coupler to receive the ocean

10   surface variables and send the atmosphere surface fluxes. The surface boundary condition fields are registered in the coupler following the NUOPC protocols with timestamps. The WRF grid information is also provided for online re-gridding by ESMF. To carry out the coupled simulation on HPC clusters, the WRF–ESMF interface also runs in parallel via MPI communications.

## 2.4 ESMF/NUOPC Coupler

The coupler is implemented using ESMF version 7.0.0. The ESMF is selected because of its high-performance and flexibility for building and coupling weather, climate, and related Earth science applications (Collins et al., 2005; Turuncoglu et al., 2013; Chen and Curcic, 2016; Turuncoglu and Sannino, 2017). It has a superstructure for representing the model and coupler components and an infrastructure of commonly used utilities, including conservative grid remapping, time management, error handling, and data communications.

The general code structure of the coupler is shown in Fig. 2. To build the ESMF/NUOPC driver, a main program is implemented to control an ESMF parent component, which controls the child components. In the present work, three child components are implemented: (1) the oceanic component; (2) the atmospheric component; and (3) the ESMF coupler. The coupler is used here because it performs the two-way interpolation and data transfer (Hill et al., 2004). In ESMF, the model components can be run in parallel as a group of Persistent Execution Threads (PETs), which are single processing units (e.g., CPU or GPU cores) defined by ESMF. In the present work, the PETs are created according to the grid decomposition, and each PET is associated with an MPI process.

The ESMF allows the PETs to run in sequential mode, concurrent mode, or mixed mode (for more than three components). We implemented both sequential and concurrent modes in SKRIPS, shown in Fig. 2(b) and 2(c). In sequential mode, a set of ESMF gridded/coupler components are run in sequence on the same set of PETs. At each coupling time step, the oceanic component is executed when the atmospheric component is completed or vice versa. On the other hand, in concurrent mode, the gridded components are created and run on mutually exclusive sets of PETs. If one component finishes earlier than the other, its PETs are idle and have to wait for the other component, shown in Fig. 2(c). However the PETs can be optimally distributed by the users to best achieve load balance.

In ESMF, the gridded components are used to represent models, and coupler components are used to connect these models. The interfaces and data structures in ESMF have few constraints, providing the flexibility to be adapted to many modeling systems. However, the flexibility of the gridded components can limit the interoperability across different modeling systems. To address this issue, the NUOPC layer is developed to provide the coupling conventions and the generic representation of the model components (e.g. drivers, models, connectors, mediators). The NUOPC layer in the present coupled model is implemented according to consortium documentations (Hill et al., 2004; Theurich et al., 2016), and the oceanic/atmospheric component each has:

1. Prescribed variables for NUOPC to link the components;

2. The entry point for registration of the components;

3. An *InitializePhaseMap* which describes a sequence of standard initialization phases, including documenting the fields that a component can provide, checking and mapping the fields to each other, and initializing the fields that will be used;

4. A *RunPhaseMap* that checks the incoming clock of the driver, examines the timestamps of incoming fields, and runs the component;

5. Timestamps on exported fields consistent with the internal clock of the component;

6. The *finalization method* to clean up all allocations.

The subroutines that handle initialization, running, and finalization in MITgcm and WRF are included in the *InitializePhaseMap*, *RunPhaseMap*, and *finalization method* in the NUOPC layer, respectively.

## 3 Experiment Design and Observational Datasets

We simulate a series of heat events in the Red Sea region, with a focus on validating and assessing the technical aspects of the coupled model. There is a desire for improved and extended forecasts in this region, and future work will investigate whether a coupled framework can advance this goal. The extreme heat events are chosen as a test case due to their societal importance. While these events and the analysis here do not highlight the value of coupled forecasting, these real-world events are adequate to demonstrate the performance and physical realism of the coupled model code implementation. The simulation of the Red Sea extends from 0000 UTC 01 June 2012 to 0000 UTC 01 July 2012. We select this month because of the record-high surface air temperature observed in the Makkah region, located 70 km inland from the eastern shore of the Red Sea (Abdou, 2014).

The computational domain and bathymetry are shown in Fig. 3. The model domain is centered at 20° N and 40° E, and the bathymetry is from the 2-minute Gridded Global Relief Data (ETOPO2) (National Geophysical Data Center, 2006). WRF is implemented using a horizontal grid of $256 \times 256$ points and grid spacing of 0.08°. The cylindrical equidistant map (latitude-longitude) projection is used. There are 40 terrain-following vertical levels, more closely spaced in the atmospheric boundary layer. The time step for atmosphere simulation is 30 seconds, which is to avoid violation of the CFL condition. The Morrison 2-moment scheme (Morrison et al., 2009) is used to resolve the microphysics. The updated version of the Kain–Fritsch convection scheme (Kain, 2004) is used with the modifications to include the updraft formulation, downdraft formulation, and closure assumption. The Yonsei University (YSU) scheme (Hong et al., 2006) is used for the planetary boundary layer (PBL), and the Rapid Radiation Transfer Model for GCMs (RRTMG; Iacono et al. (2008)) is used for longwave and shortwave radiation transfer through the atmosphere. The Rapid Update Cycle (RUC) land surface model is used for the land surface processes (Benjamin et al., 2004). The MITgcm uses the same horizontal grid spacing as WRF, with 40 vertical z-levels that are more closely spaced near the surface. The time step of the ocean model is 120 seconds. The horizontal sub-grid mixing is parameterized using nonlinear Smagorinsky viscosities, and the K-profile parameterization (KPP) (Large et al., 1994) is used for vertical mixing processes.

In the coupling process, the ocean model sends SST and ocean surface velocity to the coupler, and they are used directly as the boundary conditions in the atmosphere model. The atmosphere model sends the surface fields to the coupler, including (1) net surface shortwave/longwave radiation, (2) surface latent/sensible heat flux, (3) 10-m wind speed, (4) precipitation, (5) evaporation. The ocean model uses the atmosphere surface fields to compute the surface forcing, including (1) total net surface heat flux, (2) surface wind stress, (3) freshwater flux. The total net surface heat flux is computed by adding latent heat flux, sensible heat flux, and net surface shortwave/longwave radiation fluxes. The surface wind stress is computed by using the 10-m

wind speed (Large and Yeager, 2004). The freshwater flux is the difference between precipitation and evaporation. The latent and sensible heat fluxes are computed by using COARE 3.0 bulk algorithm in WRF (Fairall et al., 2003). In the coupled code, different bulk formulae in WRF or MITgcm can also be used.

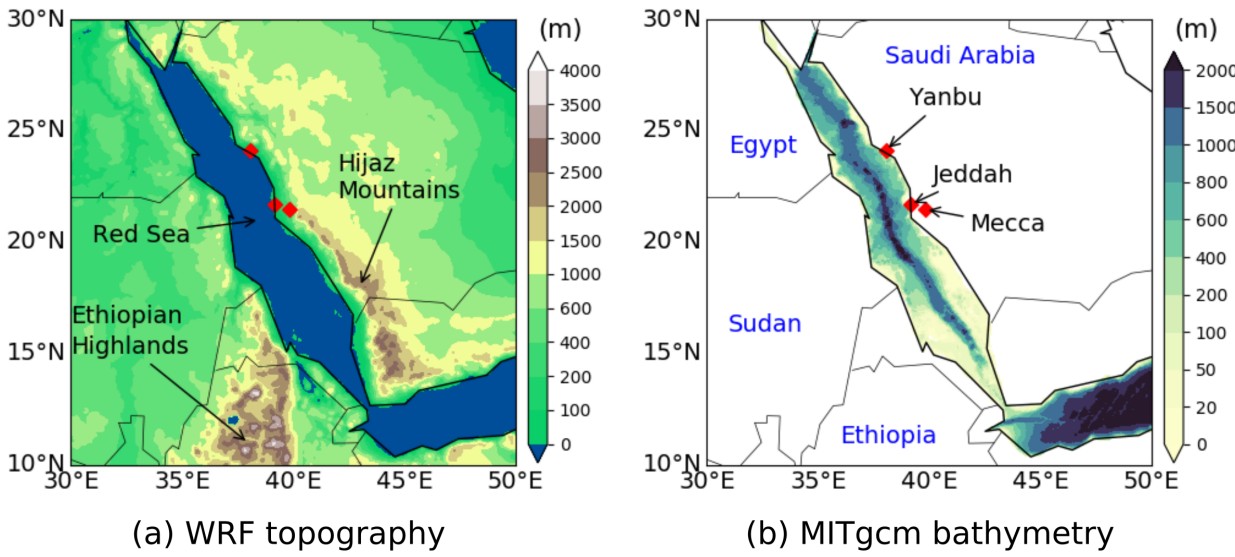

(a) WRF topography       (b) MITgcm bathymetry

**Figure 3.** The WRF topography and MITgcm bathymetry in the simulations. Three major cities near the eastern shore of the Red Sea are highlighted. The Hijaz Mountains and Ethiopian Highlands are also highlighted.

To validate the coupled model, the following sets of simulations using different surface forcings are performed according to the tests (Warner et al., 2010; Turuncoglu et al., 2013; Ricchi et al., 2016):

1. Run CPL: a two-way coupled MITgcm–WRF simulation. The coupling interval is 20 minutes to capture the diurnal cycle (Seo et al., 2014). This run tests the implementation of the two-way coupled ocean–atmosphere model.

2. Run ATM.STA: a stand-alone WRF simulation with its initial SST kept constant throughout the simulation. This run allows assessment of the WRF model behavior with realistic, but persistent SST. This case serves as a benchmark to highlight the difference between coupled and uncoupled runs, and also to demonstrate the impact of evolving SST.

3. Run ATM.DYN: a stand-alone WRF simulation with a varying, prescribed SST based on HYCOM/NCODA reanalysis data. This allows assessing the WRF model behavior with updated SST and is used to validate the coupled model. It is noted that in practice an accurately evolving SST would not be available for forecasting.

4. Run OCN.DYN: a stand-alone MITgcm simulation forced by the ERA5 reanalysis data. The bulk formula in MITgcm is used to derive the turbulent heat fluxes. This run assesses the MITgcm model behavior with prescribed lower-resolution atmospheric surface forcing, and is also used to validate the coupled model.

The ocean model uses the data assimilating HYCOM/NCODA 1/12° global reanalysis data as initial and boundary conditions for ocean temperature, salinity, and horizontal velocities (https://www.hycom.org/dataserver/gofs-3pt1/reanalysis). The boundary conditions for the ocean are updated on a 3-hourly basis and linearly interpolated between two simulation time steps. A sponge layer is applied at the lateral boundaries, with a thickness of 3 grid cells. The inner and outer boundary relaxation timescales of the sponge layer are 10 and 0.5 days, respectively. In CPL, ATM.STA, and ATM.DYN runs, we use the same initial condition and lateral boundary condition for the atmosphere. The atmosphere is initialized using the ECMWF ERA5 reanalysis data, which has a grid resolution of approximately 30 km (Hersbach, 2016). The same data also provide the boundary conditions for air temperature, wind speed, and air humidity every 3 hours. The atmosphere boundary conditions are also linearly interpolated between two simulation time steps. The lateral boundary values are specified in WRF in the 'specified' zone, and the 'relaxation' zone is used to nudge the solution from the domain toward the boundary condition value. Here we used the default width of one point for the specific zone and four points for the relaxation zone. The pressure at the top of the atmosphere is 50 hPa. In ATM.STA run, the SST from HYCOM/NCODA is used as initial and persistent SST. The time-varying SST in ATM.DYN run is also generated using HYCOM/NCODA data. We select HYCOM/NCODA data because the ocean model initial condition and boundary conditions are generated using it. For the OCN.DYN run we select ERA5 data for prescribed atmospheric state because it also provides the atmospheric initial and boundary conditions in the CPL run. The initial condition, boundary condition, and forcing terms of all simulations are summarized in Table 1.

**Table 1.** The initial condition, boundary condition and forcing terms used in present simulations.

| | initial and boundary conditions | ocean surface conditions | atmospheric forcings |
|---|---|---|---|
| CPL | ERA5 (atmosphere) HYCOM/NCODA (ocean) | from MITgcm | from WRF |
| ATM.STA | ERA5 | HYCOM/NCODA initial condition kept constant | N.A. |
| ATM.DYN | ERA5 | HYCOM/NCODA updated every 3 hours | N.A. |
| OCN.DYN | HYCOM/NCODA | N.A. | ERA5 |

The validation of the coupled model focuses on temperature, heat flux, and surface wind. Our aim is to validate the coupled model and show that the heat and momentum fluxes simulated by the coupled model are comparable to the observations or the reanalysis data. The simulated SST data are validated against the OSTIA (Operational Sea Surface Temperature and Sea Ice Analysis) system in GHRSST (Group for High Resolution Sea Surface Temperature) (Donlon et al., 2012; Martin et al., 2012). The simulated SST is also validated against HYCOM/NCODA data to show the increase of the error. The simulated 2-meter air temperature (T2) fields are validated using ERA5. In addition, the T2 in three major cities near the eastern shore of the Red Sea are validated using ERA5 and the ground observations from NOAA National Climate Data Center (NCDC

climate data online at http://cdo.ncdc.noaa.gov/CDO/georegion). For this comparison the T2 fields from both the simulations and ERA5 are interpolated to the NCDC stations. When interpolating ERA5 data to the NCDC stations near the coast, only the data saved on ERA5 land points (land-sea mask>90%) are used in the bi-linear interpolation. The high/low temperature every 24 hours from the simulations and ERA5 are compared to the daily maximum/minimum temperatures with NCDC data.

Surface heat fluxes (e.g., latent heat, sensible heat, longwave and shortwave radiations), which drives the oceanic component in the coupled model, are validated using MERRA-2 (Modern-Era Retrospective analysis for Research and Applications, version 2) data (Gelaro et al., 2017). The MERRA-2 data are selected because (1) it is an independent reanalysis data compared to the initial and boundary conditions used in the simulations, and (2) it also provides a $0.625^o \times 0.5^o$ (lon $\times$ lat) resolution reanalysis fields of turbulent heat fluxes (THF). The 10-m wind speed is also compared with MERRA-2 data to validate the momentum

flux in the coupled code. To compare with validation data, we interpolated the validation data on the lower resolution grid to the higher resolution grid of the regional model. The validation data are summarized in Table 2. The validation of the freshwater flux is shown in the Appendix because (1) the evaporation is proportional to the latent heat in the model and (2) the precipitation is zero in three major cities near the coast in Fig. 3.

**Table 2.** The observational data and reanalysis data used to validate the simulation results.

| variable | validation data |
| --- | --- |
| sea surface temperature (SST) | GHRSST and HYCOM/NCODA |
| 2-meter air temperature (T2) | ERA5 and NCDC climate data |
| turbulent heat fluxes | MERRA-2 |
| radiative fluxes | MERRA-2 |
| 10-meter wind | MERRA-2 |

## 4   Results and Discussions

The Red Sea is an elongated basin covering the area between 12-30°N and 32-43°E. The basin is 2250 km long, extending from the Suez and Aqaba gulfs in the north to the strait of Bal el-Mandeb in the south, which connects the Red Sea and the Indian Ocean. In this section, the simulation results obtained by using different model configurations are presented to show that SKRIPS is capable of performing coupled ocean–atmosphere simulations.

### 4.1   2-meter Air Temperature (T2)

We begin our analysis by examining the simulated T2 from the model experiments. Since the record-high temperature is observed in the Makkah region on June 2[nd], the simulation results on June 2[nd] (36 or 48 hours after the initialization) are shown in Fig. 4. The ERA5 data, and the difference between CPL run and ERA5 are also shown in Fig. 4. It can be seen in Fig. 4(I) that the CPL run captures the T2 patterns in the Red Sea region on June 2[nd], compared with ERA5 in Fig. 4(II). Since ERA5 air

temperature data are in good agreement with the NCDC ground observation data in the Red Sea region (detailed comparison of all stations are not shown), we use ERA5 data to validate the simulation results. The difference between the CPL run and ERA5 is shown in Fig. 4(III). The ATM.STA and ATM.DYN simulation results are close to the CPL run results and thus are not shown, but their differences with respect to ERA5 are shown in Fig. 4(IV) and 4(V), respectively. Fig. 4(VI) to 4(X) show the nighttime results after 48 hours. It can be seen in Fig. 4 that all simulations reproduce the T2 patterns over the Red Sea region reasonably well compared with ERA5. The mean T2 biases and RMSEs over the sea are shown in Table 3. The biases of the T2 are comparable with the biases reported in other WRF simulations for heat events (Imran et al., 2018). Fig. 4 also shows the diurnal variation of T2 in the Red Sea region, and the diurnal variation will be further discussed later in this section.

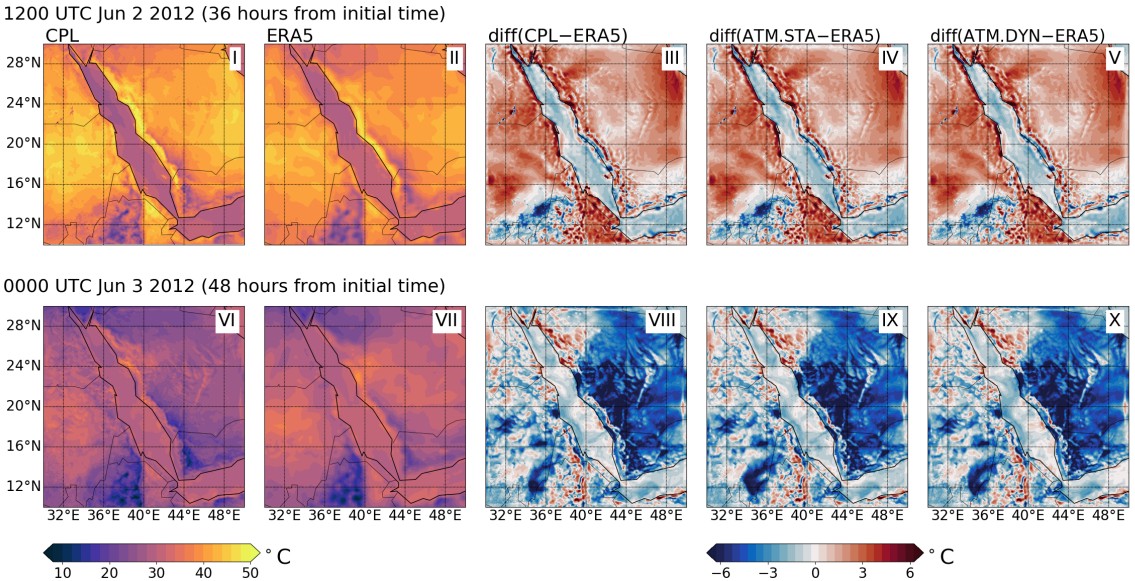

**Figure 4.** The 2-m air temperature as obtained from the CPL run, the ERA5 data, and their difference (CPL−ERA5). The differences between ATM.STA and ATM.DYN with ERA5 (i.e., ATM.STA−ERA5, ATM.DYN−ERA5) are also presented. The simulation initial time is 0000 UTC Jun 01 2012 for both snapshots. Two snapshots are selected: (1) 1200 UTC Jun 02 2012 (36 hours from initial time); (2) 0000 UTC Jun 03 2012 (48 hours from initial time). The results on Jun 02 are presented because the record-high temperature is observed in the Makkah region.

The simulation results on June 10[th] and 24[th] are shown in Fig. 5 to validate the coupled model over longer periods of time. In Fig. 5, we aim to show the difference over the sea to validate the coupled ocean–atmosphere model. It can be seen in Fig. 5(III) and 5(VIII) that the T2 patterns simulated by the coupled run are generally consistent with ERA5. The differences between ATM.STA and ATM.DYN simulation results with ERA5 are shown in Fig. 5(IV), 5(V), 5(IX), and 5(X), respectively. It can be seen in the figure that the T2 errors on land are consistent for all three simulations. However, the T2 over the sea in CPL simulation has smaller mean biases and RMSEs compared with the ATM.STA run, shown in Table 3. Although the difference of the biases is still very small compared with the mean T2 (31.92 °C), the improvement of the coupled run on the 24[th] (1.02 °C)

is comparable to the standard deviation of T2 (1.64 °C). The T2 over the water in the CPL run is closer to ERA5 because MITgcm in the coupled model provides updated warming SST, which warms the T2; the ATM.STA run uses a constant cooler SST from June 1st, and thus the T2 is determined by the constant cooler SST. On the other hand, when comparing the CPL run with the ATM.DYN run, the mean difference is smaller (10th: +0.04 °C; 24th: -0.62 °C). This shows the CPL run is comparable to the ATM.DYN run which is driven by an updated warming SST.

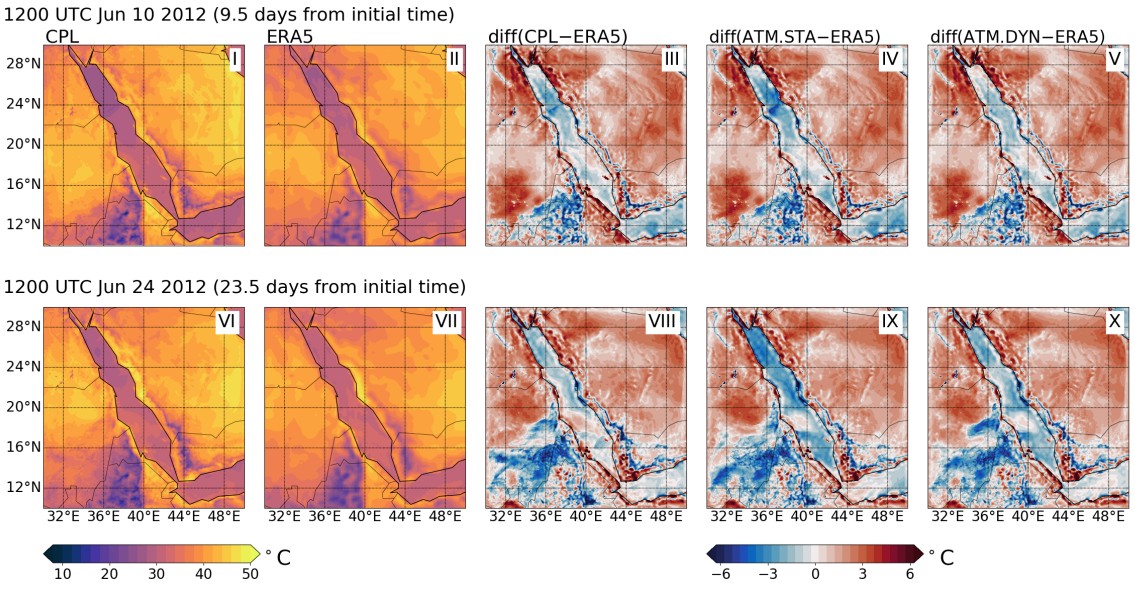

**Figure 5.** The 2-m air temperature as obtained from the CPL run, the ERA5 data, and their difference (CPL−ERA5). The difference between ATM.STA and ATM.DYN with ERA5 data (i.e., ATM.STA−ERA5, ATM.DYN−ERA5) are also presented. The simulation initial time is 0000 UTC Jun 01 2012 for both snapshots. Two snapshots are selected: (1) 1200 UTC Jun 10 2012 (9.5 days from initial time); (2) 1200 UTC Jun 24 2012 (23.5 days from initial time).

**Table 3.** The biases and RMSEs of the T2 simulated in all simulations in comparison with ERA5 data.

|  | after 36 hours | after 48 hours | after 9.5 days | after 23.5 days |
|---|---|---|---|---|
| CPL run | bias: -1.36; RMSE: 1.20 | bias: -0.82; RMSE: 1.18 | bias: -1.24; RMSE: 1.74 | bias: -0.81; RMSE: 1.59 |
| ATM.STA run | bias: -1.48; RMSE: 1.23 | bias: -0.92; RMSE: 1.21 | bias: -1.56; RMSE: 1.91 | bias: -1.83; RMSE: 1.83 |
| ATM.DYN run | bias: -1.36; RMSE: 1.21 | bias: -0.84; RMSE: 1.18 | bias: -1.20; RMSE: 1.46 | bias: -1.43; RMSE: 1.37 |

To validate the diurnal T2 variation of the coupled model in Fig. 4, the time series of T2 in three major cities as simulated in CPL and ATM.STA runs are plotted in Fig. 6, starting from June 1st. The ATM.DYN run results are similar to the CPL run results and thus are not shown. To validate the simulation results, the time series in ERA5 data and the daily observed high/low

temperature data from NOAA NCDC are also plotted. It can be seen both coupled and uncoupled simulations generally captured the four major heat events (i.e., June 2nd, 10th, 17th, and 24th) and the T2 variations during the 30-day simulation. For the daily high T2, the root mean square error (RMSE) in the CPL run (2.09 °C) is close to the ATM.STA run (2.16 °C), and the error does not increase in the 30-day simulation. For the daily low T2, before June 20th (lead time < 19 days), the CPL and ATM.STA runs have consistent RMSEs compared with ground observation (CPL: 4.23 °C; ATM.STA: 4.39 °C). In Jeddah and Yanbu, the CPL run has better captured the daily low T2 after June 20th in CPL run (Jeddah: 3.95 °C; Yanbu: 3.77 °C) than ATM.STA run (Jeddah: 4.98 °C; Yanbu: 4.29 °C) by about 1 °C and 0.5 °C, respectively. However, the difference of T2 in Mecca, which is located 70-km from the sea, is negligible (0.05 °C) between CPL and ATM.STA runs throughout the simulation. It should be mentioned that both the present simulations and ERA5 reported a T2 that is 2.8 °C cooler than the observed record-high T2 in Mecca on June 2nd. This under-estimation is comparable with the RMSE of the daily high T2 in Mecca (2.25 °C in CPL run).

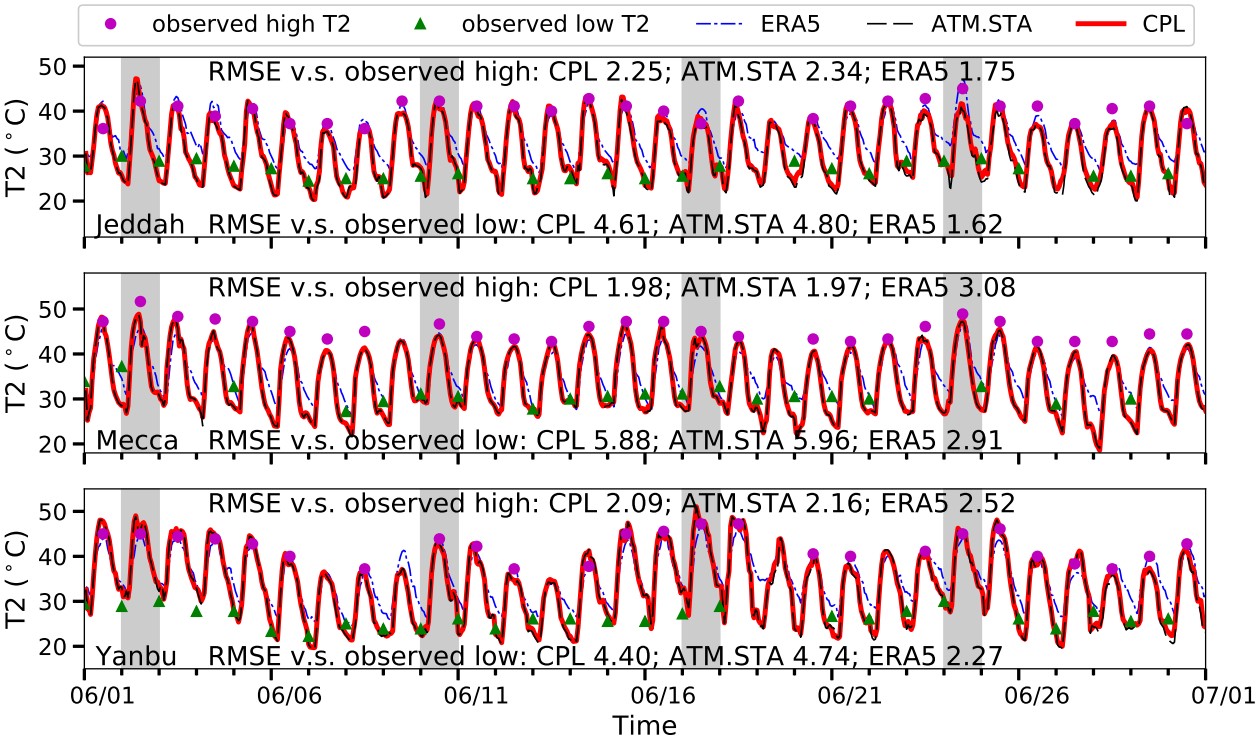

**Figure 6.** Temporal variation the 2-m air temperature at three major cities near the eastern shore of Red Sea (Jeddah, Mecca, Yanbu) as resulting from CPL and ATM.STA runs. The temperature data are compared with the time series in ERA5 and daily high/low temperature in the NOAA national data center dataset. Note that some 2-m air temperature data gaps exist in the NCDC ground observation dataset. Four representative heat events are highlighted in this figure.

The simulation error of T2 also oscillates diurnally in the present simulations. To demonstrate the diurnal variation of the simulation error quantitatively, the mean biases and RMSEs of T2 between the simulations (i.e., ATM.STA, ATM.DYN, and

CPL) and ERA5 data are shown in Fig. 7. To validate the coupled ocean–atmosphere model, only the temperature over the Red Sea is compared. It can be seen in Fig. 7 that the ATM.STA run using the static SST can still capture the T2 patterns in the first week, but it under-predicts T2 by about 2 °C after 20 days because of ignoring the SST evolution. On the other hand, CPL run has smaller bias (-0.60 °C) and RMSE (1.28 °C) compared with those in ATM.STA run (bias: -1.19 °C; RMSE: 1.71 °C) during the 30-day simulation as the SST evolution is considered. The ATM.DYN run also has smaller error than ATM.STA and its error is consistent with that in CPL run (bias: -0.72 °C; RMSE: 1.31 °C), indicating that the coupled simulation is comparable to the stand-alone atmosphere simulation driven by 3-hourly reanalysis SST. The biases and RMSEs of T2 in the present work are similar to those in the benchmark WRF-ARW simulations (Xu et al., 2009; Zhang et al., 2013a; Imran et al., 2018). The differences of the mean biases and RMSEs between the simulations and ERA5 data are also plotted to demonstrate the evolution of the CPL errors compared with ATM.STA and ATM.DYN runs. It can be seen that the CPL run has smaller bias and RMSE than the ATM.STA run throughout the entire simulation. The bias and RMSE between CPL run and ATM.DYN runs are within about 0.5 °C. This demonstrates the capability of the coupled model for performing realistic regional coupled ocean–atmosphere simulations.

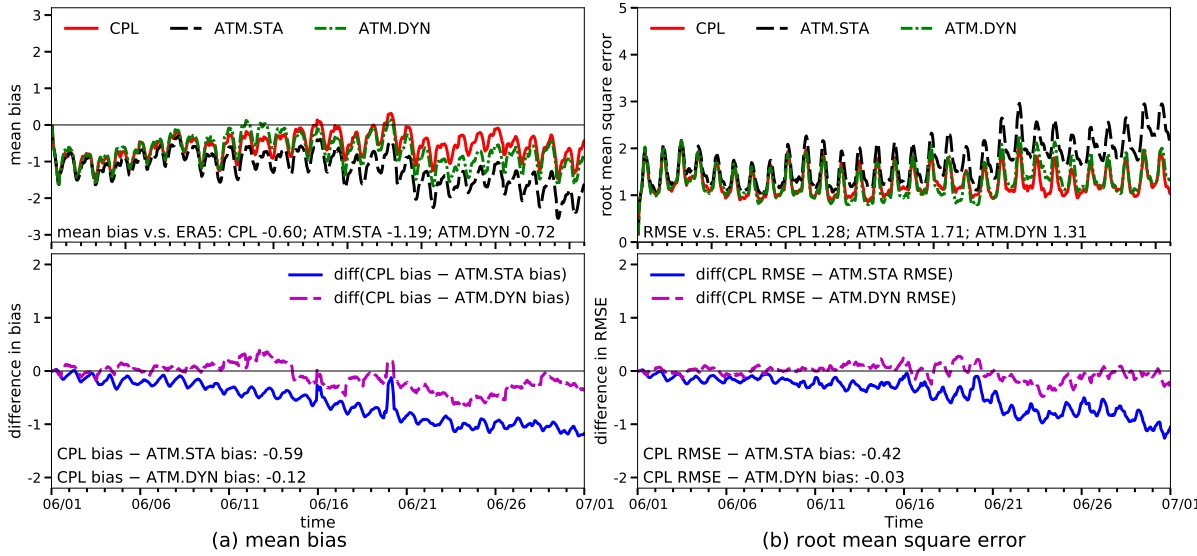

**Figure 7.** The bias and root mean square error (RMSE) between the 2-m air temperature obtained by the simulations (i.e., ATM.STA, ATM.CPL, and CPL) in comparison with ERA5 data. Only the errors over the Red Sea are considered. The differences between the simulation errors from CPL run and stand-alone WRF simulations are presented below the mean bias and the RMSE. The initial time is 0000 UTC Jun 01 2012 for all simulations.

## 4.2 Sea Surface Temperature

The simulated SST patterns are compared to the validation data to demonstrate that the coupled model can capture the ocean surface state. The SST field snapshots from CPL run on June 2$^{nd}$ and 24$^{th}$ are shown in Fig. 8(I) and Fig. 8(VI). To validate the CPL run results, the SST fields obtained in OCN.DYN runs are shown in Fig. 8(II) and 8(VII), and the GHRSST fields are shown in Fig. 8(III) and 8(VIII). The SST obtained in the model at 0000 UTC (about 3 A.M. local time in the Red Sea region) is presented because the GHRSST is produced with nighttime SST data (Roberts-Jones et al., 2012). It can be seen that both OCN.DYN and CPL runs are able to reproduce the SST patterns reasonably well in comparison with GHRSST for both snapshots. Though the CPL run uses the surface forcing fields with a higher resolution, the SST patterns obtained in both simulations are very similar after two days. On June 24$^{th}$, the SST patterns in both runs are less similar, but both simulation results are still comparable with GHRSST (RMSE $< 1°$C). Both simulations under-estimate the SST in the northern Red Sea and slightly over-estimates the SST in the central and southern Red Sea on June 24$^{th}$.

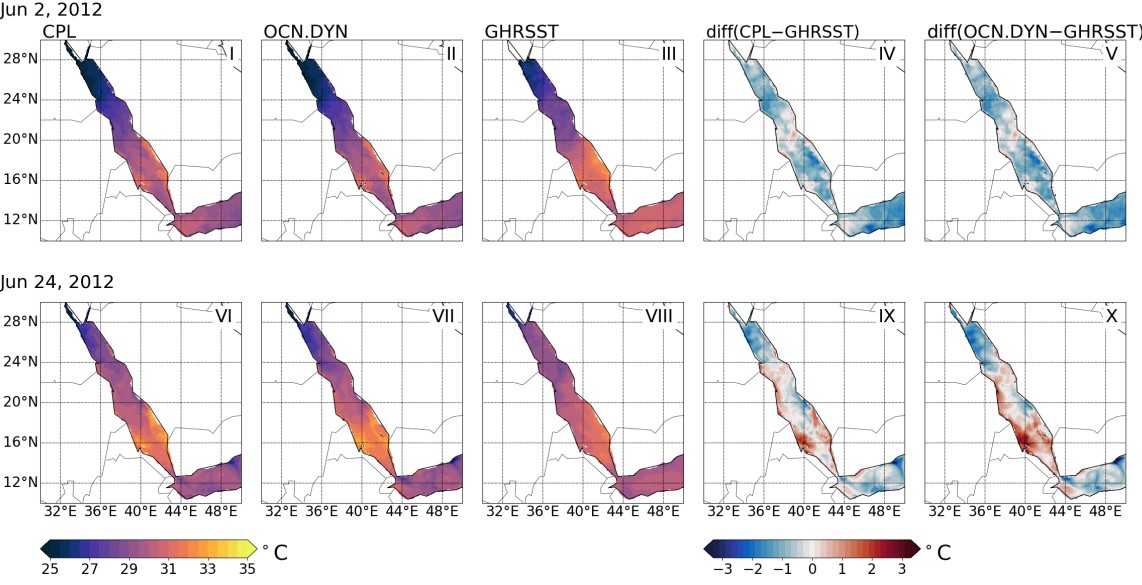

**Figure 8.** The daily SST patterns obtained by OCN.DYN and CPL runs, and GHRSST data. The corresponding differences between the simulations and the GHRSST are also plotted. Two snapshots are selected: (1) 0000 UTC Jun 02 2012; (2) 0000 UTC Jun 24 2012. The simulation initial time is 0000 UTC Jun 01 2012 for both snapshots.

To quantitatively compare the errors in SST, the time history of the SST in the simulations (i.e., OCN.DYN and CPL) and validation data (i.e., GHRSST and HYCOM/NCODA) are shown in Fig. 9. The mean bias and RMSE between simulation results and validation data are also plotted. In Fig. 9(a) the snapshots of the simulated SST are compared with available HYCOM/NCODA data, in Fig. 9(b) the snapshots of SST at 0000 UTC (about 3 A.M. local time in the Red Sea region) are compared with GHRSST. Generally, the OCN.DYN and CPL runs have a similar range of error compared to both validation

datasets in the 30-day simulations. The simulation results are compared with HYCOM/NCODA data to show the increase of RMSE in Fig. 9(a). Compared with HYCOM/NCODA, the mean differences between CPL and OCN.DYN runs are small (CPL: 0.10 °C; OCN.DYN: 0.03 °C). The RMSE increases in the first week, but does not grow after it. On the other hand, when comparing with the GHRSST, the initial SST patterns in both runs are cooler by about 0.8 °C. This is because our models are
initialized by using HYCOM/NCODA, and the temperature in the topmost model level is cooler than the estimated foundation SST reported by GHRSST. After the first 10 days, the difference between GHRSST data and HYCOM/NCODA decreases, and likewise the difference between the simulation results and GHRSST also decreases. It should be noted that the SST simulated by the CPL run has smaller error (bias: -0.57 °C; RMSE: 0.69 °C) compared with OCN.DYN (bias: -0.66 °C; RMSE: 0.76 °C) by about 0.1 °C when validated using GHRSST. This indicates the coupled model can adequately simulate the SST evolution
compared with the uncoupled model forced by ERA5 reanalysis data.

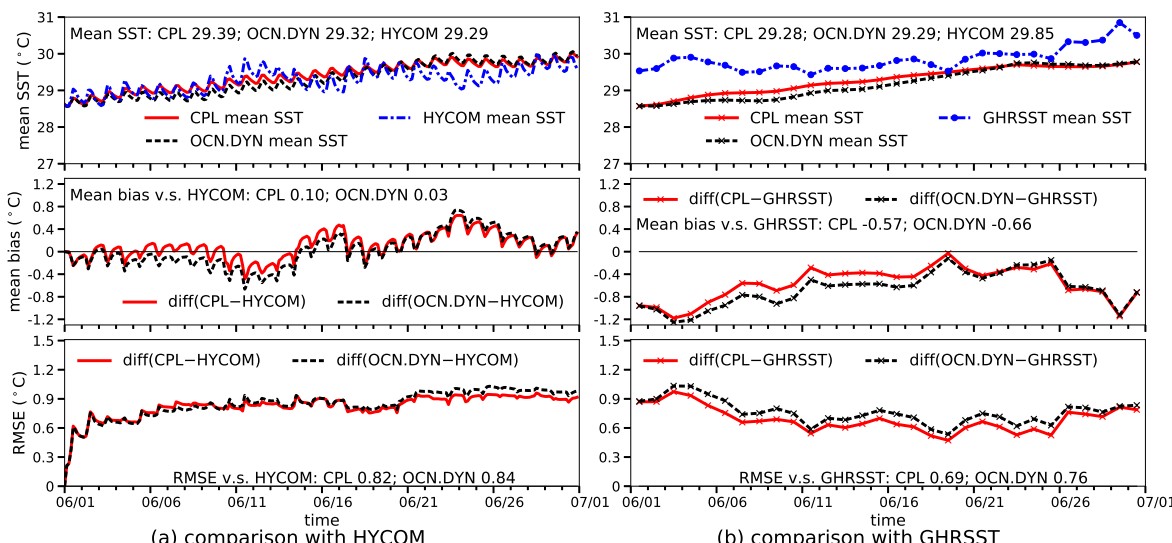

**Figure 9.** The bias and RMSE between the SST from the simulations (i.e., OCN.DYN and CPL) in comparison with the validation data. Panel (a) shows the comparison with HYCOM/NCODA data and Panel (b) shows the comparison with GHRSST. The initial time is 0000 UTC Jun 01 2012 for all simulations.

## 4.3   Surface Heat Fluxes

The atmosphere surface heat flux drives the oceanic component in the coupled model, hence we validate the heat fluxes in the coupled model as compared to the stand-alone simulations. Both the turbulent heat fluxes and the net downward heat fluxes are compared to MERRA-2 and their differences are plotted. To validate the coupled ocean–atmosphere model, we only compare
the heat fluxes over the sea.

The turbulent heat fluxes (THF; sum of latent and sensible heat fluxes) and their differences with the validation data are shown in Fig. 10. It can be seen in Fig. 10 that both CPL and ATM.STA runs capture the mean THF over the Red Sea compared with MERRA-2 (CPL: 119.4 W/m$^2$; ATM.STA: 103.4 W/m$^2$; MERRA-2: 115.6 W/m$^2$). For the first two weeks, the mean THFs obtained in CPL and ATM.STA in Fig. 10 are overlapping and all simulations exhibit similar THF patterns because

they are initialized in the same way and the SST fields are similar (see the snapshots comparison in the Appendix). After the second week, the CPL run has smaller error (bias: -1.8 W/m$^2$; RMSE: 69.9 W/m$^2$) compared with the ATM.STA run (bias: -25.7 W/m$^2$; RMSE: 76.4 W/m$^2$). This is because the SST is updated in the CPL run and is warmer compared with ATM.STA run. When forced by a warmer SST, the evaporation increases (also see the Appendix) and thus the latent heat fluxes increases. On the other hand, the THFs in the CPL run are comparable with the ATM.DYN run during the 30-day run (bias: 1.9 W/m$^2$),

showing the SKRIPS can capture the THFs over the Red Sea in the coupled simulation.

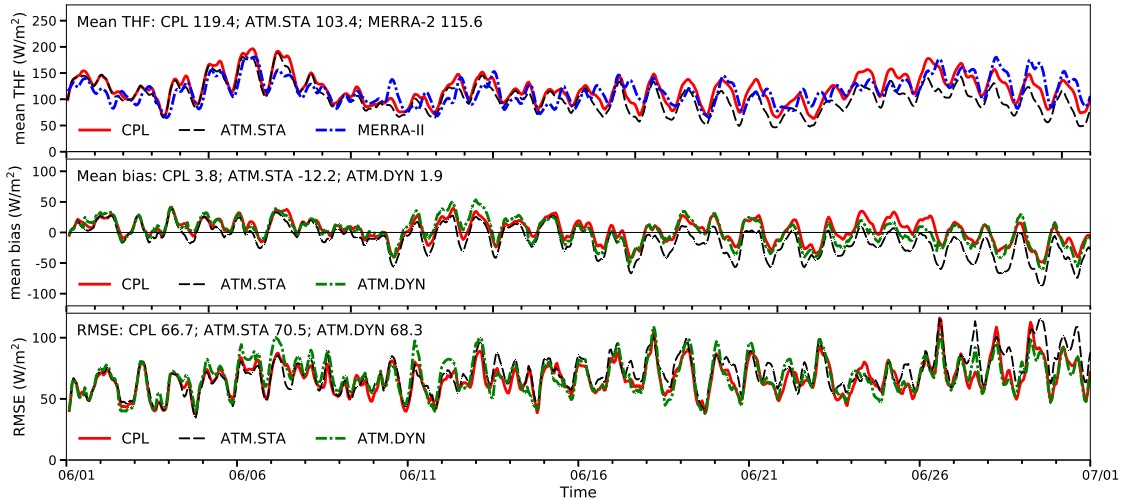

**Figure 10.** The turbulent heat fluxes out of the sea obtained in CPL and ATM.STA runs in comparison with MERRA-2. The top panel shows the mean THF; the middle panel shows the mean bias; the bottom panel shows the RMSE. Only the hourly heat fluxes over the sea are shown to validate the coupled model.

The net downward heat fluxes (sum of latent heat, sensible heat, shortwave radiation fluxes, and longwave radiation fluxes) are shown in Fig. 11. Again, for the first two weeks, the heat fluxes obtained in CPL and ATM.STA runs are overlapping and all simulations exhibit similar heat flux patterns because they are initialized in the same way the SST fields are similar (see the snapshots comparison in the Appendix). After the second week, the CPL run has slightly smaller error (bias: 11.2 W/m$^2$;

RMSE: 84.4 W/m$^2$) compared with the ATM.STA simulation (bias: 36.5 W/m$^2$; RMSE: 94.3 W/m$^2$). It should be noted that the mean bias and RMSE of the net downward heat fluxes can be as high as a few hundred W/m$^2$ or 40% compared with MERRA-2. This is because WRF over-estimated the shortwave radiations at daytime (detailed comparisons are shown in the Appendix). However, the coupled model still captures the mean and standard deviation of the heat flux compared with MERRA-2 data

(CPL mean: 110.6 W/m$^2$, standard deviation: 350.7 W/m$^2$; MERRA-2 mean 104.7 W/m$^2$, standard deviation 342.3 W/m$^2$). The over-estimation of shortwave radiation by RRTMG scheme is also reported in other validation tests in the literature under all-sky conditions due to the uncertainty of cloud or aerosol (Zempila et al., 2016; Imran et al., 2018). Although the surface heat flux is slightly over-estimated at daytime, the SST over the Red Sea is not over-estimated (shown in Section 4.2).

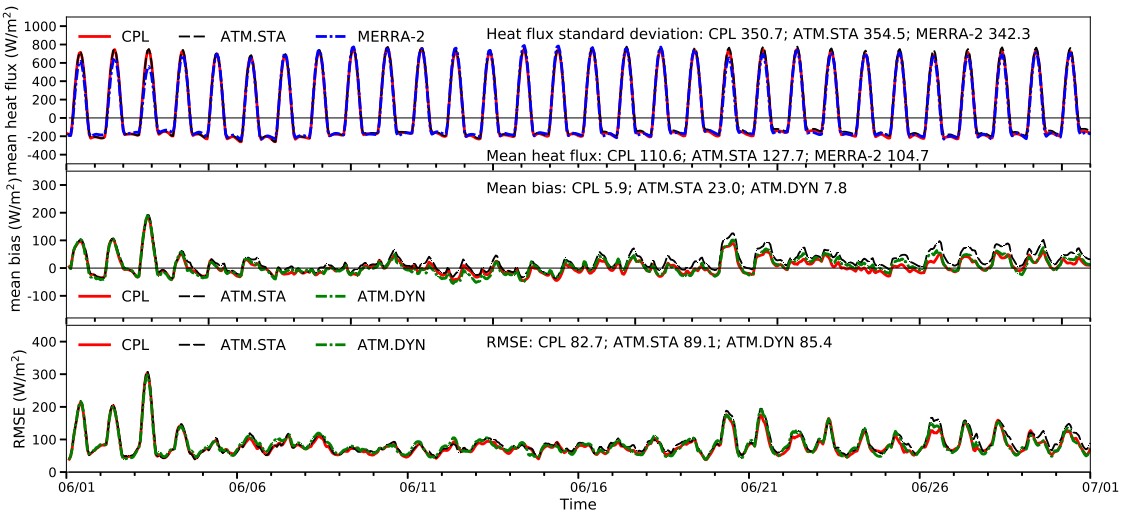

**Figure 11.** The total surface heat fluxes into the sea obtained in CPL and ATM.STA runs in comparison with MERRA-2. The top panel shows the mean surface heat flux; the middle panel shows the mean bias; the bottom panel shows the RMSE. Only the heat fluxes over the sea are shown to validate the coupled model.

## 4.4   10-m Wind

To evaluate the simulation of the surface momentum by the coupled model, the 10-m wind patterns obtained from ATM.STA, ATM.DYN, and CPL runs are presented. The MERRA-2 data are used to validate the simulation results.

The simulated 10-m wind velocity fields are shown in Fig. 12. The RMSE of the wind velocity magnitude between the CPL run and MERRA-2 data is 2.23 m/s when using the selected WRF physics schemes presented in Section 3. On June 2$^{\text{nd}}$, high-speed wind is observed in the northern and central Red Sea, and both the CPL and ATM.STA runs capture the features of wind speed patterns. On June 24$^{\text{th}}$, the high-speed wind is observed in the central Red Sea and is also captured by both CPL and ATM.STA runs. The mean 10-m wind speed over the Red Sea in the CPL and ATM.STA runs during the 30-day simulation are shown in Fig. 13. The mean error of CPL run (mean bias: -0.23 m/s; RMSE: 2.38 m/s) is slightly smaller than the ATM.STA run (mean bias: -0.34 m/s; RMSE: 2.43 m/s) by about 0.1 m/s. Although CPL, ATM.STA, ATM.DYN runs have different SST as the atmospheric boundary condition, the 10-m wind speed fields obtained in the simulations are all consistent with MERRA-2 data. The comparison shows the SKRIPS is capable of simulating the surface wind speed over the Red Sea in the coupled simulation.

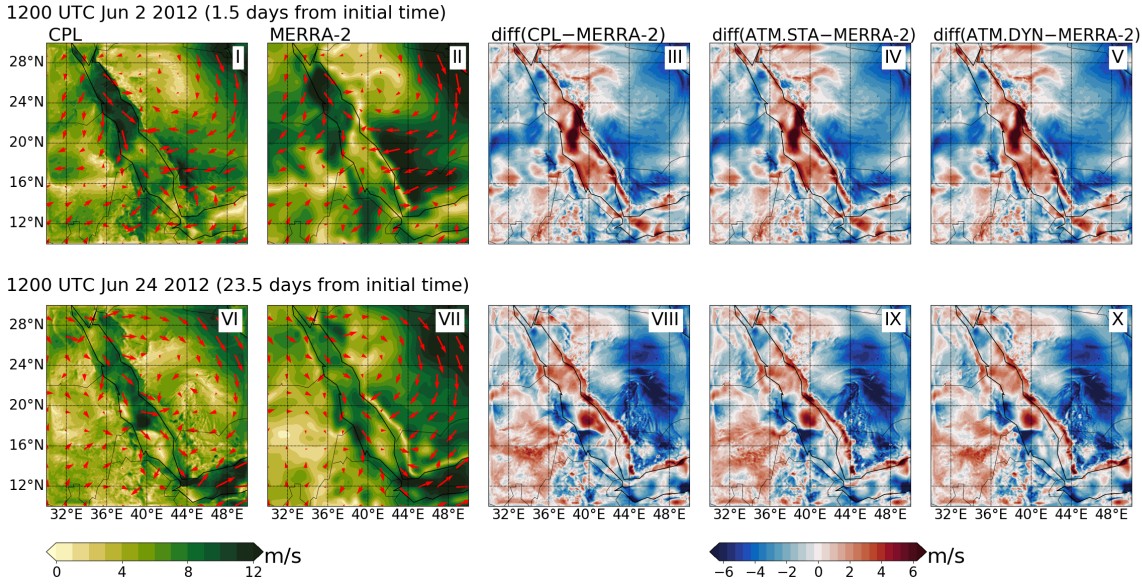

**Figure 12.** The magnitude and direction of the 10-m wind obtained in the CPL run, the MERRA-2 data, and their difference (CPL−MERRA-2). The differences between ATM.STA and ATM.DYN with MERRA-2 (i.e., ATM.STA−MERRA-2, ATM.DYN−MERRA-2) are also presented. Two snapshots are selected: (1) 1200 UTC Jun 02 2012; (2) 1200 UTC Jun 24 2012.

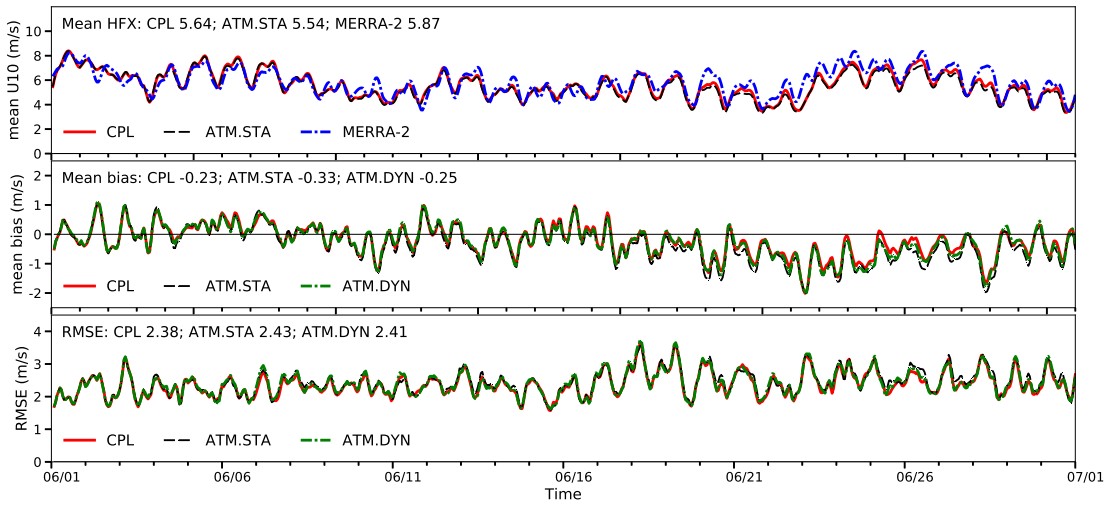

**Figure 13.** The magnitude of the 10-m wind obtained in CPL and ATM.STA runs in comparison with MERRA-2. The top panel shows the mean 10-m wind; the middle panel shows the mean bias; the bottom panel shows the RMSE. Only the hourly-averaged surface wind fields over the sea are shown to validate the coupled model.

## 5  Scalability Test

Parallel efficiency is crucial for coupled ocean–atmosphere models when simulating large and complex problems. In this section, the parallel efficiency in the coupled simulations is investigated. This aims to demonstrate (1) the implemented ESMF/NUOPC driver and model interfaces can simulate parallel cases effectively and (2) the ESMF/NUOPC coupler is not
a bottleneck of the coupled simulation. The parallel speed-up of the model is investigated to evaluate its performance for a constant size problem simulated using different numbers of CPU cores (i.e. strong scaling). Additionally, the CPU time spent on oceanic and atmospheric components of the coupled model is detailed. The parallel efficiency tests are performed on the Shaheen-II cluster in KAUST (https://www.hpc.kaust.edu.sa/). The Shaheen-II cluster is a Cray XC40 system composed of 6174 dual sockets compute nodes based on 16 cores Intel Haswell processors running at 2.3GHz. Each node has 128GB DDR4
memory running at 2300MHz. Overall the system has a total of 197,568 CPU cores (6147 nodes $\times$ 2 $\times$ 16 CPU cores) and has a theoretical peak speed of 7.2 PetaFLOPS ($10^{15}$ floating point operations per second).

The parallel efficiency of the scalability test is $N_{p0}t_{p0}/N_{pn}t_{pn}$, where $N_{p0}$ and $N_{pn}$ are the numbers of CPU cores employed in the simulation of the baseline case and the test case, respectively; $t_{p0}$ and $t_{pn}$ are the CPU time spent on the baseline case and the test case, respectively. The speed-up is defined as $t_{p0}/t_{pn}$, which is the relative improvement of the CPU time when
solving the problem. The scalability tests are performed by running 24-hour simulations for ATM.STA, OCN.DYN, and CPL cases. There are a total number of 2.6 million atmosphere cells (256 lat$\times$256 lon$\times$40 vertical levels) and 0.4 million ocean cells (256 lat$\times$256 lon$\times$40 vertical levels, but about 84% of the domain is land and masked out). We started using $N_{p0}$ = 32 because each compute node has 32 CPU cores. The results obtained in the scalability test of the coupled model are shown in Fig. 14. It can be seen that the parallel efficiency of the coupled code is close to 100% when employing less than 128 cores and is still
as high as 70% when using 256 cores. When using 256 cores, there are a maximum of 20480 cells (16 lat$\times$16 lon$\times$80 vertical levels) in each core. The decrease in parallel efficiency results from the increase of communication time, load imbalance, and I/O (read and write) operation per CPU core (Christidis, 2015). From results reported in the literature, the parallel efficiency of the coupled model is comparable to other ocean-alone or atmosphere-alone models when having similar number of grid points per CPU core (Marshall et al., 1997; Zhang et al., 2013b).
The CPU time spent on different components of the coupled run is shown in Table. 4. The time spent on the ESMF coupler is obtained by subtracting the time spent on oceanic and atmospheric components from the coupled run. The most time-consuming process is the atmospheric model integration, which accounts for 85% to 95% of the total costs. The ocean model integration is the second most time-consuming process, which is 5% to 11% of the total computational costs. The atmospheric model is much more time-consuming because it solves the entire computational domain, while the ocean model only solves
the Red Sea (16% of the domain). The atmospheric model also uses a smaller time step (30 s) than that of the ocean model (120 s). If a purely marine region is selected in an ideal case, the cost of ocean and atmosphere models would be more equal compared with the Red Sea case. The coupling process takes less than 3% of the total costs in the CPL runs using different numbers of CPU cores in this test. Although the proportion of the coupling process in the total costs increases when using more CPU cores, the total time spent on the coupling process is similar. The CPU time spent on two uncoupled runs (i.e.,

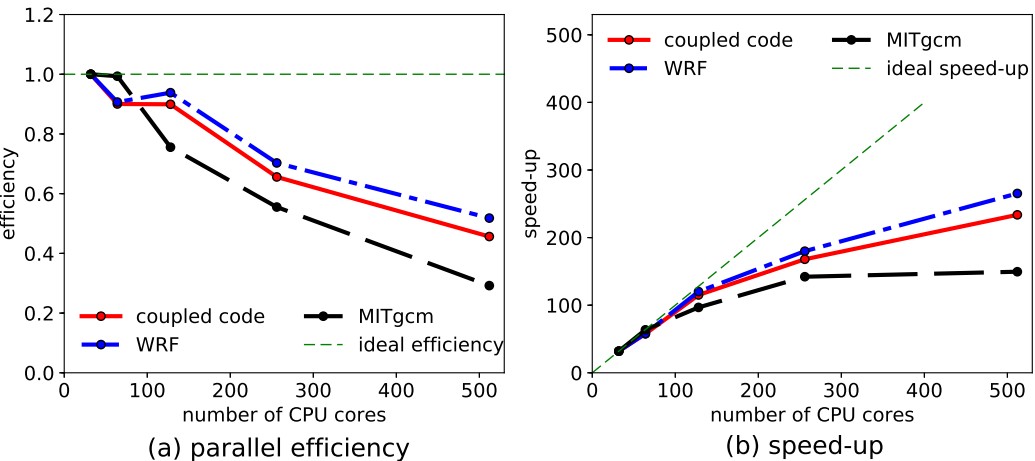

**Figure 14.** The parallel efficiency test of the coupled model in the Red Sea region. The test cases employ up to 512 CPU cores. The simulation using 32 CPU cores is regarded as the baseline case when computing the speed-up. Tests are performed on the Shaheen-II cluster in KAUST.

ATM.STA, OCN.DYN) is also shown in Table. 4. Compared with the uncoupled simulations, the ESMF-MITgcm and ESMF-WRF interfaces do not increase the CPU time in the coupled simulation. In summary, the scalability test results suggest that the ESMF/NUOPC coupler is not a bottleneck for using SKRIPS in coupled regional modeling studies.

**Table 4.** Comparison of CPU time spent on the coupled and stand-alone runs. The CPU time presented in the table is normalized by the time spent on the coupled run using 512 CPU cores. The CPU time spent on two stand-alone simulations are presented to show the difference between the CPL run and stand-alone simulations.

|  | $N_p = 32$ | 64 | 128 | 256 | 512 |
|---|---|---|---|---|---|
| CPL run | 7.27 | 4.04 | 2.02 | 1.39 | 1.00 |
| WRF in CPL run | 6.88(95%) | 3.82(94%) | 1.89(93%) | 1.25(90%) | 0.85(85%) |
| MITgcm in CPL run | 0.37( 5%) | 0.19( 5%) | 0.12( 6%) | 0.11( 8%) | 0.11(11%) |
| Coupler in CPL run | 0.02( 0%) | 0.03( 1%) | 0.02( 1%) | 0.03( 2%) | 0.03( 3%) |
| ATM.STA run | 6.89 | 3.80 | 1.84 | 1.22 | 0.83 |
| OCN.DYN run | 0.38 | 0.19 | 0.13 | 0.09 | 0.08 |

## 6  Conclusion and Outlook

5   This paper describes the development of the Scripps–KAUST Regional Integrated Prediction System (SKRIPS). To build the coupled model, we implement the coupler using ESMF with its NUOPC wrapper layer. The ocean model MITgcm and the

atmosphere model WRF are split into initialize, run, and finalize sections, with each of them being called by the coupler as subroutines in the main function.

The coupled model is validated by using a realistic application to simulate the heat events in the Red Sea region. To validate the coupled model, results from the coupled and stand-alone simulations are compared to a wide variety of available observational and reanalysis data. We focus on the comparison of the surface atmospheric and oceanic variables because they are used to drive the oceanic and atmospheric components in the coupled model. From the comparison, results obtained from various configurations of coupled and stand-alone model simulations all realistically capture the surface atmospheric and oceanic variables in the Red Sea region over a 30-day simulation period. The 2-m air temperature in three major cities obtained in the CPL and ATM.DYN runs are comparable and better than the ATM.STA run. Other surface atmospheric fields (e.g., 2-m air temperature, surface heat fluxes, 10-m wind speed) in the CPL run are also comparable with the ATM.DYN run and better than the ATM.STA run over the simulation period. The SST obtained in CPL run is also better than the OCN.DYN run by about 0.1 °C compared with GHRSST.

The parallel efficiency of the coupled model is examined by simulating the Red Sea region using increasing number of CPU cores. The parallel efficiency of the coupled model is consistent with that of the stand-alone ocean and atmosphere models when using various number of CPU cores in the test. The CPU time associated with different components of the coupled simulations is also presented, showing the ESMF/NUOPC driver is not a bottleneck in the computation. Hence the coupled model can be implemented for coupled regional modeling studies on supercomputers with comparable performance as that attained by uncoupled stand-alone models.

The results presented here motivate further studies evaluating and improving this new regional coupled ocean–atmosphere model for investigating dynamical processes and forecasting applications. This regional coupled forecasting system can be improved by developing coupled data assimilation capabilities for initializing the forecasts. In addition, the model physics and model uncertainty representation in the coupled system can be enhanced using advanced techniques, such as stochastic physics parameterizations. Future work will involve exploring these and other aspects of further developing a regional coupled modeling system suited for forecasting and process understanding purposes.

*Code and data availability.* The coupled model, documentation, and tutorial cases used in this work are available at https://library.ucsd.edu/ dc/collection/bb1847661c, and the source code is maintained on Github https://github.com/iurnus/scripps_kaust_model. ECMWF ERA5 data are used as the atmospheric initial and boundary conditions. The ocean model uses the assimilated HYCOM/NCODA $1/12°$ global analysis data as initial and boundary conditions. To validate the simulated SST data, we use the OSTIA (Operational Sea Surface Temperature and Sea Ice Analysis) system in GHRSST (Group for High Resolution Sea Surface Temperature). The simulated 2-meter air temperature (T2) is validated against the ECMWF ERA5. The observed daily maximum and minimum temperatures from NOAA National Climate Data Center is used to validate the T2 in three major cities. Surface heat fluxes (e.g., latent heat fluxes, sensible heat fluxes, longwave and shortwave radiations), which are important for ocean–atmosphere interactions, are compared with MERRA-2 (Modern-Era Retrospective analysis for Research and Applications, version 2).

## Appendix A: Snapshots of Surface Heat Fluxes

The snapshots of the THFs in the simulations at 1200 UTC June $2^{nd}$ and $24^{th}$ are presented. It can be seen that all simulations reproduce the THFs reasonably well in comparison with MERRA-2. On June $2^{nd}$, all simulations exhibit similar THF patterns since they have the same initial conditions and similar SST fields. On the other hand, for the heat event on June $24^{th}$, CPL and ATM.DYN runs exhibit more latent heat fluxes coming out of the ocean (170 and 153 W/m$^2$) than that in ATM.STA run (138 W/m$^2$). The mean biases in CPL, ATM.DYN, and ATM.STA runs are 23.1 w/m$^2$, 5.1 w/m$^2$, and -9.5 w/m$^2$, respectively. Although the CPL run has larger bias at the snapshot, the averaged bias and RMSE in CPL run is smaller (shown in Fig. 10). Compared with the latent heat fluxes, the sensible heat fluxes in the Red Sea region are much smaller in all simulations (about 20 W/m$^2$). It should be noted that MERRA-2 has unrealistically large sensible heat fluxes in the coastal regions because the land points are 'contaminated' the values in the coastal region (Kara et al., 2008; Gelaro et al., 2017), and thus the heat fluxes in the coastal regions are not shown.

The net downward shortwave and longwave heat fluxes are shown in Fig. A2. Again, all simulations reproduce the shortwave and longwave radiation fluxes reasonably well. For the shortwave heat fluxes, all simulations show similar patterns on both June $2^{nd}$ and $24^{th}$. The total downward heat fluxes, which is the sum of the results in Figs. A1 and A2, are shown in Fig. A3. It can be seen that the present simulations over-estimated the total downward heat fluxes on June $2^{nd}$ (CPL: 580 W/m$^2$; ATM.STA: 590 W/m$^2$; ATM.DYN: 582 W/m$^2$) for both heat events compared with MERRA-2 (525 W/m$^2$), especially in the southern Red Sea because of the over-estimation of the shortwave radiation. To improve the modeling of shortwave radiation, a better understanding of the cloud and aerosol in the Red Sea region is required (Zempila et al., 2016; Imran et al., 2018). Again, the heat fluxes in the coastal regions are not shown because of the inconsistency of land-sea mask. Overall, the comparison shows the present CPL simulations are capable of capturing the surface heat fluxes into the ocean.

## Appendix B: Evaporation

To examine the simulation of surface freshwater flux in the coupled model, the surface evaporation fields obtained from ATM.STA, ATM.DYN, and CPL runs are presented and validated using the MERRA-2 data.

The surface evaporation fields from CPL run are shown in Fig. B1. The MERRA-2 data and difference between CPL run and MERRA-2 are also shown to validate the CPL run. The ATM.STA and ATM.DYN simulation results are not shown, but their differences with the CPL run are also shown in Fig. B1. It can be seen in Fig. B1(III) and B1(VIII) that the CPL run reproduces the overall evaporation patterns in the Red Sea. The CPL run is able to capture the relatively high evaporation in the northern Red Sea and the relatively low evaporation in the southern Red Sea in both snapshots, shown in Fig. B1(I) and B1(VI). After 36-hours, the simulation results are close with each other (e.g., the RMSE between CPL and ATM.STA simulation is smaller than 10 cm/year). However, after 24 days, the CPL run agrees better with MERRA-2 (bias: 6 cm/year; RMSE: 59 cm/year) than the ATM.STA run (bias: -25 cm/year; RMSE: 68 cm/year). On the other hand, the CPL run results are consistent with the ATM.DYN run. This shows the CPL run can reproduce the realistic evaporation patterns over the Red Sea in the coupled

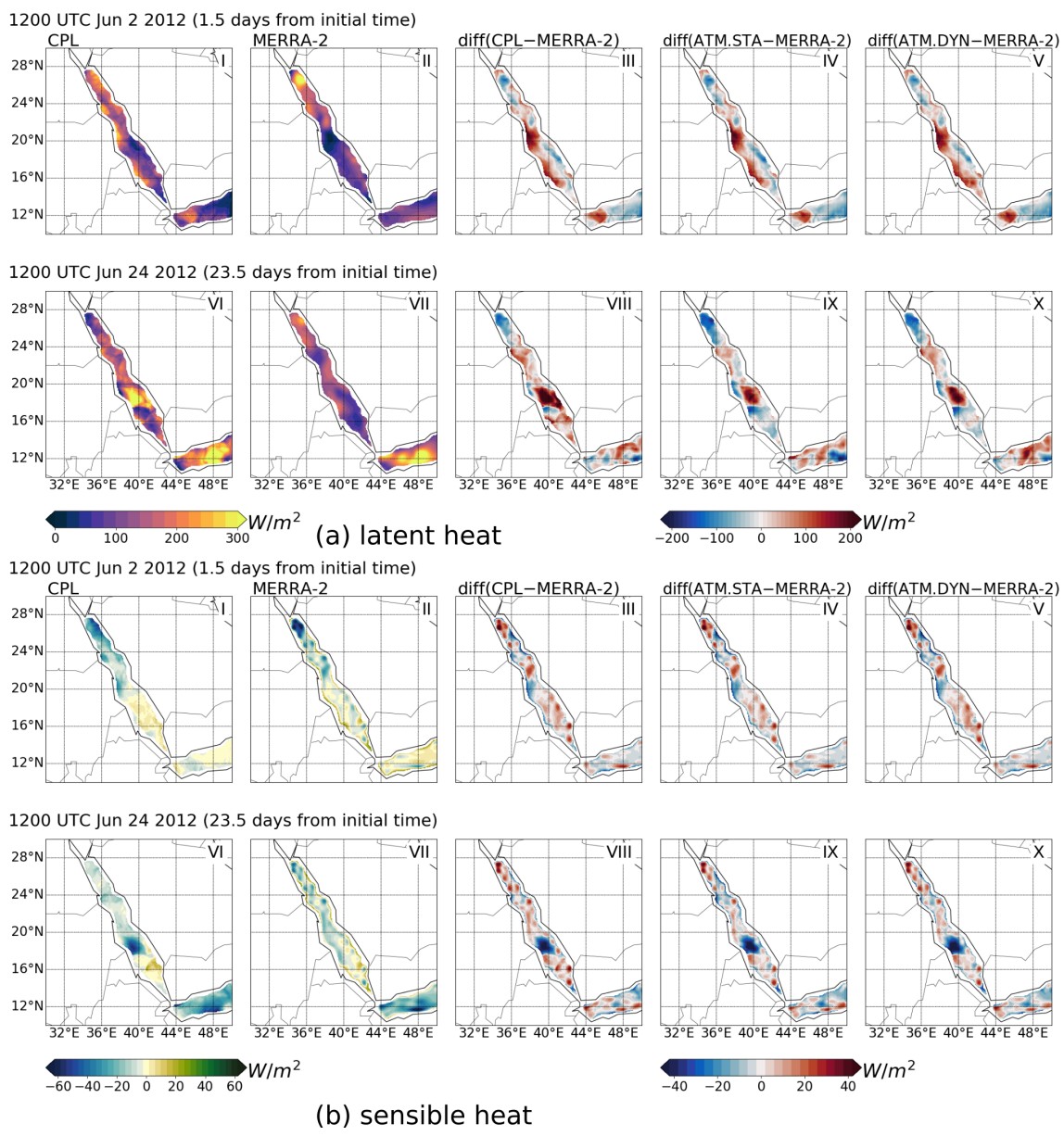

**Figure A1.** The turbulent heat fluxes out of the sea obtained in CPL run, MERRA-2 data, and their difference (CPL−MERRA-2). The differences between ATM.STA and ATM.DYN with MERRA-2 (i.e., ATM.STA−MERRA-2, ATM.DYN−MERRA-2) are also presented. Two snapshots are selected: (1) 1200 UTC Jun 02 2012; (2) 1200 UTC Jun 24 2012. The simulation initial time is 0000 UTC Jun 01 2012 for both snapshots. Only the heat fluxes over the sea are shown to validate the coupled model.

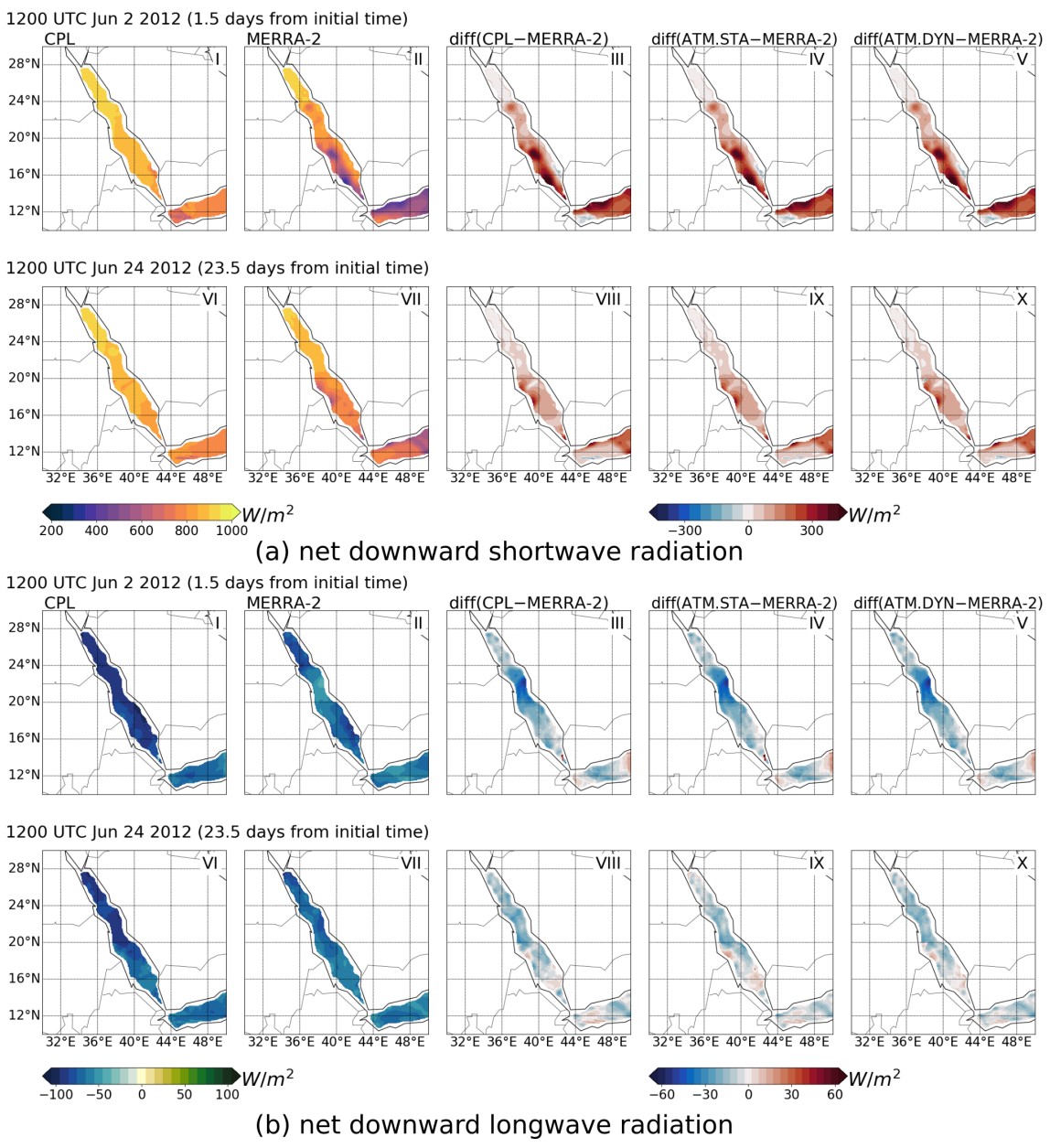

**Figure A2.** The net downward shortwave and longwave heat fluxes obtained in CPL run, MERRA-2 data, and their difference (CPL−MERRA-2). The differences between ATM.STA and ATM.DYN with MERRA-2 (i.e., ATM.STA−MERRA-2, ATM.DYN−MERRA-2) are also presented. Two snapshots are selected: (1) 1200 UTC Jun 02 2012; (2) 1200 UTC Jun 24 2012. The simulation initial time is 0000 UTC Jun 01 2012 for both snapshots. Only the heat fluxes over the sea are shown to validate the coupled model.

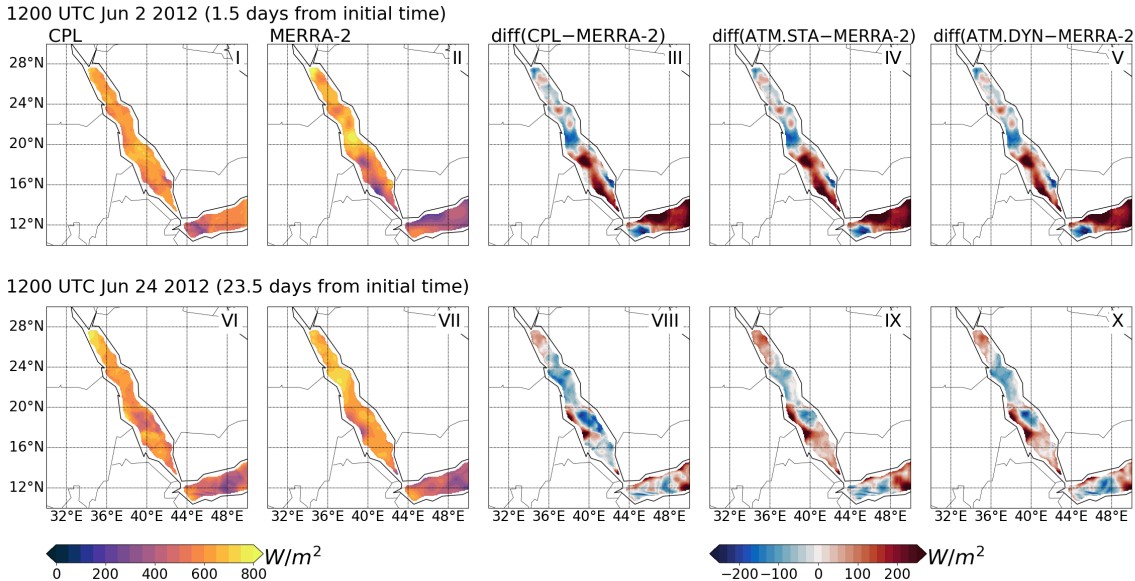

**Figure A3.** Comparison of the total downward heat fluxes obtained in CPL run, MERRA-2 data, and their difference (CPL−MERRA-2). The differences between ATM.STA and ATM.DYN with ERA5 (i.e., ATM.STA−MERRA-2, ATM.DYN−MERRA-2) are also presented. Two snapshots are selected: (1) 1200 UTC Jun 02 2012; (2) 1200 UTC Jun 24 2012. The simulation initial time is 0000 UTC Jun 01 2012 for both snapshots. Only the heat fluxes over the sea are shown to validate the coupled model.

ocean–atmosphere simulation. Since there is no precipitation in three major cities (Mecca, Jeddah, Yanbu) near the eastern shore of the Red Sea during the month according to NCDC climate data, the precipitation results are not shown.

*Author contributions.* RS worked on the coding tasks for coupling WRF with MITgcm using ESMF, wrote the code documentation, and performed the simulations for the numerical experiments. RS and ACS worked on the technical details for debugging the model and drafted the initial manuscript. All authors designed the computational framework and the numerical experiments. All authors discussed the results and contributed to the writing of the final manuscript.

*Competing interests.* The authors declare that they have no conflict of interest.

*Acknowledgements.* We appreciate the computational resources provided by COMPAS (Center for Observations, Modeling and Prediction at Scripps) and KAUST used for this project. We gratefully acknowledge the research funding from KAUST (grant number: OSR-2-16-RPP-3268.02). We are immensely grateful to Caroline Papadopoulos for helping with installing software, testing the coupled code, and using

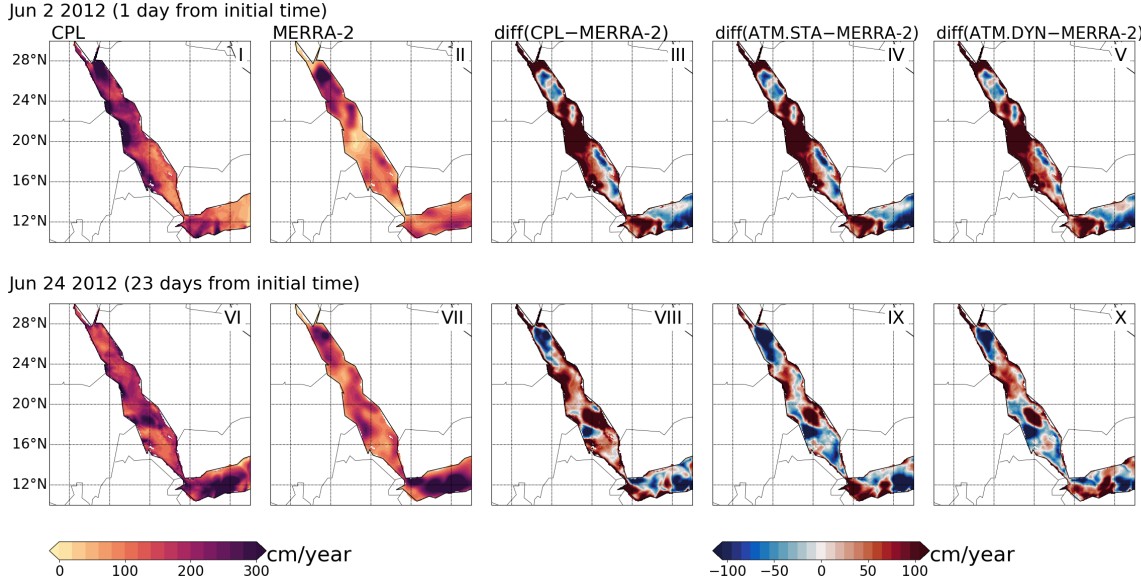

**Figure B1.** The surface evaporation patterns obtained in the CPL run, the MERRA-2 data, and their difference (CPL−MERRA-2). The differences between uncoupled atmosphere simulations with MERRA-2 (i.e., ATM.STA−MERRA-2, ATM.DYN−MERRA-2) are also presented. Two snapshots are selected: (1) 1200 UTC Jun 02 2012; (2) 1200 UTC Jun 24 2012. Only the evaporation over the sea is shown to validate the coupled ocean–Atmosphere model.

the HPC clusters. We appreciate Professor U.U. Turuncoglu sharing part of their ESMF/NUOPC code on GitHub which helped our code development. We wish to thank Dr. Ganesh Gopalakrishnan for setting up the stand-alone MITgcm simulation (OCN.DYN) and providing the external forcings. We thank Drs. Stephanie Dutkiewicz, Jean-Michel Campin, Chris Hill, Dimitris Menemenlis for providing their ESMF–MITgcm interface. We wish to thank Dr. Peng Zhan for discussing the simulations of the Red Sea. We also thank the reviewers for their insightful review suggestions.

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
