# Peer review of "SKRIPS v1.0: A regional coupled ocean—atmosphere modeling framework (MITgcm–WRF) using ESMF/NUOPC, description and preliminary results for the Red Sea"

_Geoscientific Model Development, 2018_

## Short Comment (SC1) · 18 Dec 2018

Dear authors,

in my role as Executive editor of GMD, I would like to bring to your attention our Editorial version 1.1: http://www.geosci-model-dev.net/8/3487/2015/gmd-8-3487-2015.html This highlights some requirements of papers published in GMD, which is also available on the GMD website in the 'Manuscript Types' section: http://www.geoscientific-modeldevelopment.net/submission/manuscript_types.html In particular, please note that for your paper, the following requirement has not been met in the Discussions paper:

- "The main paper must give the model name and version number (or other unique identifier) in the title."

Please provide the version numbers of MITgcm-WRF and ESMF/NUOPC in the title of your revised manuscript.

As explained in https://www.geoscientific-model-development.net/about/manuscript_types.html. GMD is encouraging authors to upload the program code of models (including relevant data sets) as supplement or make the code and data of the exact model version described in the paper accessible through a DOI (digital object identifier). In case your institution does not provide the possibility to make electronic data accessible through a DOI you may consider other providers (eg. zenodo.org of CERN) to create a DOI. Please note that in the code availability section you can still point the reader to how to obtain the newest version. If for some reason the code and/or data cannot be made available in this form the "Code Availability" section need to clearly state the reasons for why access is restricted (e.g. licensing reasons).
Especially, please note, that it is not enough, that the code will be available in the future. It must be available now and the exact version of the code published in this article needs to be made available.

Yours, Astrid Kerkweg

---

## Referee Comment (RC1) · Gill (Referee) · 19 Jan 2019

Comments

This is a review of "A regional coupled ocean–atmosphere modeling framework (MITgcm–WRF) using ESMF/NUOPC: description and preliminary results for the Red Sea", by Sun et al.

General Comments

This is an article describing the development of a geophysical model, a regional coupled ocean - atmosphere model, which fits perfectly with the journal. Having worked closely with atmosphere models for years and dabbled with coupled modeling systems, this coupled system represents an enormous effort. The timing numbers and physical results indicate that the coupled system appears to be behaving correctly and is ready for further testing. My concerns are largely with the presentation of this new tool to the community. The actual development and engineering aspects of the coupled system, which should be the priority, tend to take a back seat to a lengthy description of the physical results from a series of test cases, and the process of identifying those test cases could have been more discriminating. A better showcase of modeled physical phenomenon exists for a coupled ocean - atmosphere system than choosing a data sparse region with no obvious ocean - atmosphere feedback mechanism to model. Finally, some of the discussion about parallel issues is misleading regarding statement of fact, leading to flawed assumptions concerning implications.

There are a number of existing coupled regional ocean - atmosphere systems available to the community. With "framework" in the title, I was assuming that this article was more about a technique or related to some new tools that would assist or improve the infrastructure of model coupling.

Specific Comments

The first sentence of the abstract is that a new regional coupled model is developed. The authors then proceed to present a justification which they do not back up. 1. This will be a "new coupled regional ocean-atmosphere model with 'state of the art' physics and using modern framework". Coupled regional models within the past ten years or so include: FROALS, SCOAR, CROAM, COAWST, COSMO. Some of these models also include data assimilation, chemistry, waves, sediment transport. Various modern toolkits are used for coupling. There is plenty of state-of-the-art in the existing systems.

2. In the comparisons of 2-m temperatures for several episodic events, the month-long

diurnal 2-m temperatures, the month-long plot of deviation and RMSE, snap shots of SST over the Red Sea, and the deviations of the SST vs HYCOM and GHRSST the authors state that the regional coupled model behaved similarly to a standalone model with "dynamic" SST. So, the authors indicate that the existing (and much simpler) stand alone models work as well as the coupled system.

3. As the title says, these are preliminary results. However, there are existing global models running at 9 km, so a study of "small-scale processes" for a regional coupled model (specifically set up with an 8 km resolution for this case study) does not seem to be the best possible demonstration of the available capabilities. The selected verification data sets are fairly coarse resolution and the verification techniques are those traditionally employed for large-scale fields: bias and RMSE. The only indication of high resolution is the oft repeated "high-resolution" phrase.

4. For a demonstration of the benefits of a coupled ocean - atmosphere system, one would expect some sort of traditional ocean - atmosphere feedback mechanism to be on display: tides, storm surge, post-hurricane cold wake, inundation, sea breeze, etc.. A heatwave event in a desert region does not seem to identify and highlight the new model's coupled capabilities.

As both the atmosphere and the ocean model are widely used in the public, specific details of the changes in those codes would appropriate. For the atmosphere model component, it would be nice to know details of how the ocean model's data is hooked into the WRF model's surface layer scheme. Are the surface layer tendencies constant during the intervals between coupling? Was any sensitivity seen in this coupling frequency?

Quite a number of examples point to poor atmospheric surface comparisons after the Red Sea SST is kept constant for a month. This is physically unrealistic. While this constant SST test case may serve as a data point, a month-long constant SST experiment should not be the primary comparison to display the skill of the new coupled

modeling system.

Buried towards the end of the paper is a mention of the importance of the resultant size of the decomposed domain with strong scaling. That a reasonably well designed atmosphere or ocean model scales to 128 processors is germaine only so far as we know the number of computational cells within that MPI rank.

There are several statements that would be easy to verify, and likely that the authors' stated reason is not among the top contenders.

1. "This may be attributed to the fluctuation of the CPU time when solving the systems of linear equations. When using different numbers of processors, the decomposition of the domain leads to different linear equation systems requiring different CPU load and accordingly different convergence time." The atmosphere model accounts for 75-90% of the elapsed time, and the WRF model does not solve linear systems with convergence criteria. This assertion is not defensible.

2. "This is likely because the simulations on T2 suffer from the mismatches between the model terrain and the actual terrain, especially over complex mountains". Smooth the model data to the resolution of the validating analysis to check this out. The domain is mostly a desert, and atmosphere models tend to underestimate and bias the amplitude of the diurnal surface air temperature. Atmosphere models tend to do poorly with a diurnal amplitude when the observation site is on a coast when the sea breeze effect is not well captured. Any number of quick tests are available to find why the T2 behavior is not as expected.

3. "However, when using 256 processors, the proportion of this cost increases to 10% because of the increase of inter-processor communication with more processors." The per-MPI task cost of communications is approximately constant, but the relative cost of the communication compared to the computation becomes important as the amount of work is reduced as the number of grid cells is reduced during strong scaling tests.

Referring to 128 processors as a "large number of processors" is inaccurate for either a state of the art atmosphere or ocean model. The atmosphere model domain is 256x256 grid cells. The coupled system using this atmosphere domain is not really "simulating large" problems.

Stating that scaling to a large number of processors makes a model applicable to "high-resolution" studies does not logically follow. The scaling test was for strong scaling, the problem size remained identical. The problem was the same "high-resolution" with 8 MPI ranks as with 256 MPI ranks. Scaling, as used in this study, implies that the same problem size gets done faster.

Once the atmosphere model is decomposed onto 256 MPI ranks, the resulting computational area is 16x16 grid cells. To indicate that scaling performance tails off, the relative cost of computation and communication needs to be brought up. "The boundary tiles in each processor are 25% of the total, and the parallel communication cost increases significantly." For the atmosphere model, depending on the communication stencil, between approximately 20 and 60% of the computational area could be communicated. But until we know the relative cost of computations and communications, we are left with "cost increases significantly". There is nothing actionable for a user in that statement. Worse, users are left with the impression that after 256 MPI ranks, the communication costs increase significantly for all model configurations.

When assigning relative costs between the atmosphere and ocean model, the most important factor of the ratio of the number of computational cells between the two models is ignored and "more complex physics parameterization packages" is offered. A clear and accurate representation is the relative cost of a single-column of the atmosphere model compared to a single column of the ocean model, for a single time step. This permits a user to assign MPI ranks to different model components.

A number of external sources (books and online) define a "mean deviation" to be the same as mean absolute deviation. Perhaps bias is a better term and seems to be what

the authors are interested in (warmer, cooler, etc).

Care has to be taken when using words that have a typical meaning in a field, but that meaning is not intended.

1. significant: "but the increase is not significant after that".

2. To an atmosphere model user, the term "micro-physics" refers to the bulk or bin parameterization schemes that deal with resolved scale moist processes. "WRF micro-physics models (e.g., land surface model, the PBL model)"

Given that 30 days * 24 hr/day * 60 min/hr * 2 model time steps / min = 86k atmosphere model time steps (which each have an elapsed time reported individually), there should be a set of either error bars or standard deviation on all of the reported timing values. Similar statistical information is missing from the differences of the physical fields.

In the description of the atmosphere model, the option for the resolved-scale moist physics is not mentioned. The atmosphere model lid for the vertical coordinate is not provided. A single, deterministic simulation for a month should probably use spectral nudging in the WRF model to keep the large-scale atmospheric flow in check. An atmospheric modeler would find the model setup section incomplete.

The discussion of the selection of the ocean model should have included the benefits and applicability near coastal areas, how the horizontal and vertical resolution are modified in shallow coastal areas, the impact of the broad shallow portions of the Red Sea on the vertical levels, spin-up time from initial conditions, the sensitivity of correctly choosing boundary locations, and coupling frequency.

It is not a fair comparison to make when you keep the SST constant for a month: "Improvements of the coupled model over the stand-alone simulation with static SST forcing are observed in capturing the T2, heat fluxes, evaporation, and wind speed." Also, it was stated that the momentum fields were not impacted in your study: "This suggests that the ocean–atmosphere coupling does not significantly influence the wind

field in the Red Sea region during the heat wave events."

It is not conventional to have a statement as this in the conclusions: "On the other hand, the difference between coupled simulation and stand-alone simulations with updated forcings is also discussed."

When verifying an 8-km domain with 30-km gridded results, briefly describe the process.

Without some sort of statistical assistance, we do not know if -1.55 is mostly the same as or pretty different from -1.66. "The mean T2 differences over the sea are -1.55 âŮęC (CPL), -1.66 âŮęC (ATM.STA), and -1.70 âŮęC (ATM.DYN) after 36 hours, and -0.99 âŮęC (CPL), -1.10 âŮęC (ATM.STA), and -1.12 âŮęC (ATM.DYN) after 48 hours."

Figure 1b has lots of arrows. Are they one-way only? If the parent talks to the child components directly, why is there a child coupler component?

Most people would take Figure 1c to be an indication that modeling system is running concurrently.

Are there computational trade-offs for selecting a sequential rather than concurrent coupling mode? Does your implementation preclude selecting sequential vs concurrent as a build- or run-time option?

If a purely marine region was selected, is there an expectation that the cost of the atmosphere and ocean models would more equal?

Do the atmosphere and ocean models run on the same processor set? If so, are the parallel tests hampered with fewer ocean points as the number of processors are increased?

After 1 day of simulation, why is the modeled SST so much colder than observed in Figure 7?

There are several instances of trying to read too much into differences of the fields:

"On the other hand, for the heat wave event on June 24th, CPL and ATM.DYN runs exhibit more latent heat fluxes coming out of the ocean (157 and 131 W/m2) than that in ATM.STA run (115 W/m2)." OK, yes, but if you look at Figure 9a, VIII vs IX vs X, IX and X are more similar than VIII and X.

Technical Corrections

Figure 1 is pretty busy. 1a mentions "using bulk formulas" instead of listing the variables passed.

In several places, "access" and "assess" are swapped.

Table 1 has ATM.STA twice. The second should be ATM.DYN.

Figure 5 would benefit from having some highlight that indicated the four heatwave periods.

A number of figures would benefit from smaller color bar ranges. For example 10a has a range from about -500 W/mˆ2 to 500 w/mˆ2. The text says "However, a small improvement in the CPL (2.19 W/m2) and ATM.DYN (1.27 W/m2) runs can be observed in the longwave radiation on June 24th".

Both are used "Arakawa-C grid" and "Arakawa C-grid".

There are some clumsy wordings "This run allows to access the WRF model", "which means that the SST in CPL run is tending to be similar to the realistic."

Figure 3 has a gray bar that covers the table bar.

Figure 6 misspells deviation

Figure 6 has diffs, diffs of diffs, rmse of diffs, and diffs of rmse of diffs. The y-axis labeling and the in-plot descriptions should be more precise.

Page 15, line 2, ATM.STA should be ATM.DYN?

---

## Referee Comment (RC2) · Anonymous Referee #2 · 22 Jan 2019

Overall comments:

It is neither a technical nor a science paper. It would be beneficial to re-focus the manuscript on one aspect by clearly stating the problem, hypotheses and discuss the findings. Based on a few snippets of the manuscript it comes across that the authors are vaguely familiar with the foundations of numerical modeling in the atmosphere; they got two open source models, coupled them (no small feat!), and ran a test case. What is missing is a critical look at the approach, results, discussion of why things worked and more importantly, why not. I suggest to omit the whole section on scalability. The

experiment design does not support any meaningful conclusions for scaling purposes. I would also recommend proof-reading (not spell checking!) the manuscript.

Specific comments:

Page 6, lines 21-32: Too technical. Page 6, line 32: Why was sequential mode selected? Page 7, lines 17-31: Is it a 30-day long run? How frequently are you forcing lateral boundary conditions? What is the projection? What is the lateral boundary condition type? Why is coupling every 20 minutes? Why 8 km grid spacing? Page 8, lines 13, 16: 'accessing', 'accesses' should be 'assessing', 'assesses' Page 9, Table 1: The second ATM.STA should probably be ATM.DYN. Page 9, line 9: Is MERRA-2 really an independent data compared to ERA5? The forecast model is, but the observations do overlap quite a bit. Page 10, line 21-22: Why not use a nest with finer grid spacing to resolve the local topography? Page 10, lines 10-32: When comparing to ERA5 data, how were the statistics computed? Was the model output interpolated onto the observation points in space and time? Page 11, Figure 3: There was a gray stripe at the bottom, making it impossible to read labels of the color bars. Page 11, line 16: Land surface model and PBL model are not microphysics models. Page 13, Figure 5: Are there missing data points for the observed high and low T2, e.g. Mecca and Yanbu 6/21, Yanbu 6/8, 6/10, 6/14. . .? Page 14, Figure 6: Are model points interpolated to ERA5 points over the Red Sea? Which simulation is ATM.CPL, it has not been introduced in Table 1? How do you explain the drift (blue line)? How can RMSE be negative in the lower right figure?! Page 14, line 4-6: Where there many clouds present during that period? Page 14, line 4-6, and line 12-13: First you state the forcing is different due to 'uncertainty in cloud modeling', then you state 'both simulations are driven by realistic atmospheric forcing'. Which one is correct? Please explain. Page 16, line 13-14: Any cloud comparison? Page 16, Figure 8: Why no time series comparison to MERRA-2 dataset? Page 19, line 6: Which selected micro-physics schemes? Page 20, line 3: 64 /cm/year should read 64 cm/year? Page 21: line 8-9: What does it mean 'The decrease in parallel efficiency is because when using 256 processors, there are

only 16x16 grid points in the horizontal plane'? Page 21: line 11-13: Please elaborate: 'This may be attributed to the fluctuation of the CPU time when solving the systems of linear equations. When using different number of processors, the decomposition of the domain leads to different linear equation systems, requiring different CPU load and accordingly different convergence time. Page 21-23: Did you try weak or strong scaling? What is the communication cost? I/O cost? How many grid points per core are recommended? Are you reporting average times of multiple simulations in Table 3? How does WRF scale, MITgcm scale – do your results fit? Why did the coupling cost increase when using more cores? Page 23, Table 3: Please use the experiment names consistently throughout the manuscript.

---

## Author Comment (AC1) · 17 Apr 2019

*In my role as Executive editor of GMD, I would like to bring to your attention our Editorial version 1.1. This highlights some requirements of papers published in GMD, which is also available on the GMD website in the 'Manuscript Types' section. In particular, please note that for your paper, the following requirement has not been met in the Discussions paper:*

*"The main paper must give the model name and version number (or other unique identifier) in the title."*

*Please provide the version numbers of MITgcm–WRF and ESMF/NUOPC in the title of your revised manuscript.*

**Reply 1:** The authors thank the executive editor for pointing out this issue.

We have revised the title to include the name and version of our model. The versions of the ocean/atmosphere model components and the ESMF coupler are also highlighted in the manuscript.

Now the title of the manuscript is: SCRIPS v1.0: A regional coupled ocean–atmosphere modeling framework (MITgcm–WRF) using ESMF/NUOPC, description and preliminary results for the Red Sea

**Comment 2:** *GMD is encouraging authors to upload the program code of models (including*

*relevant data sets) as supplement or make the code and data of the exact model version described in the paper accessible through a DOI (digital object identifier). In case your institution does not provide the possibility to make electronic data accessible through a DOI you may consider other providers (eg. zenodo.org of CERN) to create a DOI. Please note that in the code availability section you can still point the reader to how to obtain the newest version. If for some reason the code and/or data cannot be made available in this form the "Code Availability" section need to clearly state the reasons for why access is restricted (e.g. licensing reasons).*

*Especially, please note, that it is not enough, that the code will be available in the future. It must be available now and the exact version of the code published in this article needs to be made available.*

**Reply 2:**    The authors thank the executive editor for pointing out this in the code availability section. We have uploaded our source code, test cases, and code documentation to a GitHub repository `https://github.com/iurnus/scripps_kaust_model`. This repository is now open to public. We have added the link of the GIT repository to the manuscript.

---

## Author Comment (AC2) · 17 Apr 2019

**Reply to Reviewer Comment 1 (RC1)**

Rui Sun, Aneesh C. Subramanian, Arthur J. Miller, Matthew R. Mazloff,
Ibrahim Hoteit, Bruce D. Cornuelle

April 16, 2019

The authors thank the reviewer for their insightful comments. To adequately address the concerns raised by Reviewer 1 in the original manuscript, we have made the following changes:

1. We have augmented the text to include more technical details.

2. We have revised the introduction section and highlighted the innovation of our work.

3. The discussion of the parallel efficiency has been re-written.

We have also proofread our manuscript according to the reviewer's comments. Some paragraphs are rewritten to address the concerns raised by the reviewer. We have also attached an annotated manuscript to highlight the revisions.

Detailed replies to specific comments by the reviewer are presented below:

**Comment 1:** *This is a review of "A regional coupled ocean–atmosphere modeling framework (MITgcm–WRF) using ESMF/NUOPC: description and preliminary results for the Red Sea", by Sun et al.*

*This is an article describing the development of a geophysical model, a regional coupled ocean-atmosphere model, which fits perfectly with the journal. Having worked closely with atmosphere models for years and dabbled with coupled modeling systems, this coupled system represents an enormous effort. The timing numbers and physical results indicate that the coupled system appears to be behaving correctly and is ready for further testing. My concerns are largely with the presentation of this new tool to the community. The actual development and engineering aspects of the coupled*

*system, which should be the priority, tend to take a back seat to a lengthy description of the physical results from a series of test cases, and the process of identifying those test cases could have been more discriminating. A better showcase of modeled physical phenomenon exists for a coupled ocean - atmosphere system than choosing a data sparse region with no obvious ocean-atmosphere feedback mechanism to model. Finally, some of the discussion about parallel issues is misleading regarding statement of fact, leading to flawed assumptions concerning implications.*

**Reply 1:** The authors thank the reviewer for acknowledging our efforts in building the coupled modeling system. We also thank the reviewer for their comments, which have helped us improve the quality of the manuscript.

**Comment 2:** *There are a number of existing coupled regional ocean-atmosphere systems available to the community. With "framework" in the title, I was assuming that this article was more about a technique or related to some new tools that would assist or improve the infrastructure of model coupling.*

**Reply 2:** We thank the reviewer for pointing out this. We have edited the paper to focus it as a technical introduction to the model. We intend this paper to be technical documentation explaining the new modeling capability, with a demonstration that has scientific value as well.

**Comment 3:** *The first sentence of the abstract is that a new regional coupled model is developed. The authors then proceed to present a justification which they do not back up.*

*1. This will be a "new coupled regional ocean-atmosphere model with 'state of the art' physics and using modern framework". Coupled regional models within the past ten years or so include: FROALS, SCOAR, CROAM, COAWST, COSMO. Some of these models also include data assimilation, chemistry, waves, sediment transport. Various modern toolkits are used for coupling. There is plenty of state-of-the-art in the existing systems.*

**Reply 3:** We agree with the reviewer that many regional coupled models have been developed using modern model toolkits. We also agree that some of these models include data assimilation, waves, sediment transport, and chemistry packages. The innovations in our work are: (1) we used

ESMF/NUOPC, which is a community supported computationally efficient coupling software for earth system models; (2) we used MITgcm and WRF, and the coupled model is being developed as a coupled forecasting tool for coupled data assimilation and S2S forecasting. By coupling of WRF and MITgcm for the first time with ESMF, we provide a regional coupled model resource to a wider community of users. These atmosphere and ocean model components have an active and well-supported user-base.

The introduction now includes a brief review of the regional coupled models developed and used over the past ten years. We have also added a discussion of our purpose in developing this new regional coupled model. We thank the reviewer for highlighting some of the vague descriptors (state-of-the-art, high res, etc.) and we have replaced these terms throughout the manuscript using more quantitative or informative words.

**Comment 4:** *2. In the comparisons of 2-m temperatures for several episodic events, the month-long diurnal 2-m temperatures, the month-long plot of deviation and RMSE, snap shots of SST over the Red Sea, and the deviations of the SST vs HYCOM and GHRSST the authors state that the regional coupled model behaved similarly to a standalone model with "dynamic" SST. So, the authors indicate that the existing (and much simpler) stand alone models work as well as the coupled system.*

**Reply 4:** We compared the coupled model with the stand-alone models with "dynamic" SST. By doing this, we aim to show that we successfully coupled the two models. In the forecast, one would not have evolving SST. We have made this point clear in the text.

In this case the goal of the simulation is to accurately estimate the ocean's effect on the atmosphere, and having the coupled SST forecast match the reanalysis is a positive outcome. We did not describe this well in the text and we have revised accordingly.

**Comment 5:** *3. As the title says, these are preliminary results. However, there are existing global models running at 9 km, so a study of "small-scale processes" for a regional coupled model (specifically set up with an 8 km resolution for this case study) does not seem to be the best possible demonstration of the available capabilities. The selected verification data sets are fairly coarse resolution and the*

*verification techniques are those traditionally employed for large-scale fields: bias and RMSE. The only indication of high resolution is the oft repeated "high-resolution" phrase.*

**Reply 5:** We thank the reviewer for pointing this out. Our aim is towards high-resolution process studies and experimental forecasting. These models have both been shown effective at simulating kilometer-scale processes in the ocean and atmosphere. We agree that we have overused the phrase high-resolution and have edited the text. We now make it clear that our example is a 0.08 $^\circ$ grid spaced model. We have emphasized our motivation in the introduction that these models do give the capability to run (down-scaled) detailed process study simulations.

**Comment 6:** *4. For a demonstration of the benefits of a coupled ocean-atmosphere system, one would expect some sort of traditional ocean-atmosphere feedback mechanism to be on display: tides, storm surge, post-hurricane cold wake, inundation, sea breeze, etc.. A heatwave event in a desert region does not seem to identify and highlight the new model's coupled capabilities.*

**Reply 6:** We investigate the heat wave events because they are extreme events and have societal relevance. We have added the discussion on why we selected the heat wave events in Section 3.

**Comment 7:** *As both the atmosphere and the ocean model are widely used in the public, specific details of the changes in those codes would appropriate. For the atmosphere model component, it would be nice to know details of how the ocean model's data is hooked into the WRF model's surface layer scheme. Are the surface layer tendencies constant during the intervals between coupling? Was any sensitivity seen in this coupling frequency?*

**Reply 7:** We did not modify the WRF model's default surface layer scheme (version 3.9.1.1). We use the coupler to provide the ocean's SST and surface velocity as the bottom boundary condition. The WRF model's surface layer schemes can read the updated boundary condition at each coupling interval. The SST and surface ocean velocity are considered constant during each coupling interval.

We select the coupling frequency to resolve the diurnal cycle according to the reference (Seo et al., 2014). We have added this in our revised manuscript. Actually, we tried a few different coupling intervals (2 min, 20 min, and 180 min), but the diurnal cycle is not sensitive to the coupling

frequency in this case.

**Comment 8:** *Quite a number of examples point to poor atmospheric surface comparisons after the Red Sea SST is kept constant for a month. This is physically unrealistic. While this constant SST test case may serve as a data point, a month-long constant SST experiment should not be the primary comparison to display the skill of the new coupled modeling system.*

**Reply 8:** We agree with the reviewer that the ATM.STA run is a physically unrealistic scenario. In our manuscript, we use the results obtained in ATM.STA as the baseline case to show the difference between coupled and uncoupled runs. We have emphasized that the constant SST test case is unrealistic in our manuscript and detailed our motivation in running the constant SST test. Please refer to Section 3.

**Comment 9:** *Buried towards the end of the paper is a mention of the importance of the resultant size of the decomposed domain with strong scaling. That a reasonably well designed atmosphere or ocean model scales to 128 processors is germaine only so far as we know the number of computational cells within that MPI rank.*

**Reply 9:** We thank the reviewer for pointing out this. We have added this to the discussion on the parallel efficiency and the number of CPUs. Although we only used up to 256 CPUs, the parallel efficiency is satisfactory when we only have about 20,000 grid cells in the coupled model. Our results are also consistent with other parallel efficiency test using similar number of processors (Zhang et al., 2013).

**Comment 10:** *There are several statements that would be easy to verify, and likely that the authors' stated reason is not among the top contenders.*

*1. "This may be attributed to the fluctuation of the CPU time when solving the systems of linear equations. When using different numbers of processors, the decomposition of the domain leads to different linear equation systems requiring different CPU load and accordingly different convergence time." The atmosphere model accounts for 75- 90% of the elapsed time, and the WRF model does not solve linear systems with convergence criteria. This assertion is not defensible.*

**Reply 10:** We thank the reviewer for pointing out this. We have removed this sentence and revised our discussion according to the literature (Christidis, 2015). The oscillation of the CPU time might be because of the increase of communication time, load imbalance, and I/O (read and write) operation per processor.

**Comment 11:** *2. "This is likely because the simulations on T2 suffer from the mismatches between the model terrain and the actual terrain, especially over complex mountains". Smooth the model data to the resolution of the validating analysis to check this out. The domain is mostly a desert, and atmosphere models tend to underestimate and bias the amplitude of the diurnal surface air temperature. Atmosphere models tend to do poorly with a diurnal amplitude when the observation site is on a coast when the sea breeze effect is not well captured. Any number of quick tests are available to find why the T2 behavior is not as expected.*

**Reply 11:** We thank the reviewer for pointing out this. The test case in this manuscript aims to demonstrate the ocean and atmosphere components are successfully coupled. We have rewritten the motivation to have greater emphasis on the technical details. The discussion of the bias from the mismatch of the terrain is removed from the revised manuscript.

Yes, the domain is mostly desert and the atmospheric model tends to underestimate the diurnal surface air temperature. Our simulation captures much of the diurnal temperature cycle in Fig. 6, but the bias is obvious. We have revised our manuscript in Section 4.

**Comment 12:** *3. "However, when using 256 processors, the proportion of this cost increases to 10% because of the increase of inter-processor communication with more processors." The per-MPI task cost of communications is approximately constant, but the relative cost of the communication compared to the computation becomes important as the amount of work is reduced as the number of grid cells is reduced during strong scaling tests.*

**Reply 12:** We thank the reviewer for the comments. Yes, our results show the per-MPI task cost of communications is approximately constant in Table 3. We also find the relative cost of the communication compared to the computation increases as the amount of computation work per-MPI is reduced. We have added this in Section 5. We have also revised our discussions on the

parallel efficiency.

**Comment 13:** *Referring to 128 processors as a "large number of processors" is inaccurate for either a state of the art atmosphere or ocean model. The atmosphere model domain is 256x256 grid cells. The coupled system using this atmosphere domain is not really "simulating large" problems.*

**Reply 13:** We thank the reviewer for pointing this out. We agree that 128 processors is not "a large number of processors" in the context of earth system model computations, and that our domain using 256x256 grid cells is not "simulating large problems" compared to global weather and climate models or regional cloud resolving models. We have re-written this discussion in the revised manuscript.

**Comment 14:** *Stating that scaling to a large number of processors makes a model applicable to "high-resolution" studies does not logically follow. The scaling test was for strong scaling, the problem size remained identical. The problem was the same "high-resolution" with 8 MPI ranks as with 256 MPI ranks. Scaling, as used in this study, implies that the same problem size gets done faster.*

**Reply 14:** Thanks. We agree with the reviewer on this and we have revised our manuscript. We changed 'scaling' to 'speed-up' in the revision.

**Comment 15:** *Once the atmosphere model is decomposed onto 256 MPI ranks, the resulting computational area is 16x16 grid cells. To indicate that scaling performance tails off, the relative cost of computation and communication needs to be brought up. "The boundary tiles in each processor are 25% of the total, and the parallel communication cost increases significantly." For the atmosphere model, depending on the communication stencil, between approximately 20 and 60% of the computational area could be communicated. But until we know the relative cost of computations and communications, we are left with "cost increases significantly". There is nothing actionable for a user in that statement. Worse, users are left with the impression that after 256 MPI ranks, the communication costs increase significantly for all model configurations.*

**Reply 15:** The aim of the parallel efficiency test is to demonstrate that the ESMF coupler interface does not slow down the simulations. We agree with the reviewer that we did not show the

relative cost of computation and communication in our manuscript. We have revised to focus on that aspect of the code.

**Comment 16:** *When assigning relative costs between the atmosphere and ocean model, the most important factor of the ratio of the number of computational cells between the two models is ignored and "more complex physics parameterization packages" is offered. A clear and accurate representation is the relative cost of a single-column of the atmosphere model compared to a single column of the ocean model, for a single time step. This permits a user to assign MPI ranks to different model components.*

**Reply 16:** We thank the reviewer for pointing out the importance of the number of grid columns between the ocean and atmosphere models. We now discuss this. Please refer to the changes in Section 5 and Table 3.

**Comment 17:** *A number of external sources (books and online) define a "mean deviation" to be the same as mean absolute deviation. Perhaps bias is a better term and seems to be what the authors are interested in (warmer, cooler, etc).*

**Reply 17:** We thank the reviewer for pointing out this. We have replaced 'mean deviation' using bias in the discussion of the results.

**Comment 18:** *Care has to be taken when using words that have a typical meaning in a field, but that meaning is not intended. 1. significant: "but the increase is not significant after that". 2. To an atmosphere model user, the term "micro-physics" refers to the bulk or bin parameterization schemes that deal with resolved scale moist processes. "WRF micro-physics models (e.g., land surface model, the PBL model)"*

**Reply 18:** Thanks. We have replaced the improper words in the manuscript. We have also gone through the entire manuscript to revise other improper words.

**Comment 19:** *Given that 30 days x 24 hr/day x 60 min/hr x 2 model time steps / min = 86k atmosphere model time steps (which each have an elapsed time reported individually), there should*

*be a set of either error bars or standard deviation on all of the reported timing values. Similar statistical information is missing from the differences of the physical fields.*

**Reply 19:**    Thanks. We have added the comparison of standard deviation in the T2 results of three major cities in Fig. 7. In three major cities, the observed T2 values are available to generate the mean value and standard deviation.

**Comment 20:**  *In the description of the atmosphere model, the option for the resolved-scale moist physics is not mentioned. The atmosphere model lid for the vertical coordinate is not provided. A single, deterministic simulation for a month should probably use spectral nudging in the WRF model to keep the large-scale atmospheric flow in check. An atmospheric modeler would find the model setup section incomplete.*

**Reply 20:**    We thank the reviewer for pointing out that we did not provide adequate details regarding the atmospheric model setup. We have added the description of the moist physics scheme (Morrison 2-moment scheme) and the top atmosphere boundary condition ($P_{top}$ = 50 hPa). In our work, we don't use spectral nudging as we also want to test the model in a forecasting framework.

**Comment 21:**  *The discussion of the selection of the ocean model should have included the benefits and applicability near coastal areas, how the horizontal and vertical resolution are modified in shallow coastal areas, the impact of the broad shallow portions of the Red Sea on the vertical levels, spin-up time from initial conditions, the sensitivity of correctly choosing boundary locations, and coupling frequency.*

**Reply 21:**    MITgcm uses a finite volume grid and the vertical resolution is not modified in shallow coastal areas. HYCOM reanalysis data is prescribed as the initial condition and we did not apply a spin-up.

We tried different coupling frequency (2 min, 20 min, 180 min), but we did not see much difference in SST and T2. We selected 20 minutes because because it is adequate to capture the diurnal SST variation according to the reference (Seo et al., 2014).

**Comment 22:** *It is not a fair comparison to make when you keep the SST constant for a month: "Improvements of the coupled model over the stand-alone simulation with static SST forcing are observed in capturing the T2, heat fluxes, evaporation, and wind speed." Also, it was stated that the momentum fields were not impacted in your study: "This suggests that the ocean–atmosphere coupling does not significantly influence the wind field in the Red Sea region during the heat wave events."*

**Reply 22:** We thank the reviewer for pointing out this. We agree that it is not proper to simply say the coupling does not influence the wind field in a technical paper. We have removed this and rewritten the discussion of the surface wind.

**Comment 23:** *It is not conventional to have a statement as this in the conclusions: "On the other hand, the difference between coupled simulation and stand-alone simulations with updated forcings is also discussed."*

**Reply 23:** We thank the reviewer for pointing out this. We have removed this sentence. We have rewritten the conclusion section to focus more on the technical aspects of the work.

**Comment 24:** *When verifying an 8-km domain with 30-km gridded results, briefly describe the process.*

**Reply 24:** We converted the validation data on the lower resolution grid to the 0.08 ° model domain using 2D spline interpolation. We have added this to Section 4.

**Comment 25:** *Without some sort of statistical assistance, we do not know if -1.55 is mostly the same as or pretty different from -1.66. "The mean T2 differences over the sea are -1.55 (CPL), -1.66 (ATM.STA), and -1.7 (ATM.DYN) after 36 hours, and -0.99 (CPL), -1.10 (ATM.STA), and -1.12 (ATM.DYN) after 48 hours."*

**Reply 25:** We thank the reviewer for pointing out this. We have added the mean temperature and standard deviation of T2 from the ECMWF data. We use the mean and the standard deviation of T2 to show the difference between simulations is very small after 36 and 48 hours.

**Comment 26:** *Figure 1b has lots of arrows. Are they one-way only? If the parent talks to the child components directly, why is there a child coupler component?*

**Reply 26:** Yes. They are one-way only. The arrows are showing how the main function calls the parent component and then calls the child components. The coupler component handles the grid interpolation and data transfer between different models. We have added this in the manuscript.

**Comment 27:** *Most people would take Figure 1c to be an indication that modeling system is running concurrently.*

**Reply 27:** We thank the reviewer for pointing out this. We have re-plotted Figs. 1b and 1c in Fig. 2 to show the system components are running subsequently.

**Comment 28:** *Are there computational trade-offs for selecting a sequential rather than concurrent coupling mode? Does your implementation preclude selecting sequential vs concurrent as a build- or run-time option?*

**Reply 28:** The sequential mode is simple when dealing with the data transfer in ESMF, especially when each processor contains the ocean and atmosphere data for the same region (Collins et al., 2005). This makes it a natural starting point and it is chosen in our work. We have added this discussion in Section 2.4.

ESMF usually makes the sequential or concurrent mode as a build-time option. Our case is built only in the sequential mode.

**Comment 29:** *If a purely marine region was selected, is there an expectation that the cost of the atmosphere and ocean models would more equal?*

**Reply 29:** Yes, the cost of atmosphere and ocean models can be more equal if a purely marine region is selected in an ideal case. In the realistic Red Sea case, the ocean only covers 16% of the entire region. We thank the reviewer for pointing out this and we have revised the discussion on the ratio between ocean and atmosphere models:

The atmospheric model is much more time-consuming because it solves the entire computational domain, while the ocean model only solves the Red Sea (16% of the domain). The atmospheric model also uses a smaller time step (30 s) than that of the ocean model (120 s) and has more complex physics parameterization packages. If a purely marine region is selected in an ideal case, the cost of ocean and atmosphere models would be more equal. [R1]

**Comment 30:** *Do the atmosphere and ocean models run on the same processor set? If so, are the parallel tests hampered with fewer ocean points as the number of processors are increased?*

**Reply 30:** Yes, the atmosphere and ocean models run on the same processor set. We also agree that the parallel tests are hampered with very few ocean points when using 256 processors. However, the parallel efficiency of the coupled code is still good compared with that in the literature (Christidis, 2015; Zhang et al., 2013). We have revised our manuscript.

**Comment 31:** *After 1 day of simulation, why is the modeled SST so much colder than observed in Figure 7?*

**Reply 31:** The coupled simulations are all initialized using HYCOM data. The initial HYCOM SST is about 1 degree cooler than GHRSST observation data.

**Comment 32:** *There are several instances of trying to read too much into differences of the fields: "On the other hand, for the heat wave event on June 24th, CPL and ATM.DYN runs exhibit more latent heat fluxes coming out of the ocean (157 and 131 W/m2) than that in ATM.STA run (115 W/m2)." OK, yes, but if you look at Figure 9a, VIII vs IX vs X, IX and X are more similar than VIII and X.*

**Reply 32:** The mean difference in Fig. 10(IX) is -9.8 w/m2 and 5.9 w/m2 in Fig. 10(X), while the mean difference in Fig. 10(VIII) is 31.8 w/m2. In the coupled run, the sea surface is warmer and the latent heat flux is higher. We have added it to our revised manuscript.

**Comment 33:** *Technical Corrections*

*Figure 1 is pretty busy. 1a mentions "using bulk formulas" instead of listing the variables passed.*

**Reply 33:**    We thank the reviewer for pointing out this. In the present work, we use COARE 3.0 bulk algorithm to calculate the turbulent heat fluxes (Fairall et al., 2003). We have added this in Section 3. We have also added the list of variables passed in the test case in Section 3:

In the coupling process, the ocean model sends SST and ocean surface velocity to the coupler. They are used directly as the boundary conditions in the atmosphere model. The atmosphere model are sending the surface fields to the coupler: (1) net surface shortwave/longwave radiation, (2) latent/sensible heat; (3) 10-m wind speed, (4) net precipitation, (5) evaporation. The ocean model uses the atmosphere surface fields to compute the surface forcing: (1) total net surface heat flux, (2) surface wind stress, (3) freshwater flux. The total net surface heat flux is computed by adding latent heat flux, sensible heat flux, and net surface shortwave/longwave radiation fluxes. The surface wind stress is computed by using the 10-m wind speed (Large and Yeager, 2004). The freshwater flux is the difference between precipitation and evaporation. The latent sensible heat fluxes are computed by using COARE 3.0 bulk algorithm in WRF (Fairall et al., 2003). In the coupled code, different bulk formulae in WRF or MITgcm can also be used. [R1]

**Comment 34:**    *A number of figures would benefit from smaller color bar ranges. For example 10a has a range from about -500 W/m2 to 500 w/m2. The text says "However, a small improvement in the CPL (2.19 W/m2) and ATM.DYN (1.27 W/m2) runs can be observed in the longwave radiation on June 24th".*

**Reply 34:**    Thanks. We have re-plotted the color bar ranges to highlight the values in the figure. We agree that improvement is too small to be observed in the figure. The text on the difference between simulations is revised:

However, compared with ATM.STA run, there is a small improvement in the CPL (2.19 W/m$^2$) and ATM.DYN (1.27 W/m$^2$) runs on June 24th.[R1]

**Comment 35:**    *Figure 6 has diffs, diffs of diffs, rmse of diffs, and diffs of rmse of diffs. The y-axis labeling and the in-plot descriptions should be more precise.*

**Reply 35:**    Thanks. We have revised the descriptions in Fig. 6 to make the label and descriptions

more precise.

**Comment 36:** *Figure 5 would benefit from having some highlight that indicated the four heatwave periods.*

*In several places, "access" and "assess" are swapped.*

*Table 1 has ATM.STA twice. The second should be ATM.DYN.*

*Both are used "Arakawa-C grid" and "Arakawa C-grid".*

*There are some clumsy wordings "This run allows to access the WRF model", "which means that the SST in CPL run is tending to be similar to the realistic."*

*Figure 3 has a gray bar that covers the table bar. Figure 6 misspells deviation*

*Page 15, line 2, ATM.STA should be ATM.DYN?*

**Reply 36:** The authors thank the reviewer for pointing out these technical issues. We have corrected them in our manuscript. We have also gone through the manuscript and revised some other technical issues.

**References**

Christidis, Z., 2015. Performance and scaling of WRF on three different parallel supercomputers. In: International Conference on High Performance Computing. Springer, pp. 514–528.

Collins, N., Theurich, G., Deluca, C., Suarez, M., Trayanov, A., Balaji, V., Li, P., Yang, W., Hill, C., Da Silva, A., 2005. Design and implementation of components in the Earth System Modeling Framework. The International Journal of High Performance Computing Applications 19 (3), 341–350.

Fairall, C., Bradley, E. F., Hare, J., Grachev, A., Edson, J., 2003. Bulk parameterization of air–sea fluxes: Updates and verification for the COARE algorithm. Journal of climate 16 (4), 571–591.

Large, W. G., Yeager, S. G., 2004. Diurnal to decadal global forcing for ocean and sea-ice models:

the data sets and flux climatologies. Tech. rep., NCAR Technical Note: NCAR/TN-460+STR. CGD Division of the National Center for Atmospheric Research.

Seo, H., Subramanian, A. C., Miller, A. J., Cavanaugh, N. R., 2014. Coupled impacts of the diurnal cycle of sea surface temperature on the Madden–Julian oscillation. Journal of Climate 27 (22), 8422–8443.

Zhang, X., Huang, X.-Y., Pan, N., 2013. Development of the upgraded tangent linear and adjoint of the Weather Research and Forecasting (WRF) model. Journal of Atmospheric and Oceanic Technology 30 (6), 1180–1188.

---

## Author Comment (AC3) · 17 Apr 2019

**Reply to Reviewer Comment 2 (RC2)**

Rui Sun, Aneesh C. Subramanian, Arthur J. Miller, Matthew R. Mazloff,
Ibrahim Hoteit, Bruce D. Cornuelle

April 16, 2019

The authors would like to thank the reviewer for his/her comments, which have helped improve the quality of the manuscript. To adequately address the concerns raised by Reviewer 2 in the original manuscript, we have made the following changes:

1. We have augmented the text to include more technical details.

2. We have revised the discussion on the scalability in Section 5 according to the reviewer's comments.

3. We have added more discussion regarding the selection of boundary condition, projection, and coupling intervals.

Our detailed replies to specific comments of reviewer 2 are presented below. We have also attached an annotated manuscript to highlight the revisions.

**Comment 1:** *It is neither a technical nor a science paper. It would be beneficial to re-focus the manuscript on one aspect by clearly stating the problem, hypotheses and discuss the findings. Based on a few snippets of the manuscript it comes across that the authors are vaguely familiar with the foundations of numerical modeling in the atmosphere; they got two open source models, coupled them (no small feat!), and ran a test case. What is missing is a critical look at the approach, results, discussion of why things worked and more importantly, why not. I suggest to omit the whole section on scalability. The experiment design does not support any meaningful conclusions for scaling purposes. I would also recommend proof-reading (not spell checking!) the manuscript.*

**Reply 1:** The authors thank the reviewer for the general comment and we completely agree with reviewer. We have revised our manuscript to focus on the technical part of our coupled model and removed the scientific discussion in a few paragraphs. We have better motivated the manuscript in the introduction. We have added a paragraph in Section 3 to emphasize that the test case aims to show the ocean and atmosphere models are successfully coupled. We have also re-written Section 5 to emphasize the purpose of the scalability test.

We have proof-read the manuscript carefully.

**Comment 2:** *Page 6, lines 21-32: Too technical.*

**Reply 2:** We thank the reviewer for pointing out that the language used in the initial draft is too technical. We have revised the introduction of the ESMF/NUOPC coupler to make it more readable in section 2.4. Please refer to the revised manuscript.

**Comment 3:** *Page 6, line 32: Why was sequential mode selected?*

**Reply 3:** The sequential mode is simple when dealing with the data transfer in ESMF, especially when each processor contains the ocean and atmosphere data for the same region (Collins et al., 2005). This makes it a natural starting point and it is chosen in our work. We have added this discussion in Section 2.4.

We have also plotted Fig. 2 to show how the sequential mode is executed in the coupled model.

**Comment 4:** *Page 7, lines 17-31: Is it a 30-day long run? How frequently are you forcing lateral boundary conditions? What is the lateral boundary condition type? What is the projection? Why is coupling every 20 minutes? Why 8 km grid spacing?*

**Reply 4:** Yes, this is a 30-day long simulation which allows validation of the coupled model.

The ocean lateral boundary conditions are specified using HYCOM/NCODA global analysis data, and are updated every 24 hours. The atmosphere lateral boundary conditions are specified using ERA5 reanalysis and are updated every 6 hours. They are linearly interpolated between two time steps. We have highlighted the boundary conditions in Section 3 and Table 1. In MITgcm, a sponge

layer is applied at the lateral boundaries, with a thickness of 3 grid cells and inner/outer boundary relaxation timescales of 10/0.5 days. In WRF, the lateral boundary values are specified in WRF in the 'specified' zone, and the 'relaxation' zone is used to nudge the solution from the domain toward the boundary condition value. Here we used the default width of one point for the specific zone and four points for the relaxation zone. We have added these details in Section 3.

We used a lat-lon projection in both the ocean and atmosphere models. The grid spacing is $0.08^o$ and we have replaced 9km by using 'approximately 9km' or '$0.08^o$' in the manuscript.

The coupling interval is 20 minutes because it was deemed short enough to capture the resolved dynamics. It is 40 atmospheric time steps and 10 ocean time steps. 20 minutes is adequate to resolve the diurnal variation of SST and atmosphere forcing (Seo et al., 2014).

We have revised our manuscript and added the detailed discussions in Section 3.

**Comment 5:** *Page 8, lines 13, 16: 'accessing', 'accesses' should be 'assessing', 'assesses'*

*Page 9, Table 1: The second ATM.STA should probably be ATM.DYN.*

**Reply 5:** The authors thank the reviewer for pointing out this. We have fixed these typos in the manuscript.

**Comment 6:** *Page 9, line 9: Is MERRA-2 really an independent data compared to ERA5? The forecast model is, but the observations do overlap quite a bit.*

**Reply 6:** The authors agree that the observation data used in producing MERRA-2 and ERA5 overlap. However the reanalysis of MERRA-2 and ERA5 are performed independently. We choose MERRA-2 because it provides us with the latent heat and sensible heat fields. Hence, we rewrite the sentence as:

The MERRA-2 dataset is selected because it is an independent reanalysis data compared to the initial and boundary conditions used in the simulations. The dataset also provides a $0.625^o \times 0.5^o$ (lon $\times$ lat) resolution reanalysis fields of turbulent heat fluxes.[R2]

**Comment 7:** *Page 10, line 21-22: Why not use a nest with finer grid spacing to resolve the local*

*topography?*

**Reply 7:** The authors agree with the reviewer that using a finer grid spacing would better resolve the local topography and improve the forecast skill of model in the mountains. However, in our manuscript, we aim to develop a model to capture the ocean-atmosphere coupling in the Rea Sea. Therefore, we did not use a finer grid to resolve the local topography in the mountains. To give our manuscript a more technical focus, we have rewritten the paragraph and removed the discussion of the topography.

**Comment 8:** *Page 10, lines 10-32: When comparing to ERA5 data, how were the statistics computed? Was the model output interpolated onto the observation points in space and time?*

**Reply 8:** We interpolated ERA5 to our model as ERA5 data has lower grid resolution (30 km) than our coupled model (approximately 9 km), but omitted this detail in the original submission. We compared the results at the same time so that the results are not interpolated in time. These details have now been included in Section 3.

**Comment 9:** *Page 11, Figure 3: There was a gray stripe at the bottom, making it impossible to read labels of the color bars.*

**Reply 9:** Thanks. We have updated the figure and removed the gray stripe of this figure.

**Comment 10:** *Page 11, line 16: Land surface model and PBL model are not microphysics models.*

**Reply 10:** The authors thank the reviewer for pointing this out. The land surface model and PBL model are WRF *physics models*. We have fixed our mistakes on page 11 and other places.

**Comment 11:** *Page 13, Figure 5: Are there missing data points for the observed high and low T2, e.g. Mecca and Yanbu 6/21, Yanbu 6/8, 6/10, 6/14. . .?*

**Reply 11:** Yes, some T2 points are missing from NOAA NCDC data. We have added this in Fig. 7 of the revised manuscript.

**Comment 12:** *Page 14, Figure 6: Are model points interpolated to ERA5 points over the Red Sea? Which simulation is ATM.CPL, it has not been introduced in Table 1? How do you explain the drift (blue line)? How can RMSE be negative in the lower right figure?!*

**Reply 12:** We interpolated ERA5 data (30 km) to our coupled model (approximately 9 km). We have added the discussion on the interpolation method to Section 3.

'ATM.CPL' is a typo. It should be 'ATM.DYN'. Here we are comparing the ATM.DYN results to CPL and ATM.STA runs. We have fixed this in Table 1.

The blue line shows that the error in CPL run is much smaller than that in ATM.STA run (in Fig. 8 of the revised manuscript). This is because a fixed SST is used in ATM.STA run and the ocean response to the atmosphere is not represented.

In the lower right figure, the magnitude is showing the difference in the RSME from all simulations. We have updated the figure and labels to try to make this more clear.

**Comment 13:** *Page 14, line 4-6: Where there many clouds present during that period?*

*Page 16, line 13-14: Any cloud comparison?*

*Page 14, line 4-6, and line 12-13: First you state the forcing is different due to 'uncertainty in cloud modeling', then you state 'both simulations are driven by realistic atmospheric forcing'. Which one is correct? Please explain.*

**Reply 13:** We thank the author for pointing this out. We focus on validating our coupled model and we assessed the surface variables to demonstrate the ocean-atmosphere coupling. Hence we did not show the cloud data obtained from our model or observational data. We aim to keep a technical focus on the coupling and have removed our discussion on the cloud.

In the OCN.DYN run, the ocean is driven by ERA5 data; in the coupled run, the ocean is driven by the atmospheric fields obtained in the WRF simulation. We revised the sentence as:

Generally, the OCN.DYN and CPL runs have a similar range of error compared to both validation datasets, which shows the skill of the coupled model in simulating the ocean SST.[R2]

**Comment 14:** *Page 16, Figure 8: Why no time series comparison to MERRA-2 dataset?*

**Reply 14:** We have added two more figures and Fig. 8 in the initial draft is Fig. 10 now. In Fig. 10(a), we compared our results with HYCOM data. Since our coupled simulations are initialized using HYCOM, this aims to show the increase of the simulation error. In Fig. 10(b), we compared our data to the GHRSST satellite observation data to further validate the simulation results. The MERRA-2 reanalysis data is not used to validate the SST because the GHRSST observational product can be used.

Actually, MERRA-2 is used to validate some simulation results (e.g., latent heat, sensible heat) when the high-resolution observational products are not available. We have added the discussion on validation data to Section 3.

**Comment 15:** *Page 19, line 6: Which selected micro-physics schemes?*

**Reply 15:** We have replaced the original sentence using 'selected WRF physics options presented in Section 3'.

**Comment 16:** *Page 20, line 3: 64 /cm/year should read 64 cm/year?*

**Reply 16:** Thanks. We have fixed this typo.

**Comment 17:** *Page 21: line 8-9: What does it mean 'The decrease in parallel efficiency is because when using 256 processors, there are only 16x16 grid points in the horizontal plane'?*

*Page 21: line 11-13: Please elaborate: 'This may be attributed to the fluctuation of the CPU time when solving the systems of linear equations. When using different number of processors, the decomposition of the domain leads to different linear equation systems, requiring different CPU load and accordingly different convergence time.*

**Reply 17:** Thanks. We have used different number of processors to investigate the parallel efficiency of the coupled code. When using up to 128 CPUs, the parallel efficiency of the coupled code is close to linear. However, when using 256 processors, the parallel efficiency decreases to 70%.

We have re-written this paragraph:

It can be seen that the parallel efficiency is close to 100% when employing less than 128 processors and is still as high as 70% when using 256 processors. When using 256 processors, there are 20480 cells (16×16×80, 16 lat×16 lon×80 vertical levels) in each processor, but there are 5120 overlap cells (4×16×80, 4 sides×16 tiles per side×80 vertical levels), which is 25% of the total cells. From results reported in previous literature, the parallel efficiency of the coupled model is comparable to other ocean-alone or atmosphere-alone models when having similar number of grid points per tile (Marshall et al., 1997; Zhang et al., 2013). The decrease in parallel efficiency results from the increase of communication time, load imbalance, and I/O (read and write) operation per processor (Christidis, 2015). It is noted in Fig. 16 that the parallel efficiency fluctuates when using 8 to 32 processors. This may be because of the fluctuation of the communication time, load imbalance, and I/O operations. The fluctuation of the CPU time can also be seen in the speed-up curve, but at smaller magnitude.[R2]

**Comment 18:** *Page 21-23: Did you try weak or strong scaling? What is the communication cost? I/O cost? How many grid points per core are recommended? Are you reporting average times of multiple simulations in Table 3? How does WRF scale, MITgcm scale - do your results fit? Why did the coupling cost increase when using more cores?*

**Reply 18:** We tried strong scaling in our test. When presenting the scaling test our aim was to demonstrate that our implementation of the coupler does not slow down the individual simulations for varying core count. We have revised our discussion of the parallel performance in Section 5.

**Comment 19:** *Page 23, Table 3: Please use the experiment names consistently throughout the manuscript.*

**Reply 19:** Thanks. We have revised the experiment names in Table 3. We have also read through the manuscript ensuring that the experiment names are consistent.

**References**

Christidis, Z., 2015. Performance and scaling of WRF on three different parallel supercomputers. In: International Conference on High Performance Computing. Springer, pp. 514–528.

Collins, N., Theurich, G., Deluca, C., Suarez, M., Trayanov, A., Balaji, V., Li, P., Yang, W., Hill, C., Da Silva, A., 2005. Design and implementation of components in the Earth System Modeling Framework. The International Journal of High Performance Computing Applications 19 (3), 341–350.

Marshall, J., Adcroft, A., Hill, C., Perelman, L., Heisey, C., 1997. A finite-volume, incompressible Navier Stokes model for studies of the ocean on parallel computers. Journal of Geophysical Research: Oceans 102 (C3), 5753–5766.

Seo, H., Subramanian, A. C., Miller, A. J., Cavanaugh, N. R., 2014. Coupled impacts of the diurnal cycle of sea surface temperature on the Madden–Julian oscillation. Journal of Climate 27 (22), 8422–8443.

Zhang, X., Huang, X.-Y., Pan, N., 2013. Development of the upgraded tangent linear and adjoint of the Weather Research and Forecasting (WRF) model. Journal of Atmospheric and Oceanic Technology 30 (6), 1180–1188.

---

## Author Comment (AC4) · 17 Apr 2019

The comment was uploaded in the form of a supplement:
https://www.geosci-model-dev-discuss.net/gmd-2018-252/gmd-2018-252-AC4-supplement.pdf

---

## Author Comment (AC5) · 17 Apr 2019

The comment was uploaded in the form of a supplement:
https://www.geosci-model-dev-discuss.net/gmd-2018-252/gmd-2018-252-AC5-supplement.pdf

---

## Referee Report (RR1)

This is the second review of "SKRIPS v1.0: A regional coupled ocean–atmosphere modeling framework (MITgcm–WRF) using ESMF/NUOPC, description and preliminary results for the Red Sea", by Sun et al.

General Comments

The validation of a number of the fields is not convincing. What is an anticipated diurnal oscillation from a 30-km reanalysis vs a 9-km model output supposed to look like? What impact is the coarsened resolution of the ERA5 near the coast, where the surface air temperature differences could be larger than 20 C with a grid cell (desert vs ocean)? The only single figure with any sort of statistical inference possible shows that the coupled model performs no better than the test specifically expected to perform poorly.

In general, a comparison against a physically unrealistic month-long constant SST is problematic. Other than possibly a single mention of CPL vs ATM.STA at the beginning of the paper, most of the comparisons should be between CPL and ATM.DYN.

There are too many instances of the authors using "may", "perhaps", "hypothesize", etc. With a numerical model, all of these uncertain statements can be directly attributed.

There is no simply stated working definition of a heatwave. From the figures, a heatwave is not entirely obvious, so "capturing" a heatwave is quite subjective.

In several places the authors conclude a paragraph with a wrap up sentence to the effect that the CPL test performs well. In most of those examples of validations or comparisons, the other tests performed well also.

A couple of the scaling comments are incomplete, such as only talking about the total number of processes or the mention of overlap cells.

Several of the references to the figures refer to labels that do not exist. Some of the figure captions would be improved if they were more stand alone.

The paper would benefit from a good proofreading. There are misspellings, missing words, undefined terms, and a few unusual phrasings.

The authors do not make a strong case for their selection of this particular domain and the simulated event. This paper has as a focus a series of heatwave events where 84% of the domain is land (and desert). For the coastal temperature comparisons, there is no mention of possible sea breeze effects. This is not an ideal set up that benefits from coupled interactions between ocean and atmosphere.

Specific Comments

Page 4 line 2:

is shown in Fig. 1(a)

There is no (a) label

Page 5, figure 2
"PETs" is not defined

In panel (a)

There is no (a) label

In panel (b)

There is no (b) label

There is no explanation what the little boxes are.

Both ocean and atmosphere appear to happen at the same time. This is inconsistent with a sequential description.

Page 5, line 14
The surface boundary fields on the ocean surface is exchanged online

are

Page 6, line 4
but we updated it to couple

What is it

Page 6, line 12
In the present work, the Advanced Research WRF dynamic version (WRF-ARW, version 3.9.1.1) is used.

Include GitHub site for source code?

Page 6, line 32
In ESMF, 'timestamp' is a sequence of number,

numbers

Page 8, line 10

The time step for atmosphere simulation is 30 seconds.

For an approximately 9 km grid distance, 30 seconds seems overly conservative. Since WRF is the most expensive component, an increase to a 50 s timestep would be a substantial performance boost in overall timing. Is there a stability problem that is introduced with the coupling?

Page 8, line 21
net precipitation

Is this just accumulated precipitation minus evaporation? If so, just add a brief parenthetical.

Page 9, line 11-12

timescales of 10/0.5 days.

I am unfamiliar with what 10/0.5 days means.

Page 10, table 1

The four rows of the table should be more identifiable. There either needs to be more space between rows, or less vertical line space used in column three when the information extends to two lines.

The second ATM.STA should be ATM.DYN.

ERA5 is sufficient, without a description of bulk formula.

Page 10, line 10

validated against the ECMWF ERA5 dataset

Verifying a 9-km simulation with a 30-km reanalysis, specifically for cities that are along the coast may not be reasonable. It is not ever made clear how the max/min temperature comparisons are made. Does this field come out of ERA5? How is this information pulled from WRF?

Page 11, line 7-9

The simulation results obtained from coupled (CPL) run, the ERA5 data, and their associated difference are shown in Fig. 4 after 36 hours and 48 hours. It can be seen in Fig. 4(I) that the CPL run captures the heat wave event in the Red Sea region on June $2^{nd}$

If the simulated period started early (May 1, for example), is the June 2 heatwave event still present? Is this simply picked up because of the memory of the initial conditions? No where is it made clear what constitutes a heatwave event.

Page 11, line 19

all simulations can capture the T2 diurnal variation in the Red Sea region

Figure 4 shows that all simulations tend to have a larger T2 diurnal oscillation than the ERA5 reanalysis. This could be due to the cities are close enough to the coast that part of the 30-km grid cell contains moderating ocean temps, or that 30-km. Perhaps compare jobs to ERA5 (not necessarily to be shown in paper) to inform readers what is happening.

Page 12, figure 4

There is a systemic bias in ERA5: it is too cold at 1200 UCTC and too warm at 0000 UTC. Is this a good choice for validation?

There is little difference between the simulations. Most of the difference is between the simulated results vs ERA5. This is an indicator that this specific domain and this type of event may not be the best to showcase the capabilities of a coupled ocean atmosphere model.

Page 12, line 9

the SST in CPL run is tending to be similar to the realistic

This is not a clear way to state this point.

Page 12, line 13-14

It can be seen that four major heat waves (i.e., June 2$^{nd}$, 10$^{th}$, 17$^{th}$, and 24$^{th}$) and the T2 variations during the 30-day simulation are all captured

What is a heatwave event, and what defines captured?

Page 13, figure 5

This is more indication that this domain and case are not well chosen. Even after 23 days of constant SST, which would produce a poor representation of reality, the pattens displayed show virtually no difference over the land between CPL, ATM.STA, and ATM.DYN. Even with a bad SST, there is no discernible impact over land. A case needs to be chosen where a coupled ocean atmosphere model is relevant: flooding, precipitation, sea breezes, tides, inundation, typhoon cold wakes, steering flows for larger storms, etc.

Page 13, line 4

ERA5 uses a lower resolution grid and is unable to capture the T2 in the coastal city

Is this statement associated only with Yanuba, only with coastal cities, only when there is a large land-water temperature contrast? The authors should be careful about sweeping statements concerning ERA5.

Page 13, line 7-8

This may be due to the errors in initial conditions, or WRF physics schemes (e.g., land surface model, the PBL model) are unable to parameterize this extreme event.

A few lines up was "probably", now we have "may". These are model simulations, so you can determine cause. If the authors think the difference is due to initial conditions, back up the starting period by a few weeks. If the authors think the difference is due to physics, then try different combinations. Is there a specific assumption made in the chosen schemes? State a cause and defend the statement.

Page 13, line 9-10

In Fig. 6, the CPL run can better reproduce the evolution of the T2 compare to

ATM.STA run during the 30-day simulation:

Yes, CPL is better than ATM.STA, but that is a low bar. It is unphysical to have a constant SST for 30 days. Other than a single mention early in the paper concerning ATM.STA, the comparison should always be CPL vs ATM.DYN.

Page 13, line 12-14

We hypothesize that Mecca is much further away from the Red Sea than Yanbu and Jeddah, which indicates that the influence of air–sea coupling is strong near the coast.

First, Mecca is farther from the sea than the other two cities, that is not a hypothesis. What likely was intended was a hypothesis about the influence of air-sea coupling. OK, you have a stated conjecture, now conduct a test to support your position. Do you see diurnally varying on-shore/off-shore winds, for example? Is that signature missing farther inland? Is there flow that is blocked with a mountain range? If you remove the mountains, does the impact go further inland, etc.

Page 14, figure 6

It is difficult to identify the four heatwave events on the figure. Some background shading might be a good idea.

Choosing > 41 C in Jeddah seems appropriate, but you need > 45 C for Mecca and Yanbu. A heatwave event definition is required.

Is Yanbu too close to the water for the 30-km ERA5 reanalysis?

How are WRF and ERA5 daily max/min temperatures selected?

Page 15, figure 7

There is no statistical difference between the full coupled model and the ATM.STA (the experiment with a bad SST). More indication that this domain and case are insufficient to identify components of a functioning coupled system.

Page 15, figure 8

For the two top panels, some statistical information would be nice to allow the readers

to evaluate the CPL vs ATM.DYN vs ATM.STA tests.

Page 16, line 3-5

The daily SST fields from CPL run on June $2^{nd}$ and $24^{th}$ are shown in Fig. 9(I) and Fig. 9(VI). To validate the CPL run results, the SST fields obtained in OCN.DYN runs are shown in Fig. 9(II) and 9(VII) and the GHRSST fields are shown in Fig. 9(III) and 9(VIII).

Provide a brief explanation of what time was selected to verify with daily SST. Was the model SST averaged for a day? If there is no impact, just briefly state that.

Page 17, figure 10

Looking at the first week: explain (a) mean bias oscillation, and (a) RMS error ramp up.

Page 18, line 11

the air–sea interactions do not significantly impact the solar radiation

This comparison with observations also shows no impact with a coupled model.

Page 18, line 15-16

simulations over-estimated the total downward heat fluxes (CPL: 646 W/m$^2$; ATM.STA: 674 W/m$^2$; ATM.DYN: 663 W/m$^2$) for both heat wave events compared with MERRA-2 dataset (495 W/m$^2$)

This shows that the simulations (coupled vs non-coupled) are more similar to each other than to the verification. For this domain and case, this is more indication that the coupled model is not required. Or, if the uncoupled models are missing some important air-sea interactions, then the coupled model is not able to demonstrate the ability to capture those interactions either.

Page 18, 21-22

the present CPL simulations are capable of well capturing all the components of the surface heat fluxes during the heat wave events

You just showed that all cases are similar to each other, so all models are capturing components. While your statement is factually correct, it lends readers to believe that CPL is an improvement.

Page 18, line 28-30

On June 2nd, high-speed wind is observed in the northern and central Red Sea, and the CPL run successfully captures the small-scale features of wind speed patterns

From figure 14, all three shown tests (CPL, ATM.STA, ATM.DYN) are virtually identical, for both the June 2 and June 24 cases. This is disingenuous of the authors to imply that only CPL captures the small-scale features.

Page 18, line 31-32

the SST in the ATM.STA run is lower than the CPL run

Maybe it is more intuitive to say warmer or cooler when referring to temperatures.

Page18, line 34

The CPL run is able to capture

Figure 15 shows for the June 2 case, all tests perform well (since it is close to the SST initialization). For the June 24 case, the coupled model is similar to the ATM.DYN. This is and should be the important point that is made.

Page 19, figure 11 caption (same with page 20, figure 12 caption)

Only the heat fluxes over the sea is shown

are

Page 21, line 1-2

all simulation results are consistent on June 2$^{nd}$ because they are driven by the same initial condition

Not exactly, but after only a single simulated forecast day, the SSTs for all of the tests are similar.

Page 21, line 5-6

This is because the CPL run over-estimated the SST than the ATM.DYN run

Missing a word somewhere.

Page 22, line 6

runing 6-hour simulations

spelling

Page 22, line 8-9

When using 256 processors, there are 20480 cells (16 lat×16 lon×80 vertical levels) in each processor

Most of the grid points for the ocean are masked out, correct? Is this still an accurate statement of the amount of work on each MPI rank?

Page 22, line 9

there are 5120 overlap cells

What is an overlap cell? That term is not used in the atmosphere model. Is an overlap cell used in the ocean model or the coupler? How does it impact performance?

Page 22, line 13-14

It is noted in Fig. 16 that the parallel efficiency fluctuates when using 8 to 32 processors.

When describing the machine, include how many processes are used per node. Only comparisons with full nodes should be used for a good comprehension of scalability.

Page 22, line 14-15

This may be because of the fluctuation of the communication time, load imbalance, and I/O operations.

Here's that indefinite "may" again. To test I/O: turn off the output and only start timing after the input is complete. To test load imbalance, choose a domain that is not 84% land, etc. To test fluctuating communication time, are the timing results reproducible. There is no need to be uncertain.

Page 23, line 4-5

The atmospheric model also uses a smaller time step (30 s) than that of the ocean model (120 s) and has more complex physics parameterization packages

Within the atmosphere model, you could legitimately compare the complexity of one scheme vs another, but probably not between the ocean model vs the atmosphere model. The time step comparison, the relative number of solved grid cells, and the cost per grid cell from ATM.DYN vs OCN.DYN would be sufficient.

Page 23, line 9

We hypothesis that the cost of the ESMF/NUOPC coupler is communication cost and it becomes important as the amount of computation work is reduced with the number of grid cells in these strong scaling tests.

We hypothesize

This is a poor attempt to describe what is happening with strong scaling.

Also, "the cost … is communication cost" should be cleaned up.

Page 24, figure 16 caption

The simulation with the smallest case is regarded as base case when computing the speed-up

What does the smallest case mean? For strong scaling, the cases are the same size.

Page 25, line 6

The development activities has been focused on providing

have

Page 25, line 14-15

Improvements of the coupled model over the stand-alone simulation with static SST forcing are observed in capturing the T2, heat fluxes, evaporation, and wind speed.

This should focus on CPL vs ATM.DYN, not CPL vs ATM.STA. Also, quite a number of the authors' validations showed that even CPL vs ATM.STA was reasonable. This points to a problem in the fundamentals of the setup. If a month-long fixed SST is giving indistinguishable results to the new coupled model, at best the coupled model is doing no worse than the poorly constructed comparison case. An appropriate domain and case need to be selected.

Page 25, line 17-18

The coupled model scales linearly for up to 128 CPUs and the parallel efficiency remains about 70% for 256 processors.

This is a meaningless statement without a total number of horizontal grid points included, without the relative number of ocean vs atmosphere grid cells, without the cost per grid cell in each model, etc.

Page 25, line 19-20

Hence the coupled model can be applied for high-resolution coupled regional modeling studies on massively parallel processing supercomputers.

High-resolution has nothing to do with the number of grid cells. Arguably, 9 km is not

actually high-resolution.

No one will associate 256 MPI processes with massively parallel.

Page 25, line 22-23

These preliminary results motivate further studies in evaluating and improving this new regional coupled ocean–atmosphere model for investigating dynamical processes and forecasting applications in regions around the globe where ocean–atmosphere coupling is important.

That is what *this* paper should be about. There is no way to evaluate this new coupled model if the model is not used over a domain and for a case that would require coupled ocean and atmosphere capabilities.

---

## Referee Report (RR2)

This is a third review of "SKRIPS v1.0: A regional coupled ocean–atmosphere modeling framework (MITgcm–WRF) using ESMF/NUOPC, description and preliminary results for the Red Sea" by Sun et al.

Accept with technical corrections.

This version of the manuscript is more narrowly focused on the implementation and development of the coupled system, with much less emphasis on the validation of a case study. There are a number of locations with improved explanations / descriptions: how the comparison with observations takes place, the purpose for the domain and case study, use and impacts of the validating data, implications of observations and reanalysis data near the coast, and a clearer presentation on scaling and timing performance. Every figure caption has been modified, making them much easier to understand and making them much more standalone.

1) Scientific significance
Does the manuscript represent a substantial contribution to modelling science within the scope of this journal (substantial new concepts, ideas, or methods)?

This is a coupled ocean-atmosphere system, using a state-of-the-art coupler and highly cited numerical models. Significant documentation and support for the individual components exist. This coupled system could substantially lower the bar for users to get into the coupled ocean-atmosphere arena, particularly with users who are already fluent in one of the models.

2) Scientific quality
Are the scientific approach and applied methods valid? Are the results discussed in an appropriate and balanced way (consideration of related work, including appropriate references)? Do the models, technical advances and/or experiments described have the potential to perform calculations leading to significant scientific results?

The reasonable approach to the coupled system would be to test each component separately and then together to determine sensitivity. The authors approached validation in this fashion. They used standard observation and reanalysis to indicate that the coupled system is producing expected results. The test case was long enough to have indicated an incorrect trend. The spatial distribution of errors is explainable and largely expected. Regional climate systems are hampered without access to a coupled ocean-atmosphere system, and this coupled system is directly intended to support that community.

3) Scientific reproducibility
To what extent is the modelling science reproducible? Is the description sufficiently complete and precise to allow reproduction of the science by fellow scientists (traceability of results)?

The authors provided extensive information on the location of component source code repositories, and also to the source of the modified code (to enable the coupled system with existing source code). For the actually running of the system, I think the user community would

benefit from more detailed information than what the authors provide: https://scripps-coupled-atmosphere-ocean-model.readthedocs.io/en/latest/tutorial/tutorial_rs.html. This page (these pages) could serve as both a tutorial and a step-by-step procedure to replicate the coupled system results. For example, instead of "generate the mesh", explicit steps could be given.

4) Presentation quality
Are the scientific results and conclusions presented in a clear, concise, and well structured way (number and quality of figures/tables, appropriate use of English language)?

The paper is reasonably well structured. The conclusions draw correctly from the results and discussion. Data that that should be presented in a tabular form for ease of understanding is done so. Several figures suffer from insufficient visual differentiation, but that is similar to most presentations with even a small ensemble. However, as the authors state in comments (not in the paper) "our goal for this work and manuscript is to (1) introduce the design of a newly developed regional coupled ocean-atmosphere modeling system with a state-of-the-art coupler (ESMF with NUOPC), (2) describe the implementation of the modern coupling framework, (3) validate the coupled model using a real-world example, and (4) demonstrate and discuss the parallelization of the coupled model." The "similarness" of a number of figure components points to the authors conclusions that the coupled system is giving results similar to the results of the standalone well-known models.

Technical Corrections:

P1 L12,13
"The coupled model, documentation, and tutorial cases used in this work are available"
What tutorial cases are used in this paper? Either be more specific, or remove this portion of the sentence.

P3 L7
"(Skamarock et al., 2005)."
(Skamarock et al., 2019).

P5 L24
"(Skamarock et al., 2005)"
(Skamarock et al., 2019)

P9 L4,5
"To validate the coupled model, the following sets of simulations using different surface forcings are performed according to the tests (Warner et al., 2010; Turuncoglu et al., 2013; Ricchi et al., 2016)"
Either I don't know what "according to the tests" means, or this should be stated in a clearer fashion. How are these citations related to "according to the tests"?

P10 L1,2
"The ocean model uses the data assimilating HYCOM/NCODA 1/12 global reanalysis data as initial and boundary conditions"
Remove "data assimilating".

P10 L10,11
"Here we used the default width"
Everything else is present tense.

P10 L21
"The simulated SST is also validated against HYCOM/NCODA data to show the increase of the error."
No idea what this means.

P11 L1
http://cdo.ncdc.noaa.gov/CDO/georegion
This link is broken.

P11 L5
"Surface heat fluxes (e.g., latent heat, sensible heat, longwave and shortwave radiations), which drives the oceanic component"
drive

P16 L2
"The SST field snapshots from CPL run on June 2nd and 24th are shown"
Remove "run". The word "run" could be used as a noun or a verb in the sentence, and the meaning of the sentence changes.

P16 L12,13
"To quantitatively compare the errors in SST, the time history of the SST in the simulations (i.e., OCN.DYN and CPL) and validation data (i.e., GHRSST and HYCOM/NCODA) are shown in Fig. 9."
Add a sentence describing the reason for the oscillatory 9a vs 9b (3h vs daily data?).

P17 Fig9b
"Mean SST: CPL 29.28; OCN.DYN 29.29; HYCOM 29.85"
"HYCOM" should be "GHRSST"

P23 L3
"The coupled model is validated by using a realistic application to simulate the heat events in the Red Sea region. "
This is the conclusion so be specific in case readers read only this and the abstract, "The coupled model is validated by using a realistic application to simulate heat events during June 2012 in the Red Sea region."

P23 L15,16
"The CPU time associated with different components of the coupled simulations is also presented, showing the ESMF/NUOPC driver is not a bottleneck in the computation. "
Remove the "is also presented" and state directly "The CPU time associated with different components of the coupled simulations shows the ESMF/NUOPC driver is not a bottleneck in our coupled implementation of SKRIPS. "

P24 L1,2
"Appendix A: Snapshots of Surface Heat Fluxes
The snapshots of the THFs in the simulations at 1200 UTC June 2nd and 24th are presented."
Would you redefine THF in the appendix or do something similar to changing the title of the section to "Snapshots of Surface Turbulent Heat Fluxes".

P24 L2
"The snapshots of the THFs in the simulations at 1200 UTC June 2nd and 24th are presented."
Maybe add "(Fig. A1)" at the end of the sentence. You don't mention A1 until you discuss A3, and then only tangentially.

P31 L33,34
"Skamarock, W. C., Klemp, J. B., Dudhia, J., Gill, D. O., Barker, D. M., Wang, W., and Powers, J. G.: A description of the advanced research
WRF version 2, Tech. rep., National Center For Atmospheric Research Boulder Co Mesoscale and Microscale Meteorology Div, 2005."
Skamarock, W. C., J. B. Klemp, J. Dudhia, D. O. Gill, Z. Liu, J. Berner, W. Wang, J. G. Powers, M. G. Duda, D. M. Barker, and X.-Y. Huang, 2019: A Description of the Advanced Research WRF Version 4. NCAR Tech. Note NCAR/TN-556+STR, 145 pp. doi:10.5065/1dfh-6p97

---

## Author Response (AR3)

**Reviewer comments Revised manuscript Replies to the reviewers**

**Reviewer 1:**

While a heatwave event is high impact, it is not the first general episodic type that comes to mind when specifically highlighting the capabilities of a coupled ocean-atmosphere model.

Reply: We thank the reviewer for the insightful comments. We agree with the reviewer that the simulation of the heatwave events is not an ideal case for testing a regional coupled model and its benefits. Yet our goal for this work and manuscript is to (1) introduce the design of a newly developed regional coupled ocean-atmosphere modeling system with a state-of-the-art coupler (ESMF with NUOPC), (2) describe the implementation of the modern coupling framework, (3) validate the coupled model using a real-world example, and (4) demonstrate and discuss the parallelization of the coupled model. This work is the first time WRF has been coupled to the MITgcm, and we are using this manuscript to present the new resource. We now highlight the repository for downloading this resource in the abstract. We chose the location because our funding from KAUST was obtained to develop and implement the model for the Red Sea region. We chose the heatwave event due to their societal importance. This test case is meant only as a proof-of-concept example to show that the regional coupled model is working as one would expect, even in an extreme weather example. We have edited the abstract and introduction of the manuscript to emphasize our motivation and goal further and to make it clear that our objective is not to test the forecast skill of the model in this work. We are doing follow-on work to test the model skill in various scenarios. Our focus here is why we have chosen to submit this manuscript to Geoscientific Model Development, which promotes publication of code development and technical aspects of geoscience research.

Demonstrating the impact of coupled models is a research topic that requires a significant effort into the future. We have tried to make the text clear that our focus here is solely on the technical development of a coupled ocean-atmosphere model, implemented using state-of-the-art components. We hope this addresses the concerns of the reviewer about the test case satisfactorily.

**General Comments**

The validation of a number of the fields is not convincing. What is an anticipated diurnal oscillation from a 30-km reanalysis vs a 9-km model output supposed to look like? What impact is the coarsened resolution of the ERA5 near the coast, where the surface air temperature differences could be larger than 20 C with a grid cell (desert vs ocean)? The only single figure with any sort of statistical inference possible shows that the coupled model performs no better than the test specifically expected to perform poorly.

Reply: Yes, the ERA5 has coarser resolution and the strong gradient from water to desert has been more carefully handled. The ERA5 reanalysis uses a fractional value between 0 and 1 as the land-sea mask. In the old manuscript, we used bilinear interpolation and under-estimated the daily-high temperature in Yanbu. We now consider the effect of the land-sea mask and updated our figures.

Regarding performance, we have removed all language implying our coupled model is superior to ERA5. In keeping with the manuscript focus, the language now reflects the fact that the coupled system we developed is performing as one would expect.

In general, a comparison against a physically unrealistic month-long constant SST is problematic. Other than possibly a single mention of CPL vs ATM.STA at the beginning of the paper, most of the comparisons should be between CPL and ATM.DYN.

Reply: Because we aim to report on the technical development of a new coupled model, we compare CPL, ATM.STA and ATM.DYN runs to show the SST and atmospheric boundary variables are reasonably updated in the coupled model. We have revised the manuscript to focus on the code development and technical aspects. In our work, we emulate previous studies which implemented regional coupled models with other components and use a stationary SST for testing the implementation of the model [Loglisci et al. 2004, Warner et al. 2010]. Differences from ATM.STA gives a measure of how fast the SST is evolving and the size of the evolved SST signal the coupled model is producing.

There are too many instances of the authors using "may", "perhaps", "hypothesize", etc. With a numerical model, all of these uncertain statements can be directly attributed.

**Reply: We have edited the manuscript and replaced these terms by directly attributing the uncertainty in these statements. We have removed the other uncertain statements that we are not able to test by using our model.**

There is no simply stated working definition of a heatwave. From the figures, a heatwave is not entirely obvious, so "capturing" a heatwave is quite subjective.

Reply: We agree with the reviewer that this was not clear in our previous manuscript. There is no universal definition for heat waves, as this is a definition that varies from region to region. Hence, we have replaced the term 'heat waves' with 'heat events' in the manuscript as we are trying to simulate the events that had record high temperature in this region.

In several places the authors conclude a paragraph with a wrap up sentence to the effect that the CPL test performs well. In most of those examples of validations or comparisons, the other tests performed well also.

**Reply: We agree that ATM.STA and ATM.DYN tests also perform well. We aim to demonstrate the coupled model is capable of performing coupled simulations. The model tests are used to validate the coupled model is performing the coupled simulation as expected. We have revised the manuscript to emphasize our purpose.**

A couple of the scaling comments are incomplete, such as only talking about the total number of processes or the mention of overlap cells.

**Reply: We are grateful that the reviewers mentioned these incomplete comments. We have revised our manuscript and have gone through the scaling test to make sure we finished all our comments.**

Several of the references to the figures refer to labels that do not exist. Some of the figure captions would be improved if they were more stand alone.

**Reply: We have revised the labels of Figs. 1 and 2. We have also gone through the figure captions to improve them.**

The paper would benefit from a good proofreading. There are misspellings, missing words, undefined terms, and a few unusual phrasings.

**Reply: We thank the reviewer again for this. We have proofread the manuscript.**

The authors do not make a strong case for their selection of this particular domain and the simulated event. This paper has as a focus a series of heatwave events where 84% of the domain is land (and desert). For the coastal temperature comparisons, there is no mention of possible sea breeze effects. This is not an ideal set up that benefits from coupled interactions between ocean and atmosphere.

Reply: We select the case to illustrate our code development and validate our implementations on technical aspects, but not to provide an in-depth analysis of ocean-atmosphere coupling in the region. We have explained the choice to model this region and the extreme heat events in the revised manuscript:

We simulate a series of heat events in the Red Sea region, with a focus on validating and assessing the technical aspects of the coupled model. There is a desire for improved and extended forecasts in this region, and future work will investigate whether a coupled framework can advance this goal. The extreme heat events are chosen as a test case due to their societal importance. While these events and the analysis here doesn't highlight the value of coupled forecasting, these real-world events are adequate to demonstrate the performance and physical realism of the coupled model code implementation.

Specific Comments Page 4 line 2: is shown in Fig. 1(a) There is no (a) label

**Reply: We have now fixed this typo.**

Page 5, figure 2 "PETs" is not defined In panel (a) There is no (a) label In panel (b) There is no (b) label There is no explanation what the little boxes are. Both ocean and atmosphere appear to happen at the same time. This is inconsistent with a sequential description.

Reply: PETs are Persistent Executions Threads, which are single processing units (e.g., CPU, GPU) defined by ESMF. We defined them in the manuscript but not under the caption of Fig. 2. Now it is added. We have now added (a) and (b) labels to Fig. 2. In panel (b), each little box is the decomposed domain. We have added the explanation in the captions.

In Fig. 2(b), in the sequential mode, each PET works in a sequential way. The small arrows in the timeline show the sequential nature of the implementations. The coupled model first integrates the ocean component, then integrates the atmosphere component. After the integration is finished in one coupling step, the ESMF connector gathers the outputs from ocean/atmosphere components and update the boundary condition for the next step. We also implemented the concurrent mode and added it in Fig. 2(b):

Panel (b) and (c) shows the sequential and concurrent mode implemented in SKRIPS, respectively. PETs (Persistent Execution Threads) are single processing units (i.e., CPU, GPU) defined by ESMF. OCN, ATM, and CON denote oceanic component, atmospheric component and connector component, respectively. The blocks under PETs are the CPU cores in the simulation; the small blocks under OCN or ATM are the small sub-domains in each core; the block under CON is the coupler. The red arrows indicate the model components are sending data to the connector and the yellow arrows indicate the model components are reading data from the connector. The horizontal arrows indicate the time axis of each component and the ticks on the time axis indicate the coupling time step.

Page 5, line 14 The surface boundary fields on the ocean surface is exchanged online Are

Reply: We have now fixed this typo.

Page 6, line 4 but we updated it to couple What is it

Reply: We now replaced 'it' using 'the baseline coupler'.

Page 6, line 12 In the present work, the Advanced Research WRF dynamic version (WRF-ARW, version 3.9.1.1) is used. Include GitHub site for source code?

Reply: We now have added the Github site for WRF:

WRF is used extensively for operational forecasts (http://www.wrf-model.org/plots/wrfrealtime.php) as well as realistic and idealized dynamical studies. The WRF code and documentation are under continuous development on Github (https://github.com/wrf-model/WRF).

Page 6, line 32 In ESMF, 'timestamp' is a sequence of number, numbers Page 8, line 10

Reply: We have now fixed this typo.

**The time step for atmosphere simulation is 30 seconds.**

For an approximately 9 km grid distance, 30 seconds seems overly conservative. Since WRF is the most expensive component, an increase to a 50 s timestep would be a substantial performance boost in overall timing. Is there a stability problem that is introduced with the coupling?

Reply: We agree with the reviewer that increasing the time step can improve computational performance. However, the computational domain is complex in the Ethiopian Highlands and Hejaz Mountains (Saudi Arabia). [Hughal et al. 2017; WRF forum] We find that the CFL number is larger than 1 if we use a 50 second timestep. We have revised the manuscript:

The time step for atmosphere simulation is 30 seconds, which is chosen to avoid violation of the CFL condition.

[Hughal et al. 2017. Wind modelling, validation and sensitivity study using Weather Research and Forecasting model in complex terrain. Environmental modeling & software] [http://forum.wrfforum.com/viewtopic.php?f=8&t=357]

Page 8, line 21 **net precipitation** Is this just accumulated precipitation minus evaporation? If so, just add a brief parenthetical.

Reply: The MITgcm gets precipitation from WRF and uses the precipitation to calculate the freshwater flux (E-P). We have replaced `net precipitation' with `precipitation'.

Page 9, line 11-12 timescales of 10/0.5 days. I am unfamiliar with what 10/0.5 days means.

**Reply: What we meant to say is 'the inner and outer boundary relaxation timescales of the sponge layer are 10 and 0.5 days, respectively'. We have rewritten this sentence.**

Page 10, table 1

The four rows of the table should be more identifiable. There either needs to be more space between rows, or less vertical line space used in column three when the information extends to two lines. The second ATM.STA should be ATM.DYN.

ERA5 is sufficient, without a description of bulk formula.

**Reply: We have revised the tables and made the four rows more identifiable. We agree that ERA5 is sufficient and removed the description of bulk formula. We also fixed the typos.**

Page 10, line 10

**validated against the ECMWF ERA5 dataset**

Verifying a 9-km simulation with a 30-km reanalysis, specifically for cities that are along the coast may not be reasonable. It is not ever made clear how the max/min temperature comparisons are made. Does this field come out of ERA5? How is this information pulled from WRF?

Reply: We agree with the reviewer that verifying the surface air temperature using the 30-km ERA5 data may not be reasonable. Hence we used the NCDC ground observations to validate the air temperature. The T2 fields are interpolated to the NCDC stations for both WRF and ERA5. The NCDC station data are the best observations available in this region. Unfortunately the observation time in NCDC data is not available, hence the daily max/min T2 every 24 hours from the model simulations are compared with the daily max/min T2 from the NCDC data. We have added a discussion on this in the revised manuscript:

Since the 30-km resolution ERA5 dataset may not be adequate to capture the sharp T2 gradients near the coast, the T2 is also compared with the available ground observations from NOAA National Climate Data Center (NCDC climate data online at http://cdo.ncdc.noaa.gov/CDO/georegion). To evaluate the modeling of T2 in three major cities near the eastern shore of the Red Sea, the T2 fields from both the simulations and ERA5 are interpolated to the NCDC stations. The daily maximum and minimum temperatures are compared with those from NCDC data.

**Page 11, line 7-9**

The simulation results obtained from coupled (CPL) run, the ERA5 data, and their associated difference are shown in Fig. 4 after 36 hours and 48 hours. It can be seen in Fig. 4(I) that the CPL run captures the heat wave event in the Red Sea region on June 2nd

If the simulated period started early (May 1, for example), is the June 2 heatwave event still present? Is this simply picked up because of the memory of the initial conditions? No where is it made clear what constitutes a heatwave event.

Reply: This is a test validation case to show that the coupled model works as expected. We agree that it is not proper to claim that the CPL run captures the heat events without a detailed discussion on what constitutes the events. We have revised our manuscript to simply emphasize that the performance is reasonable for this high heat events.

Page 11, line 19

all simulations can capture the T2 diurnal variation in the Red Sea region

Figure 4 shows that all simulations tend to have a larger T2 diurnal oscillation than the ERA5 reanalysis. This could be due to the cities are close enough to the coast that part of the 30-km grid cell contains moderating ocean temps, or that 30-km. Perhaps compare jobs to ERA5 (not necessarily to be shown in paper) to inform readers what is happening.

Reply: We agree with the reviewer that the simulations have larger T2 diurnal oscillation than ERA5 and we have rewritten this sentence:

Fig. 4 also shows the diurnal variation of T2 in the Red Sea region, and the diurnal variation will be further discussed later in this section.

We also agree that because the cities are on the sea we need to account for the land masks when interpolating. We have updated Fig. 6 with a better accounting for this mask, and the daily max/min T2 in ERA5 now agrees better with the NCDC observations.

Page 12, figure 4

There is a systemic bias in ERA5: it is too cold at 1200 UCTC and too warm at 0000 UTC. Is this a good choice for validation?

Reply: We agree with the reviewer that ERA5 is colder at 1200 UTC and warmer at 0000 UTC on land compared with the simulations. The daily max/min T2 in ERA5 agrees better with the NCDC observations compared to all simulations. Hence we use ERA5 to validate the T2 fields obtained in the simulations with the aim being to demonstrate that the coupled model is capable of reasonable simulations of the coupled ocean-atmosphere system. We have revised our discussion in the manuscript:

Since ERA5 air temperatures are in good agreement with the NCDC ground observations in the Red Sea region (detailed comparison of all stations are not shown), we use ERA5 data to validate the simulation results...... To validate the coupled ocean-atmosphere model, the mean T2 differences over the sea in the simulations are compared with the ERA5 data. The mean T2 biases and RMSEs over the sea are shown in Table 3. The biases of the T2 are comparable with the biases reported in other WRF simulations for heat events [Imran et al, 2018].

There is little difference between the simulations. Most of the difference is between the simulated results vs ERA5. This is an indicator that this specific domain and this type of event may not be the best to showcase the capabilities of a coupled ocean atmosphere model.

Reply: We agree with the reviewer that the differences in the simulations are subtle. Fig. 4 shows our model simulation does not have much error compared with ERA5 and benchmark WRF simulations (the bias and RMSE of T2 in the present work are similar to those in the benchmark WRF-ARW simulations [Xu et al. 2009, Zhang et al. 2013, Imran et al. 2018]). In Fig5, T2 differs very little on land, but the T2 in the CPL run is about 1.0 degC warmer than the ATM.STA run and has smaller error than ATM.STA run. This shows the coupled model driven by a warming SST can better capture the surface T2. We also show that the coupled run can be as good as uncoupled run using an updated SST. We have revised our manuscript:

It can be seen in the figure that the T2 errors on land are consistent for all three simulations. However, the T2 over the sea in CPL simulation has smaller mean biases with the validation ERA5 data (10th: -1.24 degC; 24th: -0.81 degC) compared with the ATM.STA run (10th: -1.56 degC; 24th: -1.83 degC)...... The T2 over the water in the CPL run is closer to the ERA5 because MITgcm in the coupled model provides updated warming SST, which warms the T2; the ATM.STA run uses a constant cooler SST from June 1st, and thus the T2 is determined by the constant cooler SST. On the other hand, when comparing the CPL run with the ATM.DYN run, the mean difference is smaller (10th: +0.04 degC; 24th: -0.62 degC). This shows the CPL run is comparable to the ATM.DYN run driven by an updated warming SST.

Page 12, line 9 the SST in CPL run is tending to be similar to the realistic This is not a clear way to state this point.

Reply: We agree with the reviewer that our statement is not clear in the old manuscript. We have removed this sentence in the manuscript.

Page 12, line 13-14 It can be seen that four major heat waves (i.e., June 2nd, 10th, 17th, and 24th) and the T2 variations during the 30-day simulation are all captured What is a heatwave event, and what defines captured?

Reply: Since there is no universal definition on the heat waves, we have replaced heat waves using heat events in the manuscript. We have also revised 'capture the heat wave events' by showing the difference of the maximum daily temperature between difference simulations.

Page 13, figure 5

This is more indication that this domain and case are not well chosen. Even after 23 days of constant SST, which would produce a poor representation of reality, the patterns displayed show virtually no difference over the land between CPL, ATM.STA, and ATM.DYN. Even with a bad SST, there is no discernible impact over land. A case needs to be chosen where a coupled ocean atmosphere model is relevant: flooding, precipitation, sea breezes, tides, inundation, typhoon cold wakes, steering flows for larger storms, etc.

Reply: We agree with the reviewer that after 23 days of constant SST, the patterns on land does not show significant differences. However, the ocean T2 differs by about 1 degC in Fig. 5(IX), showing the impact of SST. In our paper, we focus on the technical issues and the development of the coupled

model and validate that is performing in a reasonable way. This is why we have chosen to submit this manuscript to Geoscientific Model Development, which considers the publication of code development and technical aspects of research.

**Page 13, line 4**

ERA5 uses a lower resolution grid and is unable to capture the T2 in the coastal city

Is this statement associated only with Yanbu, only with coastal cities, only when there is a large land-water temperature contrast? The authors should be careful about sweeping statements concerning ERA5.

Reply: The authors thank the reviewer for pointing out this. We agree that our statement concerning ERA5 in this sentence is inaccurate and too general. We have revised our interpolation method near Yanbu by adding the land-sea mask. We have removed this sentence in the revised manuscript.

**Page 13, line 7-8**

This may be due to the errors in initial conditions, or WRF physics schemes (e.g., land surface model, the PBL model) are unable to parameterize this extreme event.

A few lines up was "probably", now we have "may". These are model simulations, so you can determine the cause. If the authors think the difference is due to initial conditions, back up the starting period by a few weeks. If the authors think the difference is due to physics, then try different combinations. Is there a specific assumption made in the chosen schemes? State a cause and defend the statement.

Reply: Yes, we have tried a few different WRF physics options and different combinations (e.g., YSU/MYJ/MYNN + Kain-Fritsch/Zhang-McFarline + Noah LSM/Noah-MP LSM/RUC + MM5/MYNN). We tried different combinations to help us model the diurnal cycle. In our test, only Kain-Fritsch+RUC can capture the diurnal cycles in the Red Sea region. Other combinations will overestimate the daily max T2 and underestimate the daily min T2 on land. Since our manuscript focuses on the development of the model, we have removed the discussion of the errors in the manuscript.

**Page 13, line 9-10**

**In Fig. 6, the CPL run can better reproduce the evolution of the T2 compared to ATM.STA run during the 30-day simulation:**

Yes, CPL is better than ATM.STA, but that is a low bar. It is unphysical to have a constant SST for 30 days. Other than a single mention early in the paper concerning ATM.STA, the comparison should always be CPL vs ATM.DYN.

Reply: We agree with the reviewer that it is unphysical to have a constant SST for 30 days. But we are using both ATM.STA and ATM.DYN cases to validate the coupled model, as done in previous studies [Loglisci et al. 2004, Warner et al. 2010]. However, since the difference is insignificant we have removed this discussion.

**Page 13, line 12-14**

**We hypothesize that Mecca is much further away from the Red Sea than Yanbu and Jeddah, which indicates that the influence of air-sea coupling is strong near the coast.**

First, Mecca is farther from the sea than the other two cities, that is not a hypothesis. What likely was intended was a hypothesis about the influence of air-sea coupling. OK, you have a stated conjecture, now conduct a test to support your position. Do you see diurnally varying on-shore/off-shore winds, for example? Is that signature missing farther inland? Is there flow that is blocked with a mountain range? If you remove the mountains, does the impact go further inland, etc.

Reply: The authors thank the reviewer for pointing out this. There is no mountain between Mecca and the sea. We aim to present the development of the coupled model and we have removed the discussion of this in our manuscript.

Page 14, figure 6

It is difficult to identify the four heatwave events on the figure. Some background shading might be a good idea.

Choosing > 41 C in Jeddah seems appropriate, but you need > 45 C for Mecca and Yanbu. A heatwave event definition is required.

Reply: The authors thank the reviewer for pointing out this. There is no universal definition of heat wave events. We have replaced 'heat wave' using 'heat events' in the manuscript.

Is Yanbu too close to the water for the 30-km ERA5 reanalysis? How are WRF and ERA5 daily max/min temperatures selected?

Reply: Yes, the ERA5 resolution may not be able to resolve the strong water to desert transition. The ERA5 reanalysis uses a fractional value between 0 and 1 as the land-sea mask. In the old manuscript, we used bilinear interpolation and under-estimated the daily-high temperature in Yanbu. We now consider the effect of the land-sea mask and updated our figures. The WRF/ERA5 daily max/min temperatures are the max/min values every 24 hours. We give the method specifics in the revised manuscript.

Page 15, figure 7

There is no statistical difference between the full coupled model and the ATM.STA (the experiment with a bad SST). More indication that this domain and case are insufficient to identify components of a functioning coupled system.

Reply: The coupled model better forecasts the daily low temperature in Jeddah and Yanbu by about 1.0 and 0.5 degC from day 20 to day 30, and we added the discussion on this. We agree with the reviewer that showing the mean value does not show any significant improvement and we plotted the RMSE values instead in this figure. We have added the statistic information in Figure 6.

Page 15, figure 8

For the two top panels, some statistical information would be nice to allow the readers to evaluate the CPL vs ATM.DYN vs ATM.STA tests.

Reply: We have added the statistical information in Fig. 8. We have also added the information for other figures.

Page 16, line 3-5

The daily SST fields from CPL run on June 2nd and 24th are shown in Fig. 9(I) and Fig. 9(VI). To validate the CPL run results, the SST fields obtained in OCN.DYN runs are shown in Fig. 9(II) and 9(VII) and the GHRSST fields are shown in Fig. 9(III) and 9(VIII).

Provide a brief explanation of what time was selected to verify with daily SST. Was the model SST averaged for a day? If there is no impact, just briefly state that.

Reply: GHRSST uses the nighttime SST and we also use the nighttime SST at 0000 UTC (about 3 A.M local time) to compare with the GHRSST. The snapshots of the SST obtained in the CPL run are also compared with the available HYCOM data.

Page 17, figure 10 Looking at the first week: explain (a) mean bias oscillation, and (a) RMS error ramp up.

Reply: We performed new simulations using 3-hourly HYCOM boundary conditions. When using this 3-hourly data the mean bias oscillation is not observed in the first week. The initial SST error is zero because both simulations are initialized using HYCOM in Fig. 10(a), but the HYCOM dataset is corrected by observations so that the error increases. We have added the discussion on RMSE increase in the manuscript.

Page 18, line 11 the air-sea interactions do not significantly impact the solar radiation This comparison with observations also shows no impact with a coupled model.

Reply: We agree with the reviewer that the impact of the coupled model is small. We have added this in our manuscript.

**Page 18, line 15-16**

simulations over-estimated the total downward heat fluxes (CPL: 646 W/m2 ; ATM.STA: 674 W/m2 ; ATM.DYN: 663 W/m2 ) for both heat wave events compared with MERRA-2 dataset (495 W/m2) This shows that the simulations (coupled vs non-coupled) are more similar to each other than to the verification. For this domain and case, this is more indication that the coupled model is not required. Or, if the uncoupled models are missing some important air-sea interactions, then the coupled model is not able to demonstrate the ability to capture those interactions either.

Reply: We have attempted to better highlight differences. We have removed the snapshot comparisons (previous Fig. 12) and instead to better quantify the heat fluxes we calculate the time series of the mean deviation and RMSE between the model simulations and MERRA-II.

Page 18, 21-22

the present CPL simulations are capable of well capturing all the components of the surface heat fluxes during the heat wave events

You just showed that all cases are similar to each other, so all models are capturing components. While your statement is factually correct, it leads readers to believe that CPL is an improvement.

Reply: We agree with the reviewer that our discussion on the surface heat flux ignored the uncoupled simulations. We have re-written this in the revised manuscript:

It can be seen in the figure that both the CPL and ATM.STA runs reproduce the mean heat flux over the Red Sea estimated by MERRA-2.

Page 18, line 28-30

On June 2nd, high-speed wind is observed in the northern and central Red Sea, and the CPL run successfully captures the small-scale features of wind speed patterns

From figure 14, all three shown tests (CPL, ATM.STA, ATM.DYN) are virtually identical, for both the June 2 and June 24 cases. This is disingenuous of the authors to imply that only CPL captures the small-scale features.

Reply: We agree with the reviewer that our discussion on U10 ignored the uncoupled simulations. We have re-written this in the manuscript:

**both the CPL and ATM.STA runs successfully capture the small-scale features of wind speed patterns**

Page 18, line 31-32 the SST in the ATM.STA run is lower than the CPL run Maybe it is more intuitive to say warmer or cooler when referring to temperatures.

Reply: We have replaced 'lower' using 'cooler' in this sentence. We have also replaced higher/lower using warmer/cooler when describing SST.

Page18, line 34

**The CPL run is able to capture**

Figure 15 shows for the June 2 case, all tests perform well (since it is close to the SST initialization). For the June 24 case, the coupled model is similar to the ATM.DYN. This is and should be the important point that is made.

Reply: We agree with the reviewer that we did not make the important points clear in the discussion of the evaporation. Because the evaporation is proportional to the latent heat flux, we have moved it to the appendix. But, we still revised the discussion on the evaporation.

Page 19, figure 11 caption (same with page 20, figure 12 caption) Only the heat fluxes over the sea is shown are

Reply: We have fixed this typo.

Page 21, line 1-2 all simulation results are consistent on June 2nd because they are driven by the same initial condition Not exactly, but after only a single simulated forecast day, the SSTs for all of the tests are similar.

Reply: We agree with the reviewer. We have replaced the old sentence with:

After 48-hours, the simulation results are close with each other (e.g., the RMSE between CPL and ATM.STA simulation is smaller than 10 cm/year).

Page 21, line 5-6 This is because the CPL run over-estimated the SST than the ATM.DYN run Missing a word somewhere.

Reply: We replaced the old sentence with 'This is because the SST in the CPL run is warmer than the ATM.DYN run'.

Page 22, line 6 runing 6-hour simulations spelling

Reply: We have now fixed this typo.

Page 22, line 8-9

When using 256 processors, there are 20480 cells (16 lat×16 lon×80 vertical levels) in each processor Most of the grid points for the ocean are masked out, correct? Is this still an accurate statement of the amount of work on each MPI rank?

Reply: We agree most of the ocean points are masked out. We have rewritten this sentence in the original manuscript:

When using 256 CPU cores, there are a maximum of 20480 cells (16 lat  $\times$  16 lon  $\times$  80 vertical levels) in each core. It is noted that the ocean model only solves the Red Sea (16% of the domain) and most of the ocean points are masked out in this real-world test.

Page 22, line 9

there are 5120 overlap cells

What is an overlap cell? That term is not used in the atmosphere model. Is an overlap cell used in the ocean model or the coupler? How does it impact performance?

Reply: The 'overlap cell' is not an appropriate expression. We have removed this sentence in our manuscript and rewritten it:

When using 256 CPU cores, there are a maximum of 20480 cells (16 lat x 16 lon x 80 vertical levels) in each core. It is noted that the ocean model only solves the Red Sea (16% of the domain) and most of the ocean points are masked out in this real-world test. From results reported in the literature, the parallel efficiency of the coupled model is comparable to other ocean-alone or atmosphere-alone models when having similar number of grid points per CPU core [Marshall 1997, Zhang 2013].

Page 22, line 13-14

It is noted in Fig. 16 that the parallel efficiency fluctuates when using 8 to 32 processors. When describing the machine, include how many processes are used per node. Only comparisons with full nodes should be used for a good comprehension of scalability.

Reply: We agree with the reviewer. We performed 'full node' test using another cluster (because our old cluster is down). We have added:

We started using  $N_p0 = 32$  because each compute node has 32 CPU cores.

Page 22, line 14-15

This may be because of the fluctuation of the communication time, load imbalance, and I/O operations.

Here's that indefinite "may" again. To test I/O: turn off the output and only start timing after the input is complete. To test load imbalance, choose a domain that is not 84% land, etc. To test fluctuating communication time, are the timing results reproducible. There is no need to be uncertain.

Reply: We agree with the reviewer. We did perform a cleaner test by turning off the unnecessary I/O. Nevertheless, the purpose of Section 5 is to show the coupler has good parallel efficiency and does not slow down the simulation. We have removed the discussion on the fluctuation of the parallel efficiency.

Page 23, line 4-5

**The atmospheric model also uses a smaller time step (30 s) than that of the ocean model (120 s) and has more complex physics parameterization packages**

Within the atmosphere model, you could legitimately compare the complexity of one scheme vs another, but probably not between the ocean model vs the atmosphere model. The time step comparison, the relative number of solved grid cells, and the cost per grid cell from ATM.DYN vs OCN.DYN would be sufficient.

**Reply: We have now removed the discussion on the parameterization schemes in our manuscript:**

The atmospheric model is much more time-consuming because it solves the entire computational domain, while the ocean model only solves the Red Sea (16% of the domain). The atmospheric model also uses a smaller time step (30~s) than that of the ocean model (120~s). If a purely marine region is selected in an ideal case, the cost of ocean and atmosphere models would be more equal compared with this Red Sea case.

**Page 23, line 9**

We hypothesis that the cost of the ESMF/NUOPC coupler is communication cost and it becomes important as the amount of computation work is reduced with the number of grid cells in these strong scaling tests.

**We hypothesize**

This is a poor attempt to describe what is happening with strong scaling. Also, "the cost ... is communication cost" should be cleaned up.

Reply: We thank the reviewer for pointing out this. We turned off unnecessary I/O in a clean test. The CPU time spent on the atmosphere model, ocean model, and the coupler are reported in the CPL run. We have replaced the old sentences with:

The coupling process takes less than 3% of the total costs in the CPL run. Although the proportion of the coupling process in the total costs will increase when using more CPU cores, the total time spent on the coupling process is similar. The CPU time spent on two uncoupled runs (i.e., ATM.STA, OCN.DYN) is also shown in Table. 3. Compared with the uncoupled simulations, the ESMF-MITgcm and ESMF-WRF interfaces do not increase the CPU time in the coupled simulation. In summary, the scalability test results suggest that the ESMF/NUOPC coupler is not a bottleneck for using SKRIPS in coupled regional modeling studies.

**Page 24, figure 16 caption**

The simulation with the smallest case is regarded as base case when computing the speed-up What does the smallest case mean? For strong scaling, the cases are the same size.

Reply: We have replaced 'the smallest case' using 'the simulation with 32 CPU cores'.

Page 25, line 6 The development activities has been focused on providing have

Reply: We have fixed this typo.

**Page 25, line 14-15**

Improvements of the coupled model over the stand-alone simulation with static SST forcing are observed in capturing the T2, heat fluxes, evaporation, and wind speed.

This should focus on CPL vs ATM.DYN, not CPL vs ATM.STA. Also, quite a number of the authors' validations showed that even CPL vs ATM.STA was reasonable. This points to a problem in the fundamentals of the setup. If a month-long fixed SST is giving indistinguishable results to the new coupled model, at best the coupled model is doing no worse than the poorly constructed comparison case. An appropriate domain and case need to be selected.

Reply: The purpose of this paper is to present the coupled modeling system and not to answer scientific questions on the impact of coupling. We have removed the discussion on the improvement of the coupled model in the conclusion.

Page 25, line 17-18

The coupled model scales linearly for up to 128 CPUs and the parallel efficiency remains about 70% for 256 processors.

This is a meaningless statement without a total number of horizontal grid points included, without the relative number of ocean vs atmosphere grid cells, without the cost per grid cell in each model, etc.

Reply: We thank the reviewer for pointing out this. We have rewritten this sentence:

The parallel efficiency of the coupled model is consistent with that of the stand-alone ocean and atmosphere models when using various numbers of CPU cores in the test.

Page 25, line 19-20

Hence the coupled model can be applied for high-resolution coupled regional modeling studies on massively parallel processing supercomputers.

High-resolution has nothing to do with the number of grid cells. Arguably, 9 km is not actually high-resolution.

No one will associate 256 MPI processes with massively parallel.

Reply: We agree with the reviewer that high-resolution and massively parallel are not used appropriately in the conclusion. We have removed 'high-resolution' and 'massively parallel' in the conclusion section and rewritten this sentence:

The CPU time associated with different parts of the coupled simulations is also presented, suggesting the ESMF/NUOPC driver is not the bottleneck of the computation. Hence the coupled model can be implemented for coupled regional modeling studies on supercomputers with comparable performance as that attained by uncoupled stand-alone models.

**Page 25, line 22-23**

These preliminary results motivate further studies in evaluating and improving this new regional coupled ocean-atmosphere model for investigating dynamical processes and forecasting applications in regions around the globe where ocean-atmosphere coupling is important.

That is what \*this\* paper should be about. There is no way to evaluate this new coupled model if the model is not used over a domain and for a case that would require coupled ocean and atmosphere capabilities.

Reply: We agree with the reviewer that the heat events do not show the strong capability of a coupled model. However, our aim of this paper is to detail the development of this coupled model and validate the performance. Investigating dynamical processes is outside the scope of this manuscript. We have re-written the abstract and introduction of the manuscript to emphasize our motivation and goal further and make it clear that our objective is not to test the forecast skill of the model in this work,

but instead to present this new coupling model resource. We are doing follow-on work to test the model skill in various regions and scenarios. This is also why we submit this manuscript to Geoscientific Model Development, which considers the publication of code development and technical aspects.

**Reviewer 2:**

The authors of the manuscript coupled an atmospheric model with an oceanic model using ESMF infrastructure. They evaluate the model performance during June 2012, which contained three heat waves over the Arabian Peninsula and the Red Sea. The model simulation lasted one month. The skill was better during the earlier simulation timeframe.

**Major comments**

The paper still needs focusing. I am still struggling to see the main point of the paper: was it a technical challenge that had to be overcome, or was it a science question that could finally be answered with the coupled system? Is the system designed for seasonal prediction, hence a 30-day run? If it is focused on the heat waves, why is the bulk of evaluation over the Red Sea? When resolving a phenomenon that goes through a diurnal cycle, such as air temperature at 2 meter height, isn't it imperative that the boundary conditions can resolve the diurnal cycle? Obtaining daily HYCOM values completely misses the diurnal variability, and SST change can be up to a few degrees Celsius, affecting surface heat fluxes.

Reply: We aim to describe the development of coupled code and use an example case to show the coupled system works. We are introducing a new resource, and we are not trying to answer a specific science question in this paper. We have added a discussion on the aim of this paper in the abstract and the introduction. We aim to validate the coupled ocean-atmosphere model and we demonstrated the capability of this model by showing the surface temperature, SST, heat flux, and wind speed are captured over the ocean compared with validation data. We now present the repository where one may download this resource in the introduction.

We agree that using a 3-hourly boundary condition will better resolve the diurnal cycle. We have replaced the daily boundary condition with 3-hourly values to capture the diurnal SST cycle of HYCOM in the region influenced by this boundary condition.

Please re-read the manuscript carefully before submitting it. The second version still contains typos.

**Reply: Thanks. We have gone through the manuscript carefully this time and fixed the typos.**

**Minor comments**

P1 I12: I am not sure if the coupled system has good skill. Some errors are quite big.

Reply: We aim to describe the development of coupled code and use an example case to show the coupled system works. We agree with the reviewer that the presented test does not show the coupled model has *better* skill than uncoupled simulations. We have revised our manuscript to reflect this.

We also agree that some errors in the heat fluxes at daytime are quite large at a few snapshots. But (1) the time-averaged error is much smaller, and (2) our model error is comparable with the errors reported in the literature by using different WRF radiation schemes (Zempila et al. 2016). We also

**tried different radiation schemes and our conclusions are consistent with the literature. We now plot the time-series of the heat fluxes and have moved the snapshot evaluations to the appendix.**

P7 112: The sequential mode might be simpler to start, but the overall performance of the system in your case will suffer – the ocean component has fewer active (water) points on which the computation is performed, thus affecting load balancing. If discussing scalability of the system, this is not the best choice.

Reply: We have now implemented both concurrent and sequential modes. We agree that the sequential mode suffers and slows down the parallel efficiency. We have revised our manuscript:

The ESMF allows the PETs to run in sequential mode, concurrent mode, or mixed mode (for more than three components). We implemented both the sequential and concurrent mode in SKRIPS, shown in Fig. 2(b). In sequential mode, a set of ESMF gridded/coupler components run in sequence on the same set of PETs. At each coupling time step, the oceanic component is executed when the atmosphere component is completed or vice versa. On the other hand, in concurrent mode, the gridded components are created and run on mutually exclusive sets of PETs.

P8 116: If I understand correctly MITgcm also runs on 256x256 grid, of which a large portion are idle (land points)?

**Reply: Yes. A large portion of the MITgcm ocean model is idle.**

P9 I9: The link to HYCOM data is incorrect - it should read '/dataserver/'.

**Reply: We have now fixed this typo.**

P10 Table1: The typo has not been corrected; it still reads ATM.STA instead of ATM.DYN.

**Reply: We have now fixed this typo.**

P11 I7: For clarity: 'from the model experiments', probably better than 'from various experiments'.

**Reply: We have replaced the original manuscript using 'from the model experiments.**

P11 I10: I am not convinced ERA5 air temperature is in good agreement with NCDC ground observations. Figure 6 indicates discrepancies up to 10 degrees Celsius!

Reply: We thank the reviewer for pointing out this. The difference is 10 degrees because we did not consider the land-sea mask when interpolating the ERA5 air temperature. We have considered the land-sea mask and re-done the interpolations in the revised manuscript.

P11 I15: Why are T2 differences discussed for hours 36 and 48?

Reply: First, the extreme heat event in Mecca is observed on June 2nd, which is 36 hours after the initialization. The T2 after 48 hours are presented to show the diurnal variation of T2 in the Red Sea. In addition, we present the T2 differences after 36 and 48 hours to show the coupled model is consistent with the uncoupled models when the SST are close in the simulations. We have added the discussion in Section 4.1.

P12 Figure 4: The model is too hot over land at daytime (36 hours) and too cold almost everywhere at night (48 hours). The temperature differences are significant!

Reply: We agree that the temperature differs by a few degrees in our simulations. We tried different combinations of WRF schemes and other combinations have much larger errors in T2 compared with the present combinations. We focused on validating the coupled model with respect to the uncoupled atmospheric run give prescribed SST, and we now emphasize this in our manuscript.

P12 I5: The CPL run results are closer to the ERA5 dataset where? Over land, over water?

Reply: The CPL run results are closer to ERA5 over water. We have revised this in the manuscript.

P12 I8: If SST fields from CPL and ATM.DYN are similar, it does not mean `SST in CPL run is tending to be more similar to the realistic'. Please reword.

**Reply: We have removed this sentence.**

P12 I13: NCDC has already been defined. Reply: We have removed the redefinition of NCDC.

P12 I16: Ground observations and ERA5 are not in good agreement according to Figure 6.

Reply: The difference is because we did not consider the land-sea mask when interpolating the ERA5 air temperature. We now consider the land-sea mask and have re-done the interpolations. Now the NCDC observation and ERA5 are consistent.

P13 Figure 5: The surface air temperature is persists to be too warm over land at daytime seen at t=9.5 days and t=23.5 days). Care to comment?

Reply: We agree that the temperature differs by a few degrees in our simulations. We hypothesize this is because WRF land surface schemes do not capture T2 in the desserts in the simulations, but we need detailed numerical tests to defend our claims. Since our aim of this paper is to show the technical development of a new coupled model, we did not investigate the T2 bias on land.

P13 I3: What is RMSE difference? Reply: Yes, it is RMSE, not RMSE difference. We have removed the 'difference' in the manuscript.

P13 I5: Are you implying that the simulation was not capable of capturing the June 2nd event was not captured in Mecca? I do not understand the meaning of the sentence.

Reply: We want to say the daily maximum T2 is not captured well by the models in Mecca, but it is captured in Jeddah and Yanbu. We have replaced the old sentence:

It should be mentioned that both the present simulations and ERA5 reported a T2 that is 2.8 degC cooler than the observed record-high T2 in Mecca on June 2nd. This under-estimation is comparable with the RMSE of the daily high T2 in Mecca (2.25 degC in CPL run).

P13 I7: If initial conditions were at fault, the first day maxima would not be captured. Why is the WRF physics scheme in error – it worked on day 1.

**Reply: We agree with the reviewer that it is not because of the initial condition. We have removed it in our manuscript.**

P13 I8: Physics schemes do not parameterize extreme events. They parameterize physical processes that can not be explicitly resolved in a numerical model.

Reply: We agree with the reviewer. We hypothesize that the physical parameterization schemes used are problematic for extreme events. However, since this paper focuses on the development of the coupled model, we did not check the detailed implementation of the WRF schemes. We have removed this sentence in our manuscript.

P13 I10: ...'T2 compare'... should read 'T2 compared'.

**Reply: We have now fixed this typo.**

P14 Figure 6: Can you mark the heat event episodes with light gray background?

**Reply: We have highlighted the events that we looked at in Fig. 6.**

P14 Figure 6: The discrepancy between ERA5 and observed maxima and minima is troublesome if ERA5 is used to evaluate the model results.

Reply: The difference is because we did not consider the land-sea mask when interpolating the ERA5 air temperature. We now consider the land-sea mask and have re-done the interpolations. After the land-sea mask is considered, the NCDC observation and ERA5 are consistent.

P15 Figure 8: 'air temperature' should read 'air temperature over the Red Sea'. I do not understand why only over water.

Reply: We have replaced 'air temperature' using 'air temperature over the Red Sea' in the caption. We don't use the land surface temperature because we want to investigate the performance of the coupled ocean-atmosphere model.

P15 Figure 8: Can you comment on the bias and rmse time drift of the ATM.STA results?

**Reply: Yes. We have added:**

The bias and RMSE of T2 in the present work are similar to those in the benchmark WRF-ARW simulations [xu et al. 2009, zhang et al. 2013]. The differences of the mean bias and RMSE between the simulations and ERA5 data are also plotted to demonstrate the evolution of the CPL errors compared with ATM.STA and ATM.DYN runs. It can be seen that the CPL run has smaller bias and RMSE than the ATM.STA run throughout the entire simulation.

P16 l6: It appears that the model captures the general meridional SST gradient in the Red Sea. But the actual temperature differences are quite large (Figure 9).

Reply: We have re-done the simulations and used 3-hourly HYCOM data as the boundary condition. The actual temperature difference is much smaller when using 3-hourly data.

P16 I14: I am afraid that daily data is not frequent enough to study diurnal phenomena. I am not familiar with HYCOM data on the aforementioned server, but I would be surprised if a global ocean model provides fields only once a day.

**Reply: We now run the simulations using 3-hourly HYCOM data. The coupled model can capture the diurnal cycle better when driven by the 3-hourly boundary condition.**

P17 I5: 'This is because the HYCOM data is cooler than GHRSST'. Have you considered the difference in HYCOM bulk SST at the topmost model level, and the actual skin SST measured by satellites, reported by GHRSST? Also, the temperatures might be lower, but they don't make the data cooler.

Reply: We have clarified this statement in our manuscript.

**This is because our models are initialized by using HYCOM/NCODA, and the temperature in the topmost model level is cooler than the estimated foundation SST reported by GHRSST.**

P17 I11: Why are surface heat fluxes evaluated only over the Red Sea?

**Reply: We focus on the Red Sea because we are validating our coupled model. We have added the explanation of this in the manuscript.**

P18 I3: CPL and ATM.DYN runs exhibit more latent heat fluxes coming out of the ocean. Why? And reword please.

Reply: We have revised this paragraph by discussing the time series of the latent heat fluxes. The previously discussed snapshots may not be adequate to describe the capability of the coupled model. When comparing the hourly latent heat fluxes, the simulations do not have significantly more latent heat flux compared with the MERRA-II data.

P18 I6: Latent heat, or latent heat fluxes (three times).

**Reply: We have now fixed the typos.**

P18 I8: Why is latent heat flux adequate close to the coastline?

**Reply: We have revised this. The MERRA-2 data have coarsened resolution and coastal region is 'contaminated' by the land points [Kara et al. 2008, Gelaro et al. 2017]. We have ignored the partial land points in the interpolation and re-calculated the heat fluxes.**

P18 I10: I am not convinced the short and longwave radiation fluxes are captured reasonably well. According to Figure 12, the differences are almost up to 30% of the total forcing.

P18 I15: Yes, the differences are almost 200 W/m2!

P18 I17: Points with discrepancies can be masked out to help with the evaluation.

P18 I21: I do not agree with the conclusion that 'overall the CPL simulations are capable of well capturing all the components of the surface heat fluxes'.

Reply: We have removed Fig. 12 from the new manuscript. We agree with the reviewer and have concluded that a snapshot example is inadequate to show the difference of the heat fluxes. Instead, we now calculate the time series of the surface heat flux and show the mean deviation and RMSE between the coupled model and MERRA-II. The RMSE is 105 Wm-2 and higher if one only considers

daytime. We compare our heat flux simulation with the literature [Zempila et al. 2018, Imran et al. 2018]; the uncertainty in the hourly flux can be 50% to 100% for a few widely used WRF schemes. We have also revised our manuscript.

P18 I24: How is latent heat flux different from evaporation?

Reply: We agree that the evaporation is a function of the latent heat flux in the coupled model. Hence, we have moved the evaporation part to the appendix.

P18 I24: Surface winds are winds at 10 meters? Are there any wind observations in the NCDC data? Why not compare sea level pressure? If the pressure field is not correct, that could explain the discrepancy in winds.

Reply: Yes, surface winds are winds at 10 meters and we now mention this in the revised manuscript. Unfortunately, the U10 data is not available from the NCDC data. Yes, we agree that presenting the sea level pressure is also helpful for investigating the difference in surface wind fields. However, here we solely present U10 to illustrate that the coupled model is capable of simulating the surface momentum transfer. We have added this in our revised manuscript.

P18 I32: Please reword: 'is small than'.

**Reply: We have now fixed this typo.**

P21 I5: Please reword: 'This is because the CPL run over-estimated the SST than the ATM.DYN run'.

**Reply: We revised that sentence with 'This is because the SST in CPL run is warmer than that in the ATM.DYN run'.**

P21 I7: Did the model create any precipitation? If there is none observed is not a sufficient reason not to include it in the discussion.

**Reply: The model observed precipitation in the southern Red Sea and the Ethiopian Highlands. But the precipitation is not observed in the three major cities. We agree with the reviewer and we have removed the discussion on the precipitation.**

P22 I9: What are overlap cells? Why does it matter they are 25% of the total cells?

Reply: Section 5 aims to show the coupler is not the bottleneck. Hence, we have removed the discussion of 'overlap cells' and rewritten this paragraph:

When using 256 CPU cores, there are 20480 cells (16 lat x 16 lon x 80 vertical levels) in each core. From results reported in the literature, the parallel efficiency of the coupled model is comparable to other ocean-alone or atmosphere-alone models when having similar number of grid points per CPU core [Marshall et al. 1997, Zhang et al. 2013].

P22 I14: Fluctuations in computational time might be resolved by performing a few runs and averaging the execution times. I am still convinced that idle ocean points when using many cores are seriously affecting load balancing and thus overall performance when using the system sequentially.

Reply: Yes, we agree with the reviewer that the fluctuation of the execution times may be due to the idle ocean points. However, Section 5 aims to show the coupled code has good scalability and the coupler does not slow down the simulation. Hence, we have revised our manuscript and removed the discussion on the fluctuation of CPU time.

P23 I9: 'We hypothesize' not 'we hypothesis'.

**Reply: We have now fixed this typo.**

P23 I11: Your results suggest that using more cores increases the cost of coupling. That is worrysome. Is there an upper limit to this cost?

Reply: Yes, we performed clean tests on another cluster (Shaheen-II in KAUST) and turned off unnecessary I/O. Our old COMPAS cluster is down and we cannot use it. We found that the cost of the coupling will increase only by 3% and the total time spent on the coupler does not increase. We have updated our manuscript.

P24 Table 3: For a clean evaluation of ESMF cost, switch off I/O, because it is handled by each component. Also, use the built-in ESMF Clock, which will tell you exactly where time is being spent.

Reply: We have performed a clean evaluation of ESMF cost by turning off unnecessary I/O. We have updated our results in Section 5 and we have shown that the coupled code does not slow down the ocean and atmosphere simulations.

**SKRIPS v1.0: A regional coupled ocean–atmosphere modeling framework (MITgcm–WRF) using ESMF/NUOPC, description and preliminary results for the Red Sea**

Rui Sun1, Aneesh C. Subramanian1, Arthur J. Miller1, Matthew R. Mazloff1, Ibrahim Hoteit2, and Bruce D. Cornuelle1

1Scripps Institution of Oceanography, La Jolla, California, USA

[revised manuscript text omitted]

---

## Author Response (AR4)

Comments from reviewer 1:
Reply: We acknowledge the reviewer for recommending our work for publication. We are grateful that the reviewer prevented us from making several mistakes in this work.

Technical Corrections:

P1 L12,13
"The coupled model, documentation, and tutorial cases used in this work are available"
What tutorial cases are used in this paper? Either be more specific, or remove this portion of the sentence.
Reply: We have rewritten this sentence in the abstract:
The coupled model and documentation are available at https://library.ucsd.edu/dc/collection/bb1847661c, and the source code is maintained at https://github.com/iurnus/scripps_kaust_model.

P3 L7
"(Skamarock et al., 2005)."
(Skamarock et al., 2019).
P5 L24
"(Skamarock et al., 2005)"
(Skamarock et al., 2019)
Reply: We have updated the reference in the manuscript.

P9 L4,5
"To validate the coupled model, the following sets of simulations using different surface forcings are performed according to the tests (Warner et al., 2010; Turuncoglu et al., 2013; Ricchi et al., 2016)"
Either I don't know what "according to the tests" means, or this should be stated in a more clear fashion. How are these citations related to "according to the tests"?
Reply: We replaced 'according to the tests' using 'according to the validation tests in the literature'.

P10 L1,2
"The ocean model uses the data assimilating HYCOM/NCODA 1/12 global reanalysis data as initial and boundary conditions"
Remove "data assimilating".
Reply: We have removed 'data assimilating' in the revised manuscript.

P10 L10,11
"Here we used the default width"
Everything else is present tense.
Reply: We have replaced 'used' with 'use'. We have also gone through the manuscript to check tense consistency.

P10 L21

"The simulated SST is also validated against HYCOM/NCODA data to show the increase of the error."

No idea what this means.

Reply: We have expanded this sentence:

In addition, the simulated SST fields are validated against HYCOM/NCODA data. Since the simulations are initialized using HYCOM/NCODA data, this aims to show the increase of the error.

P11 L1

http://cdo.ncdc.noaa.gov/CDO/georegion

This link is broken.

Reply: We have updated the link in our manuscript (https://www.ncdc.noaa.gov/cdo-web/).

P11 L5

"Surface heat fluxes (e.g., latent heat, sensible heat, longwave and shortwave radiations), which drives the oceanic component"

drive

Reply: We have replaced 'drives' using 'drive'.

P16 L2

"The SST field snapshots from CPL run on June 2nd and 24th are shown"

Remove "run". The word "run" could be used as a noun or a verb in the sentence, and the meaning of the sentence changes.

Reply: We agree that the word 'run' is misleading in this sentence. We have replaced the old sentence using:

The snapshots of SST obtained from CPL are shown in Fig. 8(I) and Fig. 8(VI).

We have also removed 'run' in other sentences that may mislead the readers.

P16 L12,13

"To quantitatively compare the errors in SST, the time history of the SST in the simulations (i.e., OCN.DYN and CPL) and validation data (i.e., GHRSST and HYCOM/NCODA) are shown in Fig. 9."

Add a sentence describing the reason for the oscillatory 9a vs 9b (3h vs daily data?).

Reply: We have added a sentence discussing the SST oscillation:

Compared with Fig. 9(b), the diurnal SST oscillation can be observed in Fig. 9(a) because the SST is plotted every 3 hours.

We have also added this sentence in the caption of the figure:

Panel (a) shows the 3-hourly SST obtained in the simulations compared with 3-hourly HYCOM/NCODA data. Panel (b) shows the daily SST at 0000 UTC (about 3 A.M. local time in the Red Sea region) obtained in the simulations compared with GHRSST.

P17 Fig9b

"Mean SST: CPL 29.28; OC.,DYN 29.29; HYCOM 29.85"

"HYCOM" should be "GHRSST"

Reply: We have replaced 'HYCOM' using 'GHRSST' in the figure. We have also gone through out figures and fixed other typos.

P23 L3

"The coupled model is validated by using a realistic application to simulate the heat events in the Red Sea region. "

This is the conclusion so be specific in case readers read only this and the abstract, "The coupled model is validated by using a realistic application to simulate heat events during June 2012 in the Red Sea region."

Reply: We have revised our conclusion to make it consistent with the abstract.

P23 L15,16

"The CPU time associated with different components of the coupled simulations is also presented, showing the ESMF/NUOPC driver is not a bottleneck in the computation. "

Remove the "is also presented" and state directly "The CPU time associated with different components of the coupled simulations shows the ESMF/NUOPC driver is not a bottleneck in our coupled implementation of SKRIPS. "

Reply: We have removed 'is also presented' and stated our conclusions more directly:

The CPU time associated with different components of the coupled simulations shows that the ESMF/NUOPC driver does not add a significant computational overhead.

P24 L1,2

"Appendix A: Snapshots of Surface Heat Fluxes

The snapshots of the THFs in the simulations at 1200 UTC June 2nd and 24th are presented."

Would you redefine THF in the appendix or do something similar to changing the title of the section to "Snapshots of Surface Turbulent Heat Fluxes".

Reply: Actually, we present both THFs and solar radiations in Appendix A. Hence, we have added a paragraph in this appendix to make it clear:

To examine the modeling of turbulent heat fluxes, shortwave radiation, and longwave radiation, the snapshots of these heat fluxes obtained from ATM.STA, ATM.DYN, and CPL are presented and validated using the MERRA-2 data.

P24 L2

"The snapshots of the THFs in the simulations at 1200 UTC June 2nd and 24th are presented."

Maybe add "(Fig. A1)" at the end of the sentence. You don't mention A1 until you discuss A3, and then only tangentially.

Reply: We have added 'Fig. A1' at the end of the sentence.

P31 L33,34

"Skamarock, W. C., Klemp, J. B., Dudhia, J., Gill, D. O., Barker, D. M., Wang, W., and Powers, J. G.: A description of the advanced research WRF version 2, Tech. rep., National Center For Atmospheric Research Boulder Co Mesoscale and Microscale Meteorology Div, 2005."
Skamarock, W. C., J. B. Klemp, J. Dudhia, D. O. Gill, Z. Liu, J. Berner, W. Wang, J. G. Powers, M. G. Duda, D. M. Barker, and X.-Y. Huang, 2019: A Description of the Advanced Research WRF Version 4. NCAR Tech. Note NCAR/TN-556+STR, 145 pp. doi:10.5065/1dfh-6p97
Reply: Thanks. We have updated the reference in the manuscript.

Comments from reviewer 2:
Thank you for substantially improving the manuscript. I have a few comments, which I hope will help you add clarity. Please add in the conclusion that your coupled system gives equal or better results than the standalone components by themselves.
Reply: We have added it in the conclusion.

I would suggest adding some explanation when introducing the model evaluation section. Air temperature at 2 m is used to compare CPL with ATM.STA and ATM.DYN, for validation of the atmospheric component. Is the coupled WRF giving similar or better results than the coupled WRF in SKRIPS. SST is similarly used to evaluate the oceanic component, coupled versus standalone. In both cases you see the results are equal or better, which is desirable. Winds and fluxes are used to assess the coupled system.
Reply: Agree. We have added some explanations when introducing the model evaluation section:
The T2 from CPL, ATM.STA, and ATM.DYN are compared with the validation data to evaluate the atmospheric component of SKRIPS; the SST obtained from CPL and OCN.DYN are used to validate the atmospheric component of SKRIPS; the surface heat fluxes and 10-m wind are used to assess the coupled system.

There are still mistakes in the manuscript that would be picked up by a native speaker (e.g. P11, L10: Usually the model data is interpolated to validation data points. Is there a good reason you are doing the opposite?
Reply: When we compare to direct observations in three major cities (Jeddah, Mecca, Yanbu), we interpolate the model output to the observation stations. When comparing with MERRA-2, ERA5, GHRSST, and HYCOM data, we interpolate disparate gridded data data onto the same high-resolution grid used by the model. We have added them to the manuscript:
When comparing T2 with NCDC ground observations, the simulation results and ERA5 are interpolated to the NCDC stations. When interpolating to NCDC stations near the coast, only the data saved on land points are used. The maximum/minimum T2 every 24 hours from the simulations and ERA5 are compared to the observed daily maximum/minimum T2. On the other hand, when comparing the simulation results with the analysis or reanalysis data (HYCOM, GHRSST, ERA5 and MERRA-2), we interpolate these validation data onto the model grid to achieve a uniform spatial scale (Maksyutov et al. 2008, Torma et al. 2011).

P12, L14: I believe RMSE appears here for the first time, so please move the definition from P14, L3 here.

Reply: We have moved the definition of RMSE to Page 12.

P13, L2: Replace "warming SST, which warms the T2", with "a dynamic SST which influences T2", or something similar. There is warming and cooling of SST in a diurnal cycle which affects T2.

Reply: We agree that there are warming and cooling SST. We have replaced our original manuscript using "a dynamic SST which influences T2".

P14, Fig 6: What is 'v.s.' – did you mean versus? If so, I believe the short version is 'vs'. Or simply replace labels with 'RMSE of maximum temperatures' and 'RMSE of minimum temperatures'.

Reply: We have replaced 'v.s.' using 'RMSE of maximum T2' and 'RMSE of minimum T2'.

P15, L1: So, you are evaluating capturing the diurnal cycle over land using the three locations (Jeddah, Mecca, Yanbu), and over water using points over the Red Sea. That could be written more concisely.

Reply: We have re-organized the discussions of the diurnal cycle to make it more concise. Now Fig. 6 discusses the evolution of errors over the points over the Red Sea; Fig. 7 discusses the validation of T2 over three major cities.

P17, L12: I believe it is the 'atmospheric surface heat flux'. Many more instances in the text. Please have a native speaker proof read.

Reply: We have replaced 'atmosphere surface heat flux' using 'atmospheric surface heat flux' in the manuscript. We have proof read this manuscript.

P18, Fig 10: Where is ATM.DYN in the top panel? If it is very similar to CPL, please explain in the text.

Reply: We have added ATM.DYN results in the top panel.

P18, L13: Please reword: "because they are initialized in the same way the SST fields are similar".

Reply: We have replaced the old sentence using:
This is because all simulations are initialized in the same way, and the SST in all simulations are similar in the first two weeks.

P18, L17: Radiations -> Radiation (throughout the text).

Reply: We have replaced 'radiations' using 'radiation' throughout the text.

P19, L4: Please reword 'slightly over-estimated', if the difference can be in excess of 100 W/m^2.

Reply: We have removed the word 'slightly' in the manuscript.

P19, Fig 11: Where is ATM.DYN?
P20, Fig13: Where is ATM.DYN?
Reply: We have added ATM.DYN results to both figures in the top panel.

P21, L15-25: Did you run the system in sequential or concurrent mode? I think I know.
Reply: We run the simulations in sequential mode. We have added this in the manuscript.

P21, L25: The cost breakdown among the component should provide you with a good first guess to address load balancing in the concurrent mode.
Reply: Agree. We have added this sentence in this paragraph:
The most time-consuming process is the atmospheric model integration, which accounts for 85% to 95% of the total cost. The ocean model integration is the second most time-consuming process, which is 5% to 11% of the total computational cost. If a purely marine region is selected in an ideal case, the cost of ocean and atmosphere models would be more equal compared with the Red Sea case. It should be noted that the test cases are run in sequential mode, and the cost breakdown among the components can be used to address load balancing in the concurrent mode.

P21, L27: It is computational cost, not costs. A few more instances on the same page, please have a native speaker proofread the manuscript.
Reply: We have replaced 'costs' using 'cost' in this section. We have proof read this manuscript.

P21, L31-32: Please remove the statement above the 'purely marine region'. There are many factors determining the cost which you do not explain (e.g. degrees of freedom).
Reply: We have removed the statement above the 'purely marine region'.

P22, L3: Please replace "is not a bottleneck", with something like "does not add a significant computational overhead".
Reply: We have replaced 'is not a bottleneck' using 'does not add a significant computational overhead'.

P23, L17: The system has not been used on thousands of cores to suggest it can be used on a supercomputer. In fact, I would argue you showed that the system can be run on a commodity cluster with 500-1000 cores, provided that the problem size is adequate.
Reply: We have replaced 'supercomputer' using 'HPC cluster' in this sentence.

[revised manuscript text omitted]

2-m air temperature (T2) fields are validated using ERA5. In addition, the simulated T2  for three major cities near the eastern shore of the Red Sea are validated using ERA5 and  ground observations from the NOAA National Climate Data

5   Center (NCDC climate data online at https://www.ncdc.noaa.gov/cdo-web/). The simulated SST data are validated against the OSTIA (Operational Sea Surface Temperature and Sea Ice Analysis) system in GHRSST (Group for High Resolution Sea Surface Temperature) (Donlon et al., 2012; Martin et al., 2012). In addition, the

10  simulated SST fields are validated against HYCOM/NCODA data. Since the simulations are initialized using HYCOM/NCODA data, this aims to show the increase of the differences. Surface heat fluxes (e.g., turbulent heat flux and radiative flux), which  drive the oceanic component in the coupled model, are validated using MERRA-2 (Modern-Era Retrospective analysis for Research and Applications,

15  version 2) data (Gelaro et al., 2017).  We use MERRA-2 data  because (1) it is an independent reanalysis data compared to the initial and boundary conditions used in the simulations, and (2) it also provides  $0.625^o \times 0.5^o$ (lon $\times$ lat) resolution reanalysis fields of turbulent heat fluxes (THF). The 10-m wind speed is also compared with MERRA-2 data to validate the momentum flux in the coupled code.

20   The validation of the freshwater flux is shown in the Appendix because (1) the evaporation is proportional to the latent heat in the model and (2) the precipitation is zero in  the cities near the coast in Fig. 3. The validation data are summarized in Table 2.

When comparing T2 with NCDC ground observations, the simulation results and ERA5 are interpolated to the NCDC stations. When interpolating to NCDC stations near the coast, only the data saved on land points are used[3]. The maximum/minimum T2 every 24 hours from the simulations and ERA5 are compared to the observed daily maximum/minimum T2. On the other hand, when comparing the simulation results with the analysis or reanalysis data (HYCOM, GHRSST, ERA5, and MERRA-2), we interpolate these data onto the model grid to achieve a uniform spatial scale (Maksyutov et al., 2008; Torma et al., 2011).

**Table 2.** The observational data and reanalysis data used to validate the simulation results.

| variable | validation data |
| --- | --- |
| sea surface temperature (SST) | GHRSST and HYCOM/NCODA |
|  2-m air temperature (T2) | ERA5 and NCDC climate data |
| turbulent heat fluxes | MERRA-2 |
| radiative fluxes | MERRA-2 |
|  10-m wind speed | MERRA-2 |

**4   Results and Discussions**

The Red Sea is an elongated basin covering the area between 12-30°N and 32-43°E. The basin is 2250 km long, extending from the Suez and Aqaba gulfs in the north to the strait of Bab el-Mandeb in the south, which connects the Red Sea and the Indian Ocean. In this section, the simulation results obtained by using different model configurations are presented to show that SKRIPS is capable of performing coupled ocean–atmosphere simulations. The T2 from CPL, ATM.STA, and ATM.DYN are compared with the validation data to evaluate the atmospheric component of SKRIPS; the SST obtained from CPL and OCN.DYN are compared to validate the atmospheric component of SKRIPS; the surface heat fluxes and 10-m wind are used to assess the coupled system.

**4.1    2-m Air Temperature**

We begin our analysis by examining the simulated T2 from the model experiments, aiming to validate the atmospheric component of SKRIPS. Since the record-high temperature is observed in the Makkah region on June $2^{nd}$, the simulation results on June $2^{nd}$ (36 or 48 hours after the initialization) are shown in Fig. 4. The ERA5 data, and the difference between CPL  and ERA5 are also shown in Fig. 4. It can be seen in Fig. 4(I) that  CPL captures the T2 patterns in the Red Sea region on June $2^{nd}$  compared with ERA5 in Fig. 4(II). Since the ERA5  T2 data are in good agreement with the NCDC ground observation data in the Red Sea region (detailed  comparisons of all stations are not shown), we use ERA5 data to validate the simulation results. The difference between  CPL and ERA5 is shown in Fig. 4(III).
* * *
[3]In ATM.STA, ATM.DYN, and CPL, we use land-sea mask=1 as land points because the land-sea mask is either 0 (sea) or 1 (land) in WRF. In ERA5, we use land-sea mask > 0.9 as land points because the land-sea mask is a fractional value between 0 (sea) to 1 (land).

The ATM.STA and ATM.DYN  results are close to the CPL  results and thus are not shown, but their differences with respect to ERA5 are shown in Fig. 4(IV) and 4(V), respectively. Fig. 4(VI) to 4(X) show the nighttime results after 48

5   hours. It can be seen in Fig. 4 that all simulations reproduce the T2 patterns over the Red Sea region reasonably well compared with ERA5. The mean T2 biases and  root mean square errors (RMSEs) over the sea are shown in Table 3. The biases of the T2 are comparable with  those reported in other  benchmark WRF simulations (Xu et al., 2009; Zhang et al., 2013a; Imran et al., 2018).

[Figure]

**Figure 4.** The 2-m air temperature as obtained from the CPL, the ERA5 data, and their difference (CPL−ERA5). The differences between ATM.STA and ATM.DYN with ERA5 (i.e., ATM.STA−ERA5, ATM.DYN−ERA5) are also presented. The simulation initial time is 0000 UTC Jun 01 2012 for both snapshots. Two snapshots are selected: (1) 1200 UTC Jun 02 2012 (36 hours from initial time); (2) 0000 UTC Jun 03 2012 (48 hours from initial time). The results on Jun 02 are presented because the record-high temperature is observed in the Makkah region.

10   The simulation results on June 10th and 24th are shown in Fig. 5 to validate the coupled model over longer periods of time.  It can be seen in Fig. 5  that the T2 patterns  in CPL are generally consistent with ERA5. The differences between the simulations (CPL, ATM.STA and ATM.DYN) and ERA5  shows that the T2  on land are consistent for all three simulations.

15   However, the T2 over the sea in CPL  has smaller mean biases and RMSEs compared with  ATM.STA, also shown in Table 3. Although the difference  in T2 is very small compared with the mean T2 (31.92 °C), the improvement of the coupled run on the 24th (1.02 °C) is comparable to the standard deviation of T2 (1.64 °C). The T2 over the

water in  CPL is closer to ERA5 because MITgcm in the coupled model provides  a dynamic SST which influences T2 . On the other hand, when comparing  CPL with ATM.DYN, the mean difference is smaller (10th: +0.04 °C; 24th: -0.62 °C). This shows  CPL is comparable to  ATM.DYN  which is driven by an updated warming SST.

[Figure]

**Figure 5.** The  T2 obtained  in CPL, the T2 in ERA5, and their difference (CPL−ERA5). The difference between ATM.STA and ATM.DYN with ERA5 data (i.e., ATM.STA−ERA5, ATM.DYN−ERA5) are also presented. The simulation initial time is 0000 UTC Jun 01 2012 for both snapshots. Two snapshots are selected: (1) 1200 UTC Jun 10 2012 (9.5 days from initial time); (2) 1200 UTC Jun 24 2012 (23.5 days from initial time).

**Table 3.** The biases and RMSEs of  T2 simulated in all simulations in comparison with ERA5 data.

|  | after 36 hours | after 48 hours | after 9.5 days | after 23.5 days |
|---|---|---|---|---|
|  Run CPL | bias: -1.36; RMSE:  1.91 | bias: -0.82; RMSE:  1.19 | bias: -1.24; RMSE:  1.96 | bias: -0.81; RMSE:  1.8 |
| Run ATM.STA  | bias: -1.48; RMSE:  2.01 | bias: -0.92; RMSE:  1.27 | bias: -1.56; RMSE:  2.27 | bias: -1.83; RMSE:  2.5 |
| Run ATM.DYN  | bias: -1.36; RMSE:  1.90 | bias: -0.84; RMSE:  1.28 | bias: -1.20; RMSE:  1.93 | bias: -1.43; RMSE:  2.1 |

The mean biases and RMSEs of T2 over the Red Sea during the 30-day simulation are shown in Fig. 6 to demonstrate the evolution of simulation errors. It can be seen that ATM.STA can still capture the T2 patterns in the first week, but it

under-predicts T2 by about 2 °C after 20 days because it has no SST evolution. On the other hand, CPL has smaller bias (-0.60 °C) and RMSE (1.28 °C) compared with those in ATM.STA (bias: -1.19 °C; RMSE: 1.71 °C) during the 30-day
5   simulation as the SST evolution is considered. The ATM.DYN case also has smaller error than ATM.STA and its error is comparable with that in CPL (bias: -0.72 °C; RMSE: 1.31 °C), indicating that the skill of the coupled model is comparable to the stand-alone atmosphere model driven by 3-hourly reanalysis SST. The differences of the mean biases and RMSEs between model outputs and ERA5 data are also plotted in Fig. 6. It can be seen that CPL has smaller error than ATM.STA throughout the simulation. The bias and RMSE between CPL and ATM.DYN are within about 0.5 °C. This shows 
[revised manuscript text omitted]
.  Compared with Fig. 9(b), the diurnal SST oscillation can be observed in Fig. 9(a) because the SST is plotted every 3 hours. Generally, OCN.DYN and CPL  have a similar range of error compared to both validation datasets in the 30-day simulations. The simulation results are compared with HYCOM/NCODA data to show the increase of RMSE in

[Figure]

**Figure 8.** The  SST  in CPL, the SST in OCN.DYN, and GHRSST. The corresponding differences between the simulations and  GHRSST are also plotted. Two snapshots of the model outputs are selected: (1) 0000 UTC Jun 02 2012; (2) 0000 UTC Jun 24 2012. The simulation initial time is 0000 UTC Jun 01 2012 for both snapshots.

Fig. 9(a). Compared with HYCOM/NCODA, the mean differences between CPL and OCN.DYN  are small (CPL: 0.10 °C; OCN.DYN: 0.03 °C). The RMSE increases in the first week, but does not grow after that. On the other hand, when comparing

5   with the GHRSST, the initial SST patterns in both runs are cooler by about 0.8 °C. This is because  the models are initialized by  HYCOM/NCODA,  which has temperature in the topmost model level  cooler than the estimated foundation SST reported by GHRSST. After the first 10 days, the difference between GHRSST data and HYCOM/NCODA decreases, and likewise the difference between the simulation results and GHRSST also decreases. It should be noted that the SST simulated by  CPL has smaller error (bias: -0.57 °C; RMSE: 0.69 °C) compared with OCN.DYN (bias: -0.66 °C; RMSE:

10   0.76 °C) by about 0.1 °C when validated using GHRSST. This indicates the coupled model can adequately simulate the SST evolution compared with the uncoupled model forced by ERA5 reanalysis data.

**4.3   Surface Heat Fluxes**

The  atmospheric surface heat flux drives the oceanic component in the coupled model, hence we validate the heat fluxes in the coupled model as compared to the stand-alone simulations. Both the turbulent heat fluxes and the net downward heat fluxes are compared to MERRA-2 and their differences are plotted. To validate the coupled ocean–atmosphere model, we only compare the heat fluxes over the sea.

[Figure]

**Figure 9.** The bias and RMSE between the SST from the simulations (i.e., OCN.DYN and CPL) in comparison with the validation data. Panel (a) shows the  3-hourly SST obtained in the simulations compared with 3-hourly HYCOM/NCODA data. Panel (b) shows the  daily SST at 0000 UTC (about 3 A.M. local time in the Red Sea region) obtained in the simulations compared with GHRSST.  Both simulations are initialized at 0000 UTC Jun 01 2012.

The turbulent heat fluxes (THF; sum of latent and sensible heat fluxes) and their differences with the validation data are shown in Fig. 10 (the snapshots are shown in the Appendix). It can be seen

5  that all simulations have similar mean THF over the Red Sea compared with MERRA-2 (CPL: 119.4 W/m$^2$; ATM.STA: 103.4 W/m$^2$; ATM.DYN: 117.5 W/m$^2$; MERRA-2: 115.6 W/m$^2$). For the first two weeks, the mean THFs obtained in  all simulations are overlapping in Fig. 10. This is because all simulations are initialized in the same way, and the SST in all simulations are similar in the first two weeks. After the second week,  CPL has

10 smaller error (bias: -1.8 W/m$^2$; RMSE: 69.9 W/m$^2$) compared with  ATM.STA  (bias: -25.7 W/m$^2$; RMSE: 76.4 W/m$^2$). This is because the SST is updated in  CPL and is warmer compared with ATM.STA. When forced by a warmer SST, the evaporation increases (also see the Appendix) and thus the latent heat fluxes increase. On the other hand, the THFs in  CPL are comparable with  ATM.DYN during the 30-day run (bias: 1.9 W/m$^2$), showing  that SKRIPS can capture the THFs over the Red Sea in the coupled simulation.

15 The net downward heat fluxes (sum of THF and radiative flux) are shown in Fig. 11 (the snapshots are shown in the Appendix). Again, for the first two weeks, the heat fluxes obtained in  ATM.STAand all simulations exhibit similar heat flux patterns because they, ATM.DYN and CPL are overlapping. This is because all simulations are initialized in the same way, and the SST in all

[Figure]

**Figure 10.** The turbulent heat fluxes out of the sea obtained in CPL, ATM.STA, and ATM.DYN in comparison with MERRA-2. The top panel shows the mean THF; the middle panel shows the mean bias; the bottom panel shows the RMSE. Only the hourly heat fluxes over the sea are shown .

simulations are similar in the first two weeks. After the second week,  CPL has slightly smaller error (bias: 11.2 W/m$^2$; RMSE: 84.4 W/m$^2$) compared with the ATM.STA

5 simulation (bias: 36.5 W/m$^2$; RMSE: 94.3 W/m$^2$). It should be noted that the mean bias and RMSE of the net downward heat fluxes can be as high as a few hundred W/m$^2$ or 40% compared with MERRA-2. This is because WRF over-estimated the shortwave  radiation in the daytime (the snapshots are shown in the Appendix). However, the coupled model still captures the mean and standard deviation of the heat flux compared with MERRA-2 data (CPL mean: 110.6 W/m$^2$, standard deviation: 350.7 W/m$^2$; MERRA-2 mean 104.7 W/m$^2$, standard deviation 342.3 W/m$^2$).

10 The over-estimation of shortwave radiation by the RRTMG scheme is also reported in other validation tests in the literature under all-sky conditions due to the uncertainty of cloud or aerosol (Zempila et al., 2016; Imran et al., 2018). Although the surface heat flux is  over-estimated  in the daytime, the SST over the Red Sea is not over-estimated (shown in Section 4.2).

**4.4    10-m Wind Speed**

To evaluate the simulation of the surface momentum by the coupled model, the 10-m wind speed patterns obtained from

15 ATM.STA, ATM.DYN, and CPL  are compared to the MERRA-2 reanalysis.

The simulated 10-m wind velocity fields are shown in Fig. 12. The RMSE of the wind  speed between CPL and MERRA-2 data is 2.23 m/s when using the selected WRF physics schemes presented in Section 3. On June 2$^{nd}$,  high wind speeds are observed in the northern and central Red Sea, and both  CPL

[Figure]

**Figure 11.** The total surface heat fluxes into the sea obtained in CPL, ATM.STA, and ATM.DYN in comparison with MERRA-2. The top panel shows the mean surface heat flux; the middle panel shows the mean bias; the bottom panel shows the RMSE. Only the heat fluxes over the sea are shown.

and ATM.STA  capture the features of the wind speed patterns. On June 24th,  high wind speeds are observed in the central Red Sea and  are also captured by both CPL and ATM.STA. The mean 10-m wind speed

[revised manuscript text omitted]

10  than  ATM.STA  over the simulation period. The SST obtained in CPL  is also better than  that in OCN.DYN  by about 0.1 °C when compared with GHRSST.

The parallel efficiency of the coupled model is examined by simulating the Red Sea region using increasing  numbers of CPU cores. The parallel efficiency of the coupled model is consistent with that of the stand-alone ocean and atmosphere models  using the same number of cores. The CPU time associated with

15  different components of the coupled simulations  shows that the ESMF/NUOPC driver  does not add a significant computational overhead. Hence the coupled model can be implemented for coupled regional modeling studies on  HPC clusters with comparable performance as that attained by uncoupled stand-alone models.

The results presented here motivate further studies evaluating and improving this new regional coupled ocean–atmosphere

20  model for investigating dynamical processes and forecasting applications. This regional coupled forecasting system can be improved by developing coupled data assimilation capabilities for initializing the forecasts. In addition, the model physics and model uncertainty representation in the coupled system can be enhanced using advanced techniques, such as stochastic physics parameterizations. Future work will involve exploring these and other aspects  a regional coupled modeling system suited for forecasting and process understanding.

25  *Code and data availability.* The coupled model, documentation, and the cases used in this work are available at https://library.ucsd.edu/dc/collection/bb1847661c, and the source code is maintained on Github https://github.com/iurnus/scripps_kaust_model. ECMWF ERA5 data are used as the atmospheric initial and boundary conditions. The ocean model uses the assimilated HYCOM/NCODA $1/12°$ global analysis data as initial and boundary conditions. To validate the simulated SST data, we use the OSTIA (Operational Sea Surface Temperature and Sea Ice Analysis) system in GHRSST (Group for High Resolution Sea Surface Temperature). The simulated 2-m air temperature (T2) is

30  validated against the ECMWF ERA5. The observed daily maximum and minimum temperatures from NOAA National Climate Data Center is used to validate the T2 in three major cities. Surface heat fluxes (e.g., latent heat fluxes, sensible heat fluxes, and longwave/shortwave radiation) are compared with MERRA-2 (Modern-Era Retrospective analysis for Research and Applications, version 2).

**Appendix A:  Snapshots of  Heat Fluxes**

To examine the modeling of turbulent heat fluxes and radiative fluxes, the snapshots of these heat fluxes obtained from ATM.STA, ATM.DYN, and CPL are presented and validated using the MERRA-2 data.

5    The snapshots of the THFs  at 1200 UTC June 2$^{nd}$ and 24$^{th}$ are presented  in Fig. A1. It can be seen that all simulations reproduce the THFs reasonably well in comparison with MERRA-2. On June 2$^{nd}$, all simulations exhibit similar THF patterns. This is because all simulations have the same initial conditions, and the SST fields in all simulations are similar within two days. On the other hand, for the heat event on June 24$^{th}$, CPL and ATM.DYN  exhibit more latent heat fluxes coming out of the ocean (170 and 153 W/m$^2$) than  those in ATM.STA  (138 W/m$^2$).

[revised manuscript text omitted]